

# Land cover and its transformation in the backward trajectory footprint region of the Amazon Tall Tower Observatory

Christopher Pöhlker[1,#], David Walter[1,#], Hauke Paulsen[2,#], Tobias Könemann[1], Emilio Rodríguez-Caballero[1,a], Daniel Moran-Zuloaga[1], Joel Brito[3,b], Samara Carbone[3,c], Céline Degrendele[4], Viviane R. Després[2], Florian Ditas[1], Bruna A. Holanda[1], Johannes W. Kaiser[1], Gerhard Lammel[1,4], Jošt V. Lavrič[5], Jing Ming[1], Daniel Pickersgill[2], Mira L. Pöhlker[1], Maria Praß[1], Nina Ruckteschler[1], Jorge Saturno[1d], Matthias Sörgel[1], Qiaoqiao Wang[1,e], Bettina Weber[1], Stefan Wolff[1], Paulo Artaxo[3], Ulrich Pöschl[1], and Meinrat O. Andreae[1,6]

[1]*Multiphase Chemistry, Biogeochemistry, and Air Chemistry Departments, Max Planck Institute for Chemistry, 55020 Mainz, Germany.*

[2] *Institute of Molecular Physiology, Johannes Gutenberg University, 55128 Mainz, Germany.*

[3] *Institute of Physics, University of São Paulo, São Paulo 05508-900, Brazil.*

[4] *Research Centre for Toxic Compounds in the Environment, Masaryk University, Faculty of Sciences, 625 00 Brno, Czech Republic.*

[5] *Department of Biogeochemical Systems, Max Planck Institute for Biogeochemistry, 07701 Jena, Germany.*

[6] *Scripps Institution of Oceanography, University of California San Diego, La Jolla, CA 92037, USA.*

[a] *now at: Department of Agronomy, Universidad de Almería, Spain.*

[b] *now at: Laboratory for Meteorological Physics, University Blaise Pascal, Clermont-Ferrand, France.*

[c] *now at: Federal University of Uberlândia, Uberlândia-MG, 38408-100, Brazil.*

[d] *now at: Physikalisch-Technische Bundesanstalt, Bundesallee 100, D-38116 Braunschweig, Germany.*

[e] *now at: Institute for Environmental and Climate Research, Jinan University, China.*

[#] *These authors contributed equally to this study.*

*Correspondence to:* C. Pöhlker (c.pohlker@mpic.de)



**Abstract**

The Amazon rain forest experiences the combined pressures from man-made deforestation and progressing climate change, causing severe and potentially disruptive perturbations of the ecosystem's integrity and stability. To intensify research on critical aspects of Amazonian biosphere-atmosphere exchange, the Amazon
5  Tall Tower Observatory (ATTO) has been established in the central Amazon Basin. Here we present a multi-year analysis of backward trajectories to derive an effective footprint region of the observatory, which spans large parts of the particularly vulnerable eastern basin. Further, we characterize geospatial properties of the footprint regions, such as climatic conditions, distribution of ecoregions, land cover categories, deforestation dynamics, agricultural expansion, fire regimes, infrastructural development, protected areas, as well
10  as future deforestation scenarios. This study is meant to be a resource and reference work, helping to embed the ATTO observations into the larger context of man-made transformations of Amazonia. We conclude that the chances to observe an unperturbed rain forest-atmosphere exchange will likely decrease in the future, whereas the atmospheric signals from man-made and climate change-related forest perturbations will likewise increase in frequency and intensity.



## 1  Introduction

The Earth is increasingly shaped by human activities (Crutzen, 2002). Concerning the atmosphere, global climate change and air quality impacts on human health are two of the most important recent consequences (e.g., Stocker et al., 2013; Lelieveld et al., 2015; Cheng et al., 2016; Reinmuth-Selzle et al., 2017). The Amazon rain forest and its atmosphere are particularly vulnerable, since they are experiencing the combined pressures from man-made deforestation and progressing climate change (Lenton et al., 2008; Malhi et al., 2008). Davidson et al. (2012) presented comprehensive perspectives on the ecological and atmospheric "transition" of the Amazon biome due to continuous land use change and a cascade of related perturbations and feedbacks. Particularly, the hydrological cycle with its large amounts of recycled water and energy represents an Achilles' heel in the ecosystem's integrity and stability (e.g., Andreae et al., 2004; Rosenfeld et al., 2008; Hilker et al., 2014; Machado et al., 2017). The Amazon is defined by a pronounced continental gradient in climatic conditions, socioeconomic activities, as well as land use change. Climatically, the northwestern part is characterized by high precipitation rates with comparatively weak seasonal amplitudes, whereas the southeastern part experiences a much stronger seasonality, associated with dry season drought stress for the vegetation (e.g., Malhi et al., 2008; 2009). Socioeconomically, the northwestern part is protected by its remoteness and, therefore, still mostly unperturbed, whereas the southeast is heavily influenced by infrastructure development, logging, and agro-industrial expansion (e.g., Soares-Filho et al., 2006; Nepstad et al., 2008; Silva et al., 2013). The regional and global consequences of the Amazon's transition process for the Earth's climate system, water resources, biodiversity, and human health are still widely unknown.

To address the mechanisms and consequences of the anthropogenic perturbation of the Earth atmosphere, a sound understanding of the starting point – the background state – of this transition process is required. However, regions of definable background state conditions are becoming increasingly rare worldwide (Andreae, 2007; Hamilton et al., 2014). To some extent, the Amazon Basin still represents one of the last continental exceptions and, thus, a unique outdoor laboratory for atmospheric science. Certain – although short – episodes in its clean wet season still open a window into the pre-industrial and unpolluted past, while the dry season is influenced by heavy pollution from numerous deforestation and land management fires (Martin et al., 2010b; Andreae et al., 2015; M. Pöhlker et al., 2017b). The Amazon Tall Tower Observatory (ATTO) has been established in the Amazon Basin for two main reasons: First, it supports a better understanding of key processes in biosphere-atmosphere exchange and, therefore, helps to assess the global relevance of the Amazon's ecosystem services. For this task, the frequent occurrence of very clean episodes provides crucially important baseline data to approximate the era before globally pervasive anthropogenic pollution. Referring to Andreae et al. (2015) it is "urgent to obtain baseline data now, to document the present [...] conditions before upcoming changes, especially in the eastern part of the basin, will forever change the face of Amazonia". Second, the ATTO research documents the progressing change in the Amazon and, thus, provides essential knowledge to try to avoid irreversible damage to this unique ecosystem. The extent and complexity of meteorological, trace gas, aerosol, and ecological studies at the ATTO site are





steadily increasing, promising more and more insights into the manifold facets of biogeochemical and hydrological cycles in this unique ecosystem (e.g., Nölscher et al., 2016; Rizzolo et al., 2016; Wang et al., 2016a; Wang et al., 2016b; Chor et al., 2017; Oliveira et al., 2017a; Yañez-Serrano et al., 2017).

Detailed knowledge on the spatial and temporal variability of the site's footprint region and, thus, the

effectively probed land cover mosaic is a prerequisite to embed atmospheric observations at ATTO into a broader Amazonian context. In the course of our recent studies, the analysis of backward trajectories (BTs) and geographic information system (GIS) data helped substantially to explore air mass history and the variability of atmospheric composition (i.e., Moran-Zuloaga et al., 2017b; Pöhlker et al., 2017a; Saturno et al., 2017a; 2017b). Along these lines, the current study presents a systematic BT and GIS data analysis, provid-

ing a robust characterization of spatiotemporal patterns in advection of air masses of the early ATTO years, relevant hydrological regimes, and current land use patterns and future trends in the ATTO footprint region. We envision that this work may serve as a helpful resource and look-up reference for the interpretation of current and future observations in the region. Furthermore, we conclude with a discussion on anticipated future developments within the ATTO footprint region in response to progressing climate and land use change,

which are influences of crucial importance for future ATTO research.





## 2 Methods and data analysis

### 2.1 The ATTO site in the central Amazon Basin

The remote ATTO site (position of the 80-m walk-up tower: 2.1441° S, 58.9999° W, 130 m above sea level) is located about 150 km northeast of Manaus, Brazil. It has been established in 2010/11 as a long-term research station for aerosol, trace gas, meteorological, and ecological studies in the Amazon forest. For detailed information we refer the reader to Andreae et al. (2015). Besides ATTO, the ZF2 site (2.59454° S, 60.20929° W, 90 m above sea level), which is located 60 km north-northwest of Manaus, has served as a key research station for atmospheric and ecological studies in the central Amazon Basin since 2008. Details regarding the ZF2 site can be found in Martin et al. (2010a). During the GoAmazon2014/5 field campaign, both sites served as background stations (called T0a and T0z), providing comprehensive data on the atmospheric state upwind of the Manaus city plume (Martin et al., 2016a; 2016b).

### 2.2 HYSPLIT backward trajectories

The systematic backward trajectory (BT) analysis is based on the Hybrid Single-Particle Lagrangian Integrated Trajectory model (HYSPLIT, NOAA-ARL) with meteorological input data from the global data assimilation system (GDAS1, 1° resolution) (Draxler and Hess, 1998; Stein et al., 2015). The HYSPLIT model is a hybrid between Eulerian and Lagrangian approaches (Draxler and Hess, 1998). In an Eulerian model, air concentrations are calculated by the integration of mass fluxes in each grid cell, based on their diffusion, advection, and local processes. In a Lagrangian model, air concentrations are computed by summing virtual air parcels of zero volume (so-called 'particles'), which are advected through a grid cell along its trajectory (e.g., Escudero et al., 2006).

As a basic data set, fourteen-day BTs were calculated every 1 h for different starting heights at ATTO for a multi-year period, namely 1 Jan 2008 until 30 Jun 2016.[1] The BTs were calculated including selected meteorological parameters (i.e., air pressure, relative humidity, and potential temperature, as well as rainfall and mixing depth) by using the HYSPLIT download package (version 4, Revision 664, Oct 2014). This resulted in an ensemble of 74 496 individual BTs, which were archived as ASCII-files. The further processing of these 'raw' BT files was conducted in IGOR Pro (version 6.3.7, Wavemetrics, Inc., Portland, OR, USA).

Figure 1 shows two exemplary 14-day BT ensembles for selected wet and dry season months. In the context of the present study, we focused on 3-day and 9-day BTs, which are shortened versions of the initially calculated 14-day files. The 3-day BTs cover the air mass movements over the South American continent and, therefore, represent the continental sources that are presumably most relevant for the ATTO site. Here, Fig. 1 illustrates that 3-day BTs in the wet and dry seasons typically span from ATTO to the Brazilian coastal regions and, in some cases, even onto the Atlantic Ocean. The 9-day BTs cover the entire Atlantic region. On these time scales, they provide (at least some) information on potential source regions of long-

---

[1] This time period from Jan 2008 to Jun 2016 was chosen for two main reasons: (i) It includes eight and a half years of BT data, which is a comparatively broad/stable statistical basis to retrieve characteristic BT clusters for the ATTO region. (ii) It includes the AMAZE-08 campaign at the ZF2 site (Feb and Mar 2008, Martin et al. 2010a), which brought together a comprehensive set of instrumentation and represents the starting point of continuous atmospheric observations in the region, since 2008 at the ZF2 site and since 2011 also at the ATTO site.





range transport (LRT) aerosol from the African continent. However, the relative error of the BT model scales with the absolute length of the BTs. The uncertainty of the location of the trajectory center of gravity of the contributing air masses is estimated to be 15-30 % of the travel distance ( https://www.arl.noaa.gov/hysplit/hysplit-frequently-asked-questions-faqs/faq-hg11/, last access 25 Feb

2018). In terms of transport time, Fig. 1 indicates 6-7 days from the African coast to ATTO.

The 14-day BTs were calculated for starting heights of 80 m, 200 m, 1000 m, 2000 m, and 4000 m above ground level (AGL). The overall BT variability of all starting heights have been compared and sum-marized in the Sections 3.1 and 3.2. However, only the BTs started at 1000 m AGL were used systematically throughout this study and particularly for the GIS analysis in section 3.3. As an implication of rather simple

terrain and because of the boundary layer (BL) height variability, this arrival height is representative for sampling heights at ATTO. A comparison of BT ensembles at 200 m vs. 1000 m AGL gave similar results, indicating that the analysis was not very sensitive to varying start heights within the lower 1000 m of the troposphere (for further details see Sect. 2.4 and 3.2). Ensembles of BTs were converted into maps of rela-tive trajectory densities by IGOR Pro routines as illustrated in Figure S1. Slightly modified, this procedure

was also used to get averaged trajectory heights. The BT density maps are called 'air mass residence time maps' throughout this study.

Precipitation was available through GDAS (HYSPLIT model) for every hourly data point of the individ-ual BTs. Based on this data, a cumulative precipitation, $P_{BT}$, for every individual 3-day BT (started at 1000 m AGL) was obtained by integrating the precipitation data along the BT track. The resulting $P_{BT}$ value

represents the amount of rain that the corresponding air parcel experienced during its last 3 days of transport towards ATTO. Accordingly, the $P_{BT}$ time series reflects the variability of cumulative rain and, thus, also the extent of rain-related aerosol scavenging of the arriving air masses. The HYSPLIT precipitation output does not depend on BT transport height (see Moran-Zuloaga et al., 2017).

**2.3   Comparison of HYSPLIT and FLEXPART backward trajectories**

Trajectory models have been constantly improved, while gridded meteorological data became more sophisti-cated (Gebhart et al., 2005). Beside the HYSPLIT model, the FLEXible PARTicle dispersion (FLEXPART) model is frequently used in the atmospheric sciences. The FLEXPART model is a Lagrangian transport and dispersion model to simulate long-range and mesoscale transport, diffusion, dry and wet deposition. A de-

tailed description can be found elsewhere (Stohl et al., 1998; Stohl and Thomson, 1999; Stohl et al., 2005).

The present study is mostly based on HYSPLIT BT data. The HYSPLIT BT ensembles combine the tra-jectories representing the center of gravity ('lines'). We conducted a comparison with FLEXPART for two periods, Mar 2014 and Sep 2014. The FLEXPART model accounts for lateral mixing (or dispersion in for-ward mode), provides distributions of residence times of air masses and, hence, reflects the relative im-

portance of potential source areas. The same starting heights and time frames were applied in both models. The purpose of this comparison was to validate the results of line trajectories (HYSPLIT model) as com-pared to backward modelling accounting for dispersion. The HYSPLIT BT ensembles for the case study pe-riods comprise 9-day BTs, started every 1 h at ATTO at altitudes of 200 and 1000 m AGL during the months




Mar and Sep 2014. Similarly, the FLEXPART BTs were started at ATTO at altitudes of 200 and 1000 m AGL with a backward integration time of 9 days, spanning the months Mar and Sep 2014.

### 2.4 Cluster analysis of backward trajectories

Based on the entire set of 74 496 3-day BTs (1 Jan 2008 - 30 Jun 2016) we conducted a $k$-means cluster analysis (CA, with Euclidian distances) – the most commonly used partitioning clustering approach – which provides a (non-supervised) classification of the trajectories' spatiotemporal variability (MacQueen, 1967; Kassambara, 2017). The 3-day BTs were selected as input data for the clustering to retrieve the BT's spatio-temporal variability over the South American continent and, particularly, over the NE Amazon Basin. The clustering was performed with the statistical software package PASW Statistics (version 18.0.0, IBM, Ar-monk, NY, USA). The 74 496 BTs, each with 72 pairs of latitude and longitude coordinates, represented the input data for the clustering. Further information, such as the altitude of BTs, was not included here. The cluster centers as well as the cluster membership of every BT represented the output data of the CA. The subsequent analysis steps of the CA output were conducted in IGOR Pro. Note that for $k$-means clustering the number of clusters, $k$, has to be pre-defined prior to the analysis. Certain criteria, such as the total within-cluster sum of squares and the silhouette coefficient, have been introduced to select an appropriate $k$ for the CA (e.g., Rousseeuw, 1987; Steinley, 2006). The total within-cluster sum of squares is supposed to be as small as possible and represents a measure for the 'compactness' of the clustering (Kassambara, 2017).

An analysis of the total within-cluster sum of squares as a function of $k$ shows its bend (i.e., the 'elbow') in the range between $k = 5$ and $k = 10$ (Fig. S2). The elbow is typically considered as an indication for an 'appropriate' $k$, however, no 'hard' criterion for the choice of $k$ exists. Here, we tested different numbers of clusters (i.e., $k = 5$, 10, and 15) to identify the most suitable $k$ within the scope of this study. The clusters of the $k = 5$ case (before the elbow is reached) capture the overall spatiotemporal BT variability and result in four major BT directions: northeast (NE), east-northeast (ENE), east (E), and east-southeast (ESE) (Fig. S3a). The clusters of the $k = 10$ case (after the elbow if reached) similarly classify the individual BTs in the main directions of advection (i.e., NE, ENE, E, and ESE), however, start separating the BTs into different wind speed regimes – for example, 2 clusters represent different wind speeds in NE direction (Fig. S3b). Upon further increase of $k$ (i.e., $k = 15$) an even finer wind speed classification is obtained (Fig. S3c). Here, 3 to 4 wind speed regimes are separated for the main directions of air mass advection (i.e., 3 for NE, 4 for ENE, 4 for E, and 3 for ESE). Furthermore, the $k = 15$ case results in a separate cluster towards the south-west (SW).

In this study, we have chosen $k = 15$, which represents a relatively high $k$ with a correspondingly high resolution of spatiotemporal BT details, for the following reasons: The separation into different wind speed regimes appears to provide useful further information beyond the general geographic direction of air mass advection. Specifically, the separated wind speed regimes reflect different residence times of the air masses. Accordingly, the contrast between slow vs. fast BTs may correspond to concentration variations in the ATTO observations (e.g., M. Pöhlker et al., 2017). Furthermore, the contrasting wind speed regimes reflect differences in cumulative precipitation and, therefore, in the extent of rain-related scavenging. Specifically,





the slower BTs such as NE1, ENE1, E1 and ESE1 receive on average significantly more precipitation than the faster ones (Fig. S4). Such proxies for the extent of scavenging are valuable for the analysis of the ATTO aerosol observations (e.g., Moran Zuloaga et al., 2017). Furthermore, for $k \geq 15$, a separate cluster towards the city of Manaus in the southwestern (SW) direction is resolved. The frequency of occurrence of the SW1

cluster helps to estimate how often the Manaus city plume may have impacted the ATTO region, which is an important aspect in the data analysis.

## 2.5  Definition of backward trajectory-based ATTO footprint region

The footprint concept has been introduced for atmospheric measurement sites to quantify the distribution

and extent of biosphere-atmosphere exchange in their surroundings, which contributes to the variability of observed trace compound concentrations (Gloor et al., 2001). Schmid (2002) formally defined the footprint (sometimes also called airshed or effective upwind fetch) of a measurement as "the transfer function between the measured value and a set of forcings on the surface-atmosphere interface" as a general description of various footprint modelling frameworks. Footprints have been mostly used in the context of long-term

trace gas observations (e.g., greenhouse gases) (e.g., Thompson et al., 2009; Winderlich et al., 2010). The size of a footprint largely depends on the height of the measurement, and tall tower sites are known to cover large footprint regions, which makes them particularly valuable for representative regional monitoring (Gloor et al., 2001). The geographic distribution of a footprint is defined by the distribution of relevant sources and the predominant wind directions. The modelling of specific footprints for the ATTO observa-

tions (i.e., for specific compounds and observational conditions) is beyond the scope of this work and will presumably be subject of dedicated future studies.

Here, we use the term footprint in a more simplistic and qualitative sense as the area on the South American continent that is covered by the air mass residence time maps, which are based on multi-year 3-day HYSPLIT BT data. The choice of 3 days for this analysis is justified, for example, by a study by Lammel et

al. (2003), reporting the characteristic formation times of secondary aerosols of about 48-72 h as well as the fact that coarse mode particles "were derived from emissions < 36 h back". With this approach, we aim to map the history of the air masses that were advected towards ATTO on the South American continent. Since primary aerosol and trace gas emissions mostly occur ground-based, only those BTs effectively pick up emissions that reach into the so called footprint layer, a "vertical layer adjacent to the ground in which sur-

face emissions are present and assumed to affect passing air tracer particles" (e.g., Hüser et al., 2017). Accordingly, only those HYSPLIT BTs being transported at lower altitudes reach into the footprint layer and, thus, identify effective source regions. In order to analyze specific relationships between pollution sources and the receptor site (ATTO), assumptions on the vertical depth of the footprint layer and the BT mixing depth have to be taken into account. However, in the present work no filtering by BT transport height has

been conducted for the following reasons: Our aim is to identify potential ATTO-relevant sources in South America more generally by means of multi-year BT data, rather than identifying source-receptor relationships for defined time periods, specific tracer compounds, and defined surface-atmosphere fluxes. Furthermore, the average transport height of the BTs in the main transport tracks is comparatively low as discussed



in more detail in Section 3.1. Accordingly, the large statistics of individual BTs as well as the rather low transport heights underline that the defined BT footprint includes most ATTO-relevant sources that interact at varying degrees with the transported air parcels. Thus, the analysis of the predominant wind directions in comparison to detailed analyses of the land cover in the corresponding upwind areas allows estimating the

relevance of potential source regions for the atmospheric observations at ATTO. In order to discriminate our approach from footprint modelling attempts according to Schmid (2002) as previously outlined, we use the term 'BT footprint' in this study.

The 'BT footprint' of the ATTO site has been defined statistically based on the entire ensemble of 3-day BTs (starting height of 1000 m): The original one hour trajectory points were interpolated to minute-wise

steps and then counted within a 0.1° by 0.1° grid according to Sect. 2.2. Within the distribution of pixel values, which represent air mass residence times, contour lines for the upper 1 %, the upper 5 %, the upper 10 %, the upper 25 % and the median level were calculated. The continental part of the area, which includes the 25 % of highest air mass residence time levels, has been defined at ATTO BT footprint. Analogously, the BT cluster footprints of the 15 clusters from the $k$-means clustering have been defined. For certain as-

pects of the geographic analysis have been weighted by the air mass residence time (see Sect. 2.7). In these case, we refer to the 'weighted BT footprint'. In order to create GIS maps, which includes the most relevant regions, we define a region of interest (ROI) that includes the ATTO-site BT footprint region (ROI$_{foot}$: 62°W, 40° W, 8° S, 6° N; see also Sect. 2.8).

**2.6  Geographic analysis by means of selected GIS data sets**

A detailed analysis of the land cover type and/or change in the ATTO BT footprint region has been conducted. Here, we used a number of different GIS data sets that were processed using the QGIS software package ('Las Palmas' version 2.18.2, QGIS development team). The QGIS software (formally known as Quantum GIS) is freely available under http://www.qgis.org (last access 25 Feb 2018). All GIS data sets

were handled using the coordinate reference system WGS84 (world geodetic system from 1984). A detailed list of all GIS datasets used in this study, together with a short summary of relevant information, web links, and references, can be found in the supplement Section S1.1.

The classification of land cover types as well as the quantification of land cover changes and dynamics is a complex task for various reasons: GIS data sets are often based on different spacecraft instruments and

data acquisition methods (e.g., satellite grid resolution, spectral retrievals and sensitivity, images processing and categorization), which could restrict their comparability. Artefacts such as cloud and/or smoke cover as well as terrain-related shadow effects have to be considered. GIS layers typically differ with respect to their acquisition time frames, which mostly represents states in the past (e.g., GlobCover 2009 representing land cover in the year 2009). Moreover, land cover is subject to dynamic seasonal and phenological changes (e.g.,

in agricultural lands), which is not covered by all GIS data sets (e.g., Ju and Roy, 2008; Jin et al., 2023; Tyukavina et al., 2017). Thus, all corresponding GIS maps are subject to limitations and uncertainties. In this work, mostly well-established GIS products have been used, which are documented and discussed in previous studies. In Section S1.1, we also point out major uncertainties and restrictions with respect to the





comparability of different GIS maps, however, for more detailed information we refer the reader to the referenced studies.

### 2.7 Analysis of backward trajectory footprint region in relation to geographic data

Using scripts written in Python 2.7, an analysis weighted by air mass residence time of several GIS datasets was conducted to assess the relative significance of land cover categories, forest cover and loss, as well as fire events for the ATTO observations. Each dataset was weighted using rasterized BT density maps. The rasterized BT density maps were calculated with cell sizes of 0.09° by 0.09° resolution. For each raster cell, the total length of intersecting trajectories within the cell was calculated. The sum of path lengths was then
set into relation to the highest sum found at the ATTO site, where all trajectories crossed the same cell. To calculate values for the cluster footprints, the GIS raster maps for land cover, forest cover and loss were combined with the air mass residence time map. In the case of the land-cover maps, the relative area of each land cover category within each cell of the BT density maps was multiplied by the corresponding relative density value. In the case of the forest cover map, the relative forest cover within each BT density map cell
was also multiplied by the corresponding relative density value. For the forest loss maps, the same weighting was applied, yielding the relative forest loss in relation to the whole BT footprint region. The relative forest loss was further set into relation to the previously calculated weighted forest cover of the footprint region in the year 2000. For the fire maps, before processing the fire events, the fire map was overlaid onto the land cover raster maps, thus assigning land-cover categories, which were used to characterize the fire events. The
weighting of the number of fires was done analogously to the forest cover data. Therefore, for each BT cluster footprint, we calculated the following output:

(i)      The air mass residence time weighted part of each land cover category and forest cover in all 15 BT cluster footprints.

(ii)     The air mass residence time weighted forest loss for each BT footprint in relation to the forest cover
25           in 2000, per year.

(iii)    The air mass residence time weighted fire counts for each BT footprint, per year and land cover category.

### 2.8 Precipitation, sea surface temperature, and anomalies

The precipitation data used in this study is based on the Precipitation Estimation from Remotely Sensed Information using Artificial Neural Networks for Climate Data Record (PERSIANN-CDR) data product (Ashouri et al., 2015), which has been obtained from Google Earth Engine via https://code.earthengine.google.com/ (last access 04 Mar 2018) (Gorelick et al., 2017). The PERSIANN-CDR data has been analyzed for selected regions: a relatively small area around the ATTO site (59.1°W, 58.6° W, 2.25° S, 1.75°
S, Fig. 4), the continental part of the $ROI_{foot}$, see Sect. 2.5), the Amazon watershed region, and the combined areas of the Brazilian states Acre and Rondônia.





The following anomaly indices for the Pacific and Atlantic sea surface temperatures (SST), which were obtained from https://stateoftheocean.osmc.noaa.gov/sur/ (last access 04 Mar 2015), were used: (i) The Oceanic Niño Index (ONI) within the Pacific Ocean area 170° W- 120° W; 5° S - 5° N, as a measure for the strength and phase of the El Niño-Southern Oscillation (ENSO); (ii) the Tropical Northern Atlantic (TNA)

5    index, within the Atlantic Ocean area 55° W- 5° W; 5° N - 25° N; and (iii) the Tropical Southern Atlantic (TSA) index, within the Atlantic Ocean area 30° W- 10° E; 20° S - 0°. Furthermore, the Atlantic Multidecadal Oscillation (AMO) index was obtained from https://www.esrl.noaa.gov/psd/data/timeseries/AMO/ (last access 04 Mar 2018).

Anomalies in precipitation for aforementioned regions as well as for BT frequency of occurrence have

10   been calculated as the relative differences of the monthly averaged values to a multi-year monthly mean. For PERSIANN-CDR precipitation anomalies, the reference time frame spans from Jan 1983 to Dec 2016. For BT frequency of occurrence anomalies, the reference time frame span from Jan 2008 to Jun 2016.





### 3 Results and discussion

The following sections, 3.1 and 3.2, summarize the results on BT spatiotemporal patterns. Based on that, the subsequent section, 3.3, discusses the land use and anticipated land use change in the ATTO-relevant BT footprint region. All BT and GIS results obtained for the ATTO site also generally apply to the nearby ZF2 site, due to the fact that both sites are located close enough to each other (air-line distance 144 km) to be influenced by similar (large-scale) circulation patterns (Fig. S5). During wet and dry season conditions, the air masses first pass ATTO before reaching the ZF2 region after ~8 h on average (see also Saturno et al., 2017b). This aligned geographic configuration of two broadly equipped atmospheric measurement stations in the Amazon has opened interesting opportunities to study the temporal evolution of atmospheric phenomena.

### 3.1 General backward trajectory circulation towards the ATTO site

The annual north-south oscillation of the intertropical convergence zone (ITCZ) defines the large-scale trade wind circulation patterns in the Atlantic region, which govern the atmospheric seasonality in the central Amazon (compare Martin et al., 2010b; Andreae et al., 2012; Andreae et al., 2015; Moran-Zuloaga et al., 2017b). Here we conducted a multi-year HYSPLIT BT analysis (for five different starting heights: 80, 200, 1000, 2000, and 4000 m) to visualize the large-scale trends in the ITCZ-related air mass advection towards ATTO with respect to BT geographic patterns and transport altitudes (Fig. 2 and S6). For the lower starting heights (i.e., 80, 200, and 1000 m), the overall circulation pattern is predominantly defined by the seasonal ITCZ oscillation and shows two comparatively narrow paths: A northeasterly path during the wet season (Feb-May) and a southeasterly path during the dry season (Aug-Nov). The center of the northeasterly path spans straight from ATTO to the Cape Verde Islands and the north-west African coast (area of Mauretania and Western Sahara). The air mass transport from the Western African coast towards ATTO takes approximately 6-7 days (Fig. S1). The center of the southeasterly trajectory track represents a curved circulation pattern, which is directed eastwards over the mouth of the Amazon River and then curves towards the southeast along the Brazilian coast. For increasing start heights, the separation into distinct northeast and southeast paths becomes more and more smeared out. While the northeast and southeast paths are still somewhat resolved for the 2000 m case, the separation mostly disappears for 4000 m. The observation of the tightest BT bundles close to the ground and their divergence with increasing altitude results from the Hadley cell circulation. Its low-level trade winds feed boundary layer air into the deep-convective ITCZ belt as confluence between northeast and southeast trades (Talbot et al., 1990; Shpund et al., 2011; Dudley et al., 2012; Makowski Giannoni et al., 2016).

Transport within the northeast and southeast BT paths generally occurs at comparatively low altitudes for all start heights (Fig. 2 and S5). This is clearest for the northeasterly circulation during the wet season with its low-level trades. The average transport height of the southeasterly trades tends to be somewhat higher than the northeasterly trades, which can be seen for all starting height cases. For example: At a BT starting height at 200 m (Fig. 2b), the northeast trades are mostly located below 300 m over the Atlantic Ocean, whereas the southeast trades range mostly up to 800 m. At a BT starting height of 1000 m (Fig. 2d),





the northeast trades are mostly below 800 m, whereas the southeast trades are mostly below 1400 m. A side aspect in the context of the low-level air mass circulation is that the HYSPLIT BT results (for starting heights 80 and 200 m) indicate a topography effect over the northeast basin. On their way to ATTO, the air masses tend to follow rather closely the Amazon River valley (Fig. S7), underlining the relative importance

of trace gas and aerosol sources along the river as discussed in more detail in Section 3.3.

The HYSPLIT BTs were found to be a useful tool in the context of this study to analyze temporal and spatial trends of atmospheric circulation patterns. However, it has to be kept in mind that the individual BTs represent a simplified picture by providing center of gravity lines of the transported air parcels, not account- ing for dispersion. To assess the relative importance of dispersion for the large-scale circulation patterns, we

compared the HYSPLIT with corresponding FLEXPART results (accounting for dispersion) for wet and dry season periods. Selected results are shown in Fig. 3 and illustrate that dispersion yield substantial differences between the HYSPLIT and FLEXPART outputs. Generally, The FLEXPART BTs cover a much larger area than the HYSPLIT BT ensembles, which is particularly obvious over the Atlantic region. The comparison further illustrates that deviations get larger the further the BTs reach into the past, as lateral mixing (i.e., dis-

persion in forward mode) becomes more significant. With respect to air mass transport over the northeast Basin (i.e., from ATTO to the Brazilian coast), however, the main BT paths of both model outputs appear to be relatively similar.

### 3.2   Precipitation regimes and seasonality in backward trajectory advection

The $k$-means cluster analysis (CA) partitions the BT's spatiotemporal variability into dominant circulation patterns. It further provides information on the (daily) frequency of occurrence of the clusters and, thus, a time-resolved view on the circulation patterns and corresponding seasonality. Figure 4 shows the 15 BT clusters on a map of the northeast Amazon Basin for a BT starting height of 1000 m AGL (for details re- garding the choice of $k$ and starting height see Sect. 2.2 and 2.4). The clustering partitions the spatiotemporal

variability of the BTs by geographic directions and wind speed regimes. Accordingly, northeasterly (NE), east-northeasterly (ENE), easterly (E), and east-southeasterly (ESE) BTs were separated. In addition, shorter trajectories (low-wind speed regimes) and longer trajectories (high-wind speed regimes) were also separated from each other. Figure 4 shows that the air masses arrived almost exclusively in a rather narrow easterly wind sector (between 45° and 120°) for the 1000 m BT ensemble, which can be subdivided into the four ma-

jor wind directions and subsequently into different wind speed regimes:

- The first group includes three NE trajectory clusters (i.e., NE1, NE2, and NE3), which intersect the coastline in the region of French Guiana and then pass over forest areas towards ATTO. All three tra- jectory clusters follow roughly the same geographic track, however, they represent different wind speed regimes: The longest – and therefore on average fastest – cluster NE3 spans a distance of ~2600 km

and, thus, represents an average air mass velocity of ~870 km d$^{-1}$, while the shortest – and therefore slowest – cluster NE1 spans only ~1300 km, representing an average air mass velocity of ~430 km d$^{-1}$.




- The second group includes four ENE BT clusters (i.e., ENE1, ENE2, ENE3, and ENE4), which intersect the Atlantic coast north of the Amazon River delta (over the Brazilian state of Amapá). These clusters also represent different wind velocities, with cluster ENE1 being, on average, the slowest (~140 km d$^{-1}$) and cluster ENE4 being the fastest (~900 km d$^{-1}$).

- The third group includes four E BT clusters (i.e., E1, E2, E3, and E4), which follow the Amazon River valley. The BT clusters meet the Atlantic Ocean in the area of the Amazon River delta. In this group, cluster E1 represents the slowest (~400 km d$^{-1}$), whereas cluster E4 represents the fastest air mass movement (~930 km d$^{-1}$).

- The fourth group includes four inland BT clusters in ESE and southwesterly (SW) directions. The east-

southeasterly clusters ESE1, ESE2, and ESE3 cross the states of Pará and Maranhão and (on average) do not reach the Atlantic Ocean during the analyzed 3-day period. Cluster SW1 points from ATTO in the direction of the city of Manaus.

For comparison, Fig. S9b shows the clustering results for the 200 m BT ensemble with a partitioning into 15 clusters that is comparable to Fig. 4. Generally, the clusters for the 200 m starting height are shorter, which

can be explained by a higher surface shear and friction as well as topography effects. Overall, the comparison of Fig. 4 and S9b underlines that the observed trends do not vary substantially within the chosen starting height range.

The absolute numbers of individual BTs in the 15 clusters as well as their frequency of occurrence, $f$, are summarized in Table 1. Overall, the group of E clusters was most abundant with $f_E$ = 33.4 %, followed by

ENE with $f_{ENE}$ = 26.8 %, NE with $f_{NE}$ = 19.8 %, and finally ESE with $f_{ESE}$ = 18.5 %. The SW1 cluster BTs, which may transport urban emissions from the city of Manaus to ATTO, were rather rare. Analogous to Fig 2, air mass residence time maps were calculated for the individual 15 BT clusters and can be regarded as cluster-specific BT footprints (shown in Fig. S10). Note that the shapes of these cluster BT footprints are rather diverse. For example, the clusters ENE3, NE2, and NE3 are all characterized by high-wind speed re-

gimes and, thus, by narrow, long, and 'directed' BT footprints. In contrast, the low-wind speed regime clusters SW1, ESE1, and ENE1 are characterized by broader and short BT footprints with air mass advection from different directions. The different BT footprint shapes determine the mix of land cover types that is covered by them, as discussed in detail in Section 3.3.

The clustering further provides time-resolved information on the frequency of occurrence and, thus, sea-

sonality of the individual BT clusters, as shown in Fig. 5 along with the seasonal cycles of selected precipitation products. In terms of rainfall, the Amazon region shows heterogeneous patterns with different precipitation regimes. Figure 5a compares the characteristic seasonality in precipitation rates $P$ within the ATTO proximity, the rather large continental part of the ROI$_{foot}$, the southwestern states Acre and Rondônia, as well as the entire Amazon Basin. The following similarities and differences in $P$ amplitude and phase stand out:

The phase of the average $P$ within the ROI$_{foot}$ and at ATTO resembles the phase within the entire basin (maximum around Mar/Apr vs. minimum around Sep). The $P$ amplitude within the ROI$_{foot}$ ranges above the level for the entire basin and below the level at ATTO. The precipitation regime within the ROI$_{foot}$, which represents the northeast basin, is clearly different from the regime in the southwest basin, here represented



by Acre and Rondônia, showing a $P$ maximum in Feb and a minimum in Jul. The seasonality (top row in Fig. 5), which has been defined in M. Pöhlker et al. (2016), primarily with respect to the aerosol pollution levels, agrees well with the precipitation regime within the $ROI_{foot}$.

For further comparison, we added the cumulative precipitation along the BT tracks, $P_{BT}$, according to the HYSPLIT model. Note that the $P_{BT}$ time series reflects the amount of rain that the air parcels received *en route* and, therefore, provides a measure for the rain-related aerosol scavenging. Moran-Zuloaga et al. (2017a) showed that the $P_{BT}$ is a valuable parameter to explain parts of the aerosol variability at ATTO (e.g., the LRT from Africa). The seasonal cycle in $P_{BT}$ – and thus rain-related scavenging – has its minimum around Aug and Sep, which implies that the abundant biomass burning smoke emitted at that time is effectively distributed over the entire basin. The $P_{BT}$ maximum occurs around Apr and May, which represents (besides the minimum in pollution emission in African and South America at that time) a main reason for the occurrence of very clean episodes within this time window (see M. Pöhlker et al., 2017a).

In relation to the precipitation regimes, the pronounced seasonality in BT frequency of occurrence is summarized for the main wind directions in Fig. 5b as well as for all BT clusters in Fig. 5c. In terms of main wind directions, the following aspects are worth noting: The ATTO site receives rather stable advection from the northern hemisphere ($f_{NE} + f_{ENE}$ up to 90 %) during the transition period from dry to wet season and in the first half of wet season. However, a certain level of $f_E$ ~10 % also prevails during this period. During the second half of the wet season, the BT advection migrates southwards. During the transition period from wet to dry season and the first half of the dry season, BTs from the southern hemisphere predominate ($f_E + f_{ESE}$ up to 100 % in Jul). Note that the E BTs occur throughout the year with varying $f$, whereas ESE BTs, which cover the southeastern Brazilian states, occur only during a comparatively narrow time window (i.e., May to Sep with a maximum in Jul). The (rather rare) SW BTs mostly occur during the late dry season.

Beyond the separation into main wind directions, the representation of $f$ for the individual BT clusters in Fig. 5c resolves further details: As an example, the classification into wind speed regimes by means of the individual BT clusters illustrates that the rather fast NE2 and NE3 BTs reach highest levels from Jan to Apr ($f_{NE2} + f_{NE3}$ up to 50 %), which corresponds with the most frequent arrival of Saharan dust plumes in the ATTO region, due to the fact that the fast NE advection tends to bypass strong precipitation in the ITCZ belt (see details in Moran-Zuloaga et al., 2017b). A further remarkable observation is the relatively fast north-to-south swing of the BTs, spanning over ~3 months (i.e., Apr to Jun), in contrast to the rather gradual south-to-north swing, extending over ~6 months (i.e., Aug to Jan). Overall, the BT seasonal cycle (i.e., northernmost circulation around Feb vs. southernmost circulation around Jul) is phase-shifted relative to the $P$ seasonality. Along these lines, knowledge on the characteristic seasonal patterns in BT advection has proven to be valuable in explaining central aspects of the aerosol variability at ATTO (see Moran-Zuloaga et al., 2017b; Pöhlker et al., 2017a; Saturno et al., 2017b).

Figure 6 shows the inter-annual variation and anomalies in BT advection. It is well known that atmospheric circulation, moisture transport, and precipitation patterns over Amazonia are linked through teleconnections to the variability of the tropical Pacific and Atlantic sea surface temperatures (SST) (e.g., Good et





al., 2008; Fernandes et al., 2015; Tyaquiçã et al., 2017). In particular, the tropical Atlantic meridional gradient has a direct influence on the position of the ITCZ and trade wind patterns towards and over the basin (e.g., Chiang et al., 2002). Specifically, a warming of the tropical north Atlantic (TNA) relative to the tropical south Atlantic (TSA) is associated with a northwards shift of the ITCZ and corresponding weakening of

the northeasterly trades, whereas an anomalously warm TSA relative to the TNA tends to cause a southwards shift of the ITCZ and a weakening of the southeasterly trades (e.g., Cox et al., 2008; Espinoza et al., 2014; Marengo and Espinoza, 2016; Erfanian et al., 2017). Figure 6 confirms these trends by comparing the anomalies in BT frequency (i.e., $f_{NE}$, $f_{ENE}$, $f_{E}$, and $f_{ESE}$) with the anomalies in TNA and TSA SSTs. For several episodes, diametral patterns with an anomalous increase (decrease) in $f_{NE}$ and corresponding decrease

(increase) in $f_{E}$ and/or $f_{ESE}$ can be seen. Four characteristic examples are highlighted in Fig. 6c-e as cases *i*, *ii*, *iii*, and *iv*. For instance, case *i* in 2009 shows a cooling in the TNA and a simultaneous warming in the TSA, being associated with anomalously high $f_{NE}$ and low $f_{E}$ levels. In case *iii*, a cooling of the TNA occurs with an even stronger cooling of the TSA, which is associated by a strong positive anomaly in $f_{E}$. The aforementioned results indicate that the air mass advection and, thus, the atmospheric state at ATTO is teleconnected

to the Atlantic SSTs. However, the statistical basis for the present analysis are only 8 years of BT data and future studies may be needed to explore the role of teleconnections in more detail.

Closely linked to their influence on atmospheric circulation in the basin, teleconnections to Pacific and Atlantic SSTs play a crucial role in the occurrence of droughts and floods in Amazonia (e.g., Fu et al., 2001; Zeng et al., 2008; Fernandes et al., 2015; Marengo and Espinoza, 2016). The Pacific SST variability, which

is represented by the Oceanic Niño Index (ONI), plays a central role in the El Niño-Southern Oscillation (ENSO) and has a pronounced influence on the Amazonian hydrological cycle (e.g., Asner et al., 2000; Ronchail et al., 2002). Periods with a high ONI indicate El Niño conditions and are typically associated with dry or even drought years in the central Amazon (e.g., Lewis et al., 2011; Marengo et al., 2011). A negative ONI indicates La Niña conditions, which are typically associated with rain-rich years. The Atlantic SST further

modulates the hydrological conditions and can intensify ENSO-related anomalies or even cause hydrological extremes itself (i.e., anomalously high TNA can cause droughts, whereas anomalously high TSA can cause floods) (Zeng et al., 2008; Lewis et al., 2011; Marengo and Espinoza, 2016). The ATTO-relevant long-term rainfall anomalies for the ROI$_{foot}$ are shown in Fig. S11 along with the Pacific and Atlantic SST variability. This comparison clearly shows that ENSO has a significant influence on hydrological extremes

within the ROI$_{foot}$. For example, El Niño periods caused severe droughts in the years 1983, 1993, 1997/98, 2003/04, 2009/10, and 2015/16, whereas La Niña episodes caused large positive rainfall anomalies in the years 1988/89, 1996, 1999/2000, and 2011. The drought periods 2009/10 and 2015/6 and an associated increase in fire activity, which also strongly impacted the atmospheric state at the ATTO and ZF2 sites, have been documented in previous studies (e.g., Saturno et al., 2017b; Tyukavina et al., 2017).



### 3.3 Land cover analysis within ATTO site backward trajectory footprint region

The BT analysis defines the areas in the northeastern Amazon Basin that can be regarded as the ATTO site BT footprint region. Its land cover status and anticipated future land cover change is the subject of the analysis in the subsequent sections. As a general overview, Figure 7 shows the geographic extent of the Amazon

Basin in combination with the air mass residence time map. The distribution of air mass residence times shows steeply decreasing values with increasing distance from the ATTO site. The upper 1 % of air mass residence times cover an area of $0.10 \cdot 10^6$ km$^2$ towards the east of ATTO, including regions in the Brazilian states Amazonas and Pará. The upper 5 % (continental area $0.48 \cdot 10^6$ km$^2$) as well as upper 10 % (continental area $0.81 \cdot 10^6$ km$^2$) include in the northeast areas of the state of Amazonas, the northern half of the state of

Pará, as well as the state of Amapá. Accordingly, this region to the north of the Amazon River – including the Amazon River valley itself – appears to be a very important source region for the ATTO observations. The land cover, recent dynamics of land cover change, and any ongoing or planned man-made perturbation (e.g., expanding agriculture, large-scale infrastructure, and mining) in this region are of particular relevance for the (future) ATTO research. The region of the upper 25 % of air mass residence times, which has been

defined as ATTO BT footprint region in the context of this work, covers a continental area of ~$1.46 \cdot 10^6$ km$^2$ and also includes French Guiana and parts of Suriname as well as the eastern Brazilian state Maranhão.

     The forest cover and forest loss map in Fig. 7 illustrates the pronounced northwest to southeast gradient, with the northwest being mostly unperturbed and the southeast being subject of intense, large-scale deforestation and land use change (Davidson et al., 2012). Within this gradient, the forest loss data emphasizes

the geographic extent of the so-called arc of deforestation at the southern and southeastern margins of the Amazon forest, which has been a very active frontier of total forest loss (Morton et al., 2006). The arc of deforestation spans from southern Pará and Maranhão in the southeast over Mato Grosso and Rondônia in the south to Acre in the southwest of the Basin. Here, the majority of forest clearance has been concentrated over the last decades, mainly driven by agricultural expansion (Malhi et al., 2008). The following sections

will zoom into this ROI$_{foot}$, as shown in Figure 6, and analyze the ATTO-relevant land cover properties and trends by means of selected GIS data layers. As general background information, an elevation map, which characterizes the terrain topography in the ROI$_{foot}$, can be found in Fig. S12.

### 3.3.1 Climatic conditions, biomes, ecoregions and the "last of the wild"

This section provides a characterization of the BT footprint region from a climatic and ecological perspective. Figure 8 shows maps of the mean daily temperature and annual precipitation in the ROI$_{foot}$, underlining the (mostly) moist and warm tropical conditions. However, precipitation patterns are relatively heterogeneous: Comparatively dry regions (i.e., annual precipitation <1500 mm) can be found toward the southeast as well as in northern Roraima, whereas the highest annual precipitation occurs along the Guianan coast, over

the Amazon River delta, and towards the southwest of ATTO. In this context, an annual precipitation above the threshold of ~1500 mm is considered as being required for the existence of moist tropical forests, whereas an annual precipitation <1500 mm tends to support savanna-like vegetation types (Malhi et al., 2009). The heterogeneous precipitation patterns are consistent with four different Köppen-Geiger climate





classes being included in the ROI$_{foot}$: the rain-rich regions correspond with tropical rain forest (Af) and tropical monsoon areas (Am), whereas the regions with comparatively low precipitation correspond with tropical savannah (Aw) and hot arid steppe (Bsh) areas (compare Fig. 8b and S13) (Kottek et al., 2006; Peel et al., 2007; Rubel and Kottek, 2010).

Figure 9a presents a geographic biome classification according to Olson et al. (2001), with the following five biomes being included in the ROI$_{foot}$: (i) tropical and subtropical moist broadleaf forests, which occupy most of the area (89.9 % of the continental part of the BT footprint region and 84.4 % of the continental ROI$_{foot}$), (ii) tropical and subtropical grasslands, savannas, and shrublands (8.2 % of footprint and 10.1 % of ROI$_{foot}$), (iii) deserts and xeric shrublands, which occur in the southeast of the ROI$_{foot}$ (0.6 % of footprint and

4.4 % of ROI$_{foot}$), (iv) mangrove forests at the Brazilian and Guianan coasts (1.3 % of the ATTO BT footprint and 0.9 % of the ROI$_{foot}$), and (v) tropical and subtropical dry broadleaf forests in the SE, which are merely of marginal extent in the ROI$_{foot}$. The Olson biome classification and the Köppen-Geiger climate classification generally show consistent geographic features on large scales. The Olson classification represents a hierarchical approach with 867 ecoregions being nested within the larger biome regions. Olson et al.

(2001) defined the ecoregions as "relatively large units of land containing distinct assemblage of natural communities and species, with boundaries that approximate the original extent of natural communities prior to major land-use change". In other words, the Olson ecoregion classification takes into account that "Amazonia is not a single biogeographic entity", but rather "a mosaic of distinct areas of endemism separated by the major rivers, each with their own evolutionary relationships and biotic assemblages" (Da Silva et al.,

2005). These areas of endemism have been defined rather differently, depending on the specific biogeographic distributions of the groups of organisms (e.g., birds, reptiles) that constitute the basis for the resulting cartography (e.g., Cracraft, 1985; Naka, 2011; Oliveira et al., 2017b). However, any classification of ecoregions and/or areas of endemism shows a generalized picture. Irrespective of this uncertainty, our rationale to show the ecoregions by Olson et al. (2011) here is to provide an impression of the biogeographic

diversity within the ROI$_{foot}$.

      Most of these ecoregions within the BT footprint belong to the biome category of tropical and subtropical moist broadleaf forests (Fig. 9b). The ecoregion with the highest overlap with the ATTO footprint is the Uatumã-Trombetas moist forest. Moreover, regions with so-called várzea forest (i.e., the Monte Alegre, Gurupa, and Marajó várzea forests) are located within the footprint, which represent frequently flooded Am-

azonian white-water forests along the rivers Solimões, Madeira, and others (e.g., Wittmann et al., 2004; Junk et al., 2012; Junk et al., 2015; Myster, 2016). As the várzea forests are frequently flooded, they comprise vegetation with special adaptations. Due to the frequent flooding, the soil in these forest regions experiences periodically anoxic conditions, which have been reported to generate significant methane flux to the atmosphere (e.g., Engle and Melack, 2000). Accordingly, the location of frequently flooded areas – such as the

várzea forests in the core region of the footprint – are potentially of high relevance for the methane observations at ATTO (see Andreae et al., 2015). Besides the forests, tropical savanna ecoregions with shrub- and/or grassland vegetation (i.e., the Guianan Savannas and the Cerrado region) are also covered by the footprint



(Dixon et al., 2014). For comparison, Fig. S14 shows the potential natural vegetation in the absence of human alterations, which illustrates that the overall distribution of forests, savannas, and shrublands (i.e., Cerrado region and Guianan Savannas) corresponds with the patterns in climatic conditions (i.e., annual precipitation, see Fig. 8b) rather than human influences. Overall, this general characterization of different/contrasting biomes and ecoregions in the $ROI_{foot}$ may be of value – for instance – for future bioaerosol studies at ATTO as the different ecoregions are presumably associated with varying bioaerosol populations and emission patterns (Després et al., 2012).

An additional GIS layer in Fig. 9a visualizes areas with the lowest anthropogenic influence – so called last-of-the-wild areas – as an approximation of the biosphere in pristine state (Sanderson et al., 2002).[2] The last-of-the-wild map suggests that comparatively large parts of the ATTO footprint region – particularly in the northeast – could be considered as mostly untouched regions (~54 % of the BT footprint). However, it also visualizes that the last-of-the-wild regions are fragmented by rather broad corridors of man-made perturbation along the rivers and highways, as discussed in greater detail in the subsequent sections 3.3.3 and 3.3.6. In this context, a lively discussion on the distribution and extent of human settlements and landscape transformations in a pre-Columbian era has arisen (e.g., Piperno et al., 2015). This debate has refuted the perception that the entirety of the Amazon Basin has been a 'virgin forest' in a pristine state before European arrival in 1491 AD (Bush et al., 2015). Instead archaeological, palaeoecological, and ethnographic research has collected evidence for complex regional settlements, cultural forests, agricultural areas, infrastructure, and, thus, large-scale landscape transformations (e.g., Heckenberger et al., 2003; Clement et al., 2015). These ancient human activities were spatially and temporally heterogeneous with certain areas being highly reshaped and others remaining mostly untouched. Specifically, it has been found that ancient activities were mostly concentrated in riverine settings, such as the várzea floodplain forests, along the major rivers in the central and eastern basin, as well as in savanna and seasonally flooded forest environments (McMichael et al., 2014). Although rapid forest regrowth occurred in abandoned settlements – particularly after the collapse of the native civilization starting in 1491 AD – it is debated whether the ancient human activities have had enduring consequences for the present forest structure and diversity, due to burning, hunting, and the enrichment/depletion of useful/unwanted plants (e.g., Bush et al., 2015; McMichael et al., 2017). The studies by McMichael et al. (2014; 2017) suggest that rather high levels of ancient human activities (e.g., slash and burn agriculture) were concentrated along the major river corridors in the $ROI_{foot}$. This could be of relevance for the contemporary forest structure and biodiversity in the ATTO footprint and has to be considered in future studies.

---

[2] Sanderson et al. (2002) used selected proxies for contemporary human influence, such as population density, land transformation, and infrastructure as a basis to define the last-of-the-wild areas. Note here that the last-of-the-wild map provides only a very general visualization of biosphere regions in untouched state for the following reasons: (i) the map represents the status prior to the year 2000, which means that since then the geographic extent of the last-of-the-wild regions likely has shrunk substantially, (ii) within regions with human influence the severity of human impact/pressure on the biosphere can be rather variable, and (iii) the specification of regions with human influence does clearly not account of the complexity of human activities and their interaction with the biosphere. A more detailed discussion along these lines can be found in Sanderson et al. (2002).





### 3.3.2 Land cover

Figure 10 shows the land cover classification according to the GlobCover 2009 data set within the ROI$_{foot}$ (Arino et al., 2008; Congalton et al., 2014). The location of the arc of deforestation with its extended agricultural areas (represented by the land cover categories 14, 20, and 30) is clearly visible in the SE of the ROI$_{foot}$.

Major roads were added as an additional GIS layer to Fig. 10, underlining their role as starting points for forest fragmentation and clearing (e.g., Fearnside and Graca, 2006 and references therein). This effect can clearly be seen for the Trans-Amazon Highway (BR-230). Furthermore, some of the aforementioned ecoregions can be identified in the land cover categorization. A prominent example are the grass- and shrublands (categories 120, 130, and 140) of the Guianan savanna regions, which represent 'islands' in the extended

moist forests (category 40) (de Carvalho and Mustin, 2017). Moreover, the land cover classes representing frequently or permanently flooded areas such as the aforementioned várzea forests (categories 160, 170, and 180) can be recognized along the major (white water) rivers (e.g., Junk, 2013).

Beyond the qualitative analysis, we quantified the 'land cover mix' within the BT cluster footprints (Fig. S11). For this analysis, the land cover analysis has been weighted by the air mass residence time in the clus-

ters (see Sect. 2.7). Accordingly, regions within the footprint that are located close to the ATTO site were crossed more frequently by BTs and, thus, are weighted more strongly than regions in the periphery of the footprint. As a result, eleven GlobCover 2009 categories account for 99.9 % of the land cover variability within the BT footprint region as summarized in Table 2. The categories 40 (broadleaved evergreen or semi-deciduous forest) and 210 (water bodies, mostly as part of the Atlantic Ocean) are expectedly dominating the

results (i.e., accounting for 87.3 %). Agricultural areas (i.e., categories 14, 20, and 30, accounting for 4.2 %), wetlands (i.e., 160 and 180, accounting for 4.7 %) and shrub- and grasslands (i.e., 110, 130, and 140, accounting for 3.4 %) represent minor fractions of the land cover mix.

For the footprints of the individual BT clusters, the land cover categorization is summarized in Fig. 11. The following trends can be observed: (i) Agricultural lands contribute negligibly to the NE and ENE clus-

ters (sum of categories 14, 20, and 30: ≤1 %), whereas their contribution is noticeable for the E clusters (~3-5 %) and strongest for the ESE clusters (6-20 %). The highest relative fraction was observed for cluster ESE3 (~20 %), which reaches directly into some 'hot spots' of intense agriculture in southern Pará and Maranhão (compare Sect. 3.3.3). (ii) The categories 160 and 180 are associated with regularly flooded areas. These categories are rather rare in the NE cluster (sum of 160 and 180: ≤2 %), show variable contributions

to the ENE clusters (2-5 %), and are comparatively abundant in the E and ESE clusters (4-11 %). The comparison of Fig. 10 and 11 shows that particularly those BT clusters that have a rather high residence time over the Amazon River (i.e., ENE1, E1-4, SW1) show the strongest 'floodplain contribution', with the aforementioned potential relevance for methane observations. (iii) The categories 110, 130, and 140 represent areas with grass, shrub, and/or moss/lichen coverage. They are comparatively rare in the NE and ENE clusters

(<3 %), whereas larger contributions are found for the E and ESE clusters (2-12 %).

Based on the land cover characterization, we further conducted an analysis of the forest phenology within the ROI$_{foot}$ by means of normalized difference vegetation index (NDVI) data. The NDVI targets specific spectral properties of the plants' chlorophyll absorption and, thus, represents a measure for vegetation





'greenness' or net primary productivity (e.g., Pettorelli et al., 2011; Wu et al., 2016). In numerous previous studies, the response of Amazonian phenological cycles to climatic, environmental, and biological factors has been investigated, which helps to assess the Amazon's vulnerability towards climate change (e.g., Atkinson et al., 2001; Schucknecht et al., 2013; Silva et al., 2013; Hilker et al., 2014). It has been shown that

vegetation phenology responds primarily – and typically with time-lags – to rainfall and radiation as well as, with somewhat less relevance, to temperature variations (Zhao et al., 2017). Generally, phenological cycles in the Amazon are highly complex (e.g., Bradley et al., 2011; Zhao et al., 2017). Accordingly, the phenological aspects discussed here focus only on the overall trends for the $ROI_{foot}$-relevant land cover categories in Table 2.

The results in Fig. 12 show a pronounced NDVI seasonality for all relevant land cover classes. Specifically, the seasonal cycles follow two contrasting patterns in relation to rainfall and cloud fraction with the latter one being an indirect proxy for incoming solar radiation (e.g., Hilker et al., 2014). The NDVI results for the evergreen moist forest categories (i.e., 40 and 160) show a minimum around February and a maximum around July. Thus, the seasonality is generally in-phase with solar radiation and suggests sunlight-en-

hanced growth upon decreasing cloud cover in the dry season, in combination with a time lag of greening after the rain-rich wet season (Hilker et al., 2014 and references therein). Apparently, the increasing drought stress in the dry season is buffered by the deep-rooting trees in the moist soils and, therefore, does not (significantly) affect the NDVI (Nepstad et al., 2008). In contrast, land cover categories representing agricultural lands (i.e., 14, 20, and 30), shrub- and grasslands (i.e., 110 and 130) as well as deciduous forests (i.e., 50)

show a NDVI maximum in May (~2 months after the rain maximum) and a minimum around September to October. Thus, the seasonality is 'in-phase' with precipitation (although with a certain time lag) and suggests a rainfall-constrained growth (Atkinson et al., 2011). Apparently, the dry season drought stress affects the greenness of these 'low/sparse vegetation' categories severely. The land cover categories 140 and 180 show an NDVI seasonality, which resembles an intermediate state of both afore mentioned cases, generally show-

ing an in-phase relationship with radiation, however, with a secondary minimum in Sep to Oct upon dry season drought stress. The presented seasonal patterns within the $ROI_{foot}$ are generally consistent with previous studies, which have analyzed the heterogeneous distributions of phenoregions in the Amazon region as well as their diverse seasonality in growing seasons and the associated NDVI patterns (Bradley et al., 2011; Silva et al., 2013).

### 3.3.3  Deforestation and agro-industrial expansion

Amazonian deforestation and further forest degradation activities (e.g., ecosystem fragmentation, fires, selective logging, illegal mining, overhunting, etc.) as a function of biophysical, climatic, socioeconomic, and cultural factors have been addressed by a large number of previous studies (e.g., Nepstad et al., 1999;

Laurance et al., 2002; Asner et al., 2005; Malhi et al., 2008; Godar et al., 2012a; Cisneros et al., 2015). Here, we discuss the ATTO-relevant deforestation trends and drivers.

Figure 13a provides an overview of the deforestation patterns within the $ROI_{foot}$ by showing forest cover and forest loss maps according to Hansen et al. (2013) in comparison with the biome classification from Fig.



9a. The geographic extent of the tropical and subtropical moist broadleaf forest biome provides an estimate of the area that was originally covered by primary moist forests within the $ROI_{foot}$ (compare also ter Steege et al., 2015). For the Amazon Basin, the total cumulative deforested area by 2017 equals about 800 000 km², which is ~20 % of the original Brazilian rain forest area (Fearnside, 2005).[3] Further note that the satellite-based deforestation monitoring used here does not detect selective logging and surface fires in standing forests, which have a significant forest degrading effect (Nepstad et al., 1999; Cochrane and Laurance, 2008). Souza et al. (2013) estimated that the area of degraded forest is equivalent to 30 % of the area deforested at the same time. The map in Fig. 13a underlines the northwest to southeast gradient indicated by an increasing extent of forest fragmentation or complete forest loss towards the southeast. With respect to the ATTO site BT clusters, the following three regions with different deforestation states along this gradient can be defined:

- The ESE clusters cover the areas with the strongest forest fragmentation and perturbation within the $ROI_{foot}$. This region – particularly in southern Pará and Maranhão – has been a very active frontier within the arc of deforestation (e.g., Soares-Filho et al., 2013). In fact, Pará is one of the two Brazilian states (the second one is Mato Grosso) with the largest tree cover loss in recent years (Tyukavina et al., 2017). The clusters overpass formerly forested areas that have been cleared and converted (almost) completely into agricultural lands (i.e., extended regions in Southern Pará), on one hand, and 'active hotspots' with currently progressing deforestation (also in Southern Pará), on the other hand.

- The Amazon River and its shores are covered by the E clusters. Here, some deforestation hotspots (i.e., at the northern shore, about half way between the ATTO region and the river delta) have emerged and could potentially develop into large-scale forest destruction in the future, depending on the overall socioeconomic trends and conservation efforts (Fearnside, 2007). Accordingly, these clusters represent a semi-perturbed area of the basin and, thus, an 'intermediate state in the Amazon's transition' (Davidson et al., 2012).

- The ENE and NE clusters cover areas, where deforestation activities are comparatively low (i.e., northern Pará, Amapá, and French Guiana). Therefore, the ENE and NE clusters still represent a mostly unperturbed state of the forest.

The so called 'fish bones' along the major highways represent a typical deforestation pattern, which consist of perpendicular smaller and mostly illegal access roads, penetrating (deeply) into the surrounding forest (Laurance et al., 2009). In Fig. 13, these patterns can be recognized, for example along the highways BR-163 (the so-called "soybean corridor" connecting the international port in Santarém with the soybean production in the Southern states) and BR-230 (Trans-Amazon highway) (Soares et al., 2004). The deforestation is typically associated with a strong fragmentation of the remaining forest areas. The fragmentation creates so-called edge effects, which perturb the humid, dark, and stable microclimate in the forest's canopy and understory with impacts on forest structure, tree mortality, and biodiversity (e.g., Wirth et al., 2007; Broadbent et al., 2008; Dohm et al., 2011; Laurance et al., 2011).

---

[3] For this calculation, the following numbers have been used: Fearnside (2005) reported a total deforested area of ~650 000 km² by 2003. From 2004 to 2017, further ~150 000 km² have been deforested according to http://www.obt.inpe.br/OBT/assuntos/programas/amazonia/prodes (last access 23 Feb 2018). The original rain forest area in Brazil was estimated as ~4 000 000 km² (Fearnside, 2005).



The main socioeconomic drivers of Amazonian deforestation can be grouped into two categories: The first category comprises subsistence and family agriculture, including some extractive activities, such as logging and hunting. These individual actions typically occur on rather small scales, however, are practiced by a rather large number of smallholders and colonists, which ultimately sums up to a substantial level of deforestation pressure (Godar et al., 2012a). The second category comprises deforestation on larger scales, which has been mostly driven by international economic interests, market demands, and government policies/subsidies (e.g., Soares et al., 2014). It is conducted by a comparatively small number of largeholders[4], including industrial logging to create soy plantations, cattle ranches, large-scale timber production, and mining. In the Amazon, cattle ranching has clearly been the dominating factor, accounting for about 70 % of deforestation (Souza, 2006; Malhi et al., 2008; Barona et al., 2010). Note that the relative contributions of small- vs. largeholders as a function of socioeconomic conditions is still a subject of active and controversial debate (e.g., Ludewigs et al., 2009; Pacheco, 2012; Pereira et al., 2016).

Figures 13b and c zoom into two selected areas within the ROI$_{foot}$ that illustrate the typical deforestation trends, on the one hand, and represents regions of particular relevance for the ATTO site, on the other. Figure 13b covers part of the region along the northern shore of the Amazon River that represents a recent deforestation hotspot (i.e., in the municipals Oriximiná, Óbidos, Curuá, Alenquer, and Monte Alegre) in the center of the ATTO footprint (within 1 % contour of highest residence times, Fig. 7). Due to a northwards migration of investors and colonists, the extent of cattle ranching, soy farming, and related deforestation has increased over the last years in this region (Fearnside, 2007; Bowman et al., 2012; Cisneros et al., 2015). Figure 13b highlights one of the expanding 'fish bone branches' in this area. Besides the agriculture-related forest loss, mining activities represent an additional perturbation of the forest. As one example, Fig. 13b shows the extent and growth of a large-scale bauxite mine in the municipality of Oriximiná, which is discussed in further detail in section 3.3.6.

Figure 13c zooms into a region in central Pará at the Trans-Amazon highway (BR-230), which has been a corridor of continuous and active deforestation of the last decades. This region has been selected for three reasons: (i) the characteristic fish bone deforestation patterns are particularly pronounced here, (ii) the region represents a deforestation hotspot that is comparatively close to ATTO (within 10 % contour of highest residence times, Fig. 7) and, thus, presumably has noticeable influence on the atmospheric observations, and (iii) its deforestation dynamics have been well documented previously by Godar et al. (2012a; 2012b). Specifically, the municipalities of Mediciurília and Novo Brazil in Fig. 13c are characterized by contrasting socioeconomic deforestation trends: in Mediciurília, smallholder (family) agriculture has been predominant, which corresponds with a comparatively low degree of deforestation, rather short 'fine bones', small deforested patches that are rather close to the BR-230, as well as declining deforestation rates (Godar et al., 2012a). In contrast, in Novo Brazil, largeholder agriculture with extensive cattle ranching and industrialized soy farming has had a much more pronounced influence, which corresponds to a stronger degree of frag-

---

[4] The term largeholder is frequently used as the opposite to smallholder in the literature on deforestation (e.g., Godar et al., 2012a, 2012b; Pacheco, 2012). We have adopted this notation here.




mentation, longer 'fish bones', many large rectangular patches (mostly for cattle ranching with several hundreds of ha) that are located rather far from the BR-230, and increasing deforestation rates (Godar et al., 2012a; Cisneros et al., 2015).

Based on the annual forest loss data in Fig. 13, we conducted a quantitative analysis of the deforestation levels and dynamics within the weighted footprints of the individual BT clusters (Fig. 14 and Table S2). Figure 14 shows that the overall annual forest loss levels span from ~0.01 to ~1 % $a^{-1}$. For comparison, the basin-wide forest loss levels range between ~0.1 % $a^{-1}$ (for 2012) and ~0.5 % $a^{-1}$ (for 2004).[5] The highest deforestation rates in the basin reach up to about 3 % $a^{-1}$ and have been observed in hotspots such as the BR-163 corridor (Müller et al., 2016). Furthermore, the cluster-resolved forest loss data in Fig. 14 reflects the

northwest-southeast gradient with highest forest loss levels in the south of the $ROI_{foot}$ and lowest levels in the north (i.e., NE < ENE < E < ESE). In terms of forest loss temporal trends within the period 2001 to 2014, we found a (slightly) decreasing trend for the ESE clusters (on average -0.0013 % $a^{-1}$), which can probably be explained with political efforts to reduce forest destruction in the arc of deforestation (i.e., blacklisting) (e.g., Cisneros et al., 2015). In the most recent years, the basin-wide deforestation rates have

increased again after a minimum in 2012 in response to a relaxation of environmental laws (Tollefson, 2016). In contrast to the ESE trend, the forest loss rates in the NE, ENE, and E clusters tend to increase (0.0013 % $a^{-1}$ for NE and ENE vs. 0.0032 % $a^{-1}$ for E). The increasing forest loss trends in the NE, ENE, and particularly in the E clusters reflect the northwards migration of colonists and investors with the associated increasing pressure on the forest ecosystem in the affected areas. In addition to the overall trends, we also

observed a rather strong year-to-year variability of the forest loss rates, which is related to regionally heterogeneous biomass burning activities with strong interannual variations as discussed in the following section.

### 3.3.4 Fires

This section presents the spatiotemporal patterns of fire occurrence within the $ROI_{foot}$. Fires are of significant

importance for the ATTO observations since they are highly destructive or transformative for the rain forest ecosystem (Cochrane and Laurance, 2008) and represent the primary pollution source with fundamental consequences for atmospheric composition and processes (e.g., Andreae et al., 2004; Lin et al., 2006; Aragao et al., 2008; Artaxo et al., 2013; Saturno et al., 2017b). In an unperturbed state, the humid, dense, and high-canopy Amazonian forests are mostly nonflammable, which has made wild fires rare events (Cochrane,

2003; Nepstad et al., 2008). Accordingly, the Amazonian vegetation is evolutionary ill-adapted to the occurrence of (even low-intensity surface) fires and typically reacts highly vulnerable to it (e.g., Uhl and Kauffman, 1990; Cochrane and Schulze, 1999). However, fires have become ubiquitous in the basin over the last decades due to man-made activities: Numerous agriculture-related fires are ignited on purpose. Moreover, the progressing forest fragmentation and degradation perturbs the moist forest climate and, thus, tends

to increase the forest's flammability, which makes the uncontrolled escape of fires more likely (Morton et

---

[5] The calculations are based on the annually deforested area of ~27 800 $km^2$ for 2004 and 4 600 $km^2$ for 2012 according to http://www.obt.inpe.br/OBT/assuntos/programas/amazonia/prodes (last access 23 Feb 2018) and a total rain forest area of ~5.52·$10^6$ $km^2$ according to Goulding et al. (2003).



al., 2011; Alencar et al., 2015). During El Niño years and the associated drought conditions, the flammability can be even further enhanced (e.g., Nepstad et al., 2001; Fernandes et al., 2011; Cano-Crespo et al., 2015). In this context, further changes of the Amazon fire regime may play critical roles in positive feedback mechanisms and, thus, foster and accelerate large-scale rain forest dieback, as projected by modelling stud-

ies (e.g., Nepstad et al., 2008; Cochrane and Barber, 2009).

Figure 15 shows a fire map within the $ROI_{foot}$, representing the average fire-related carbon emission flux from 2000 to 2016, according to Kaiser et al. (2012). Generally, a rather heterogeneous distribution of fires can be observed. On large scales, a northwest to southeast gradient, with significantly higher fire abundance towards the southeast, characterizes the $ROI_{foot}$. This gradient is closely linked to the agro-industrial expan-

sion and associated deforestation, since fires are the primary tool for land clearing, which typically occurs as slash-and-burn conversion of forest into fields and/or pasture and, furthermore, produces ash as precious fertilizer for the acid, infertile soils (e.g., Nepstad et al., 2001; Cochrane and Laurance, 2008). Moreover, fires represent an efficient land management tool for maintenance and improvement of fields and pastures. In the $ROI_{foot}$, the highest fire activities have been observed in the state of Maranhão, which has a dense coverage

of cropland and pasture. Furthermore, several fire hotspots are located in Southern Pará, along the major highways and rivers, representing the expanding agricultural frontiers. Many of these regions are covered by the footprint of the ESE clusters and are, therefore, within the scope of the ATTO observations. In contrast, the ATTO-relevant NE BTs cover extended unperturbed areas that have maintained their 'fire-immune' moist climate and, thus, showed no fire activity.

In addition to the agriculture-related fire patterns, Fig. 15 further shows substantial fire densities in the savanna ecoregions (i.e., high activity in the Cerrado and modest activity in the Guianan savanna regions), which are characterized by lower precipitation, a different vegetation type, and a tendency to higher flammability. In contrast to the fires in moist forested regions, which are the result of man-made ignition or the consequence of man-made forest degradation and increased flammability, the low precipitation levels in the sa-

vanna regions makes them more prone to the occurrence of fires. Accordingly, the savanna vegetation is more adapted to an (infrequent) occurrence of natural fires as well as more frequent use of fire by indigenous people since pre-Columbian times (de Carvalho and Mustin, 2017). Parts of both ecoregions, the Cerrado and Guianan savannas, with their characteristic fire regimes are ATTO-relevant as they are located within the BT footprint. On a year-to-year basis, our analysis showed comparable geospatial patterns in fire occur-

rence within the $ROI_{foot}$, however, with an annual variability in fire intensities on regional scales (see fire anomaly maps in Fig. S18). For instance, the warm and dry years 2005, 2007, and 2010 showed a basin-wide increase in fire activity (Alencar et al., 2015). The El Niño years 2009 and 2015 showed intense fires along the ATTO-relevant part of the Amazon River valley, corresponding directly with strongly increasing aerosol concentrations at ZF2 and ATTO (Saturno et al., 2017b).

Beyond the fire map in Fig. 15, we conducted a quantitative classification of the detected fires by two metrics: the weighted BT cluster footprints and the land cover type in which they were detected. Figure 16 shows the resulting relative fractions of fires per land cover type and grouped into the four main BT direc-



tions, NE, ENE, E and ESE. Note that this analysis has been weighted by BT residence time and, thus, provides an estimate for the ATTO-relevant 'mix of fires' (e.g., forest fires vs. savanna fires vs. agricultural management fires). This is relevant for the ATTO observations since the fuel types (e.g., forest vs. agricultural waste) and corresponding combustions modes (i.e., flaming vs. smoldering) typically emit gas phases

and aerosol particles of different composition and properties (e.g., Janhäll et al., 2010). For example, break-out understory fires (escaping from ignited land clearing fires) are typically associated with less efficient smoldering combustion, whereas pasture burning and high intensity deforestation fires (after clear-cutting and drying of the vegetation) tend to be predominantly flaming combustion (e.g., Tang and Arellano, 2017). Figure 16 shows different 'fire mixes' for the NE, ENE, E, and ESE clusters. The dominant contribution in

all cases are fires in rain forest areas (land cover category 40), which accounts for 60-63 % in NE, ENE, and E directions as well as for 54 % in ESE direction. Fires in shrub and grassland categories (i.e., 110, 130, 140) account for comparable fraction (10-16 %) for all directions. Fires associated with agricultural categories (i.e., 14, 20, 30) show a pronounced gradient from NE (11 %) over ENE (13 %) and E (17 %) towards ESE (28 %). This is consistent with the properties of absorbing aerosols measured at ATTO (e.g., the rela-

tive fractions of black vs. brown carbon), which have been related to the air mass origin by means of BT directions (Saturno et al., 2017b). Finally, we analyzed the seasonal cycle in fire occurrence for the entire BT footprint, resolved by land cover categories and found a rather uniform seasonality with its minimum around April and its maximum around September (see Fig. S19). Also these trends, particularly the onset of the biomass burning season, are consistent with the observed atmospheric composition and variability at ATTO

(e.g., Andreae et al., 2015; M. L. Pöhlker et al., 2016; Saturno et al., 2017b).

### 3.3.5 Infrastructure, cities, traffic and mining

Biomass burning represents the predominant forest perturbation and source of atmospheric pollution in the Amazon Basin. However, several further categories of man-made activities and infrastructure also impact

the biosphere-atmosphere exchange to a significant extent. Accordingly, this section addresses the following infrastructure classes: (i) Population density and urban centers with their related emissions, (ii) thermoelectric power plants as major fossil fuel burning sources, (iii) major dams and reservoirs with their significant environmental impacts, (iv) major highways as key drivers for forest fragmentation and degradation, and (v) (cargo) ship traffic as further pollution source.

The population density map in Fig. 17 shows that most settlements and cities are located along the Atlantic coast in the southeast of the $ROI_{foot}$ (see also Andreae et al., 2015). The associated emissions from these densely populated regions comprise a complex mixture of primary and secondary pollutants from traffic, heating, cooking, industry, waste, landfills etc. (e.g., Gentner et al., 2017). Specifically, fossil fuel fired power plants represent one of the dominant sources of pollution aerosols and gases in this context (Kuhn et

al., 2010). Figure 17 shows that thermoelectric power plants within the $ROI_{foot}$ are collocated with the densely populated areas: A comparatively large number of (relatively small) power plants can be found along the Amazon River as well as close to the urban centers at the Atlantic coast. Two major natural gas





fired power plants are located close to Belém in the Amazon River delta and, thus, within the ATTO foot-print. A yet unquantified fraction of the urban pollution mixture (i.e., from power plants, traffic, industry etc.) is likely being transported towards ATTO with the E and ESE BTs. The significance of these emissions in comparison to biomass burning for the ATTO site observations is subject of currently ongoing analysis

(e.g., Carbone et al., 2017; Saturno et al., 2017c).

Dams and reservoirs worldwide have a substantial impact on rivers and their ecology for a variety of reasons (Lehner et al., 2011). An increase in methane (also carbon dioxide) emissions from reservoirs due to anoxic microbial decomposition of flooded biomass has been one particular impact on the biosphere-atmosphere exchange – presumably also with direct relevance for the ATTO site observations (e.g., Abril et al.,

2005; Kemenes et al., 2007; Fearnside and Pueyo, 2012). Figure 17 shows that several major dams are located within the ROI$_{foot}$. Examples are the Curuá-Unã Dam on the Curuá-Unã River in central Pará (flooded area ~120 km$^2$; ~4 km$^2$ MW$^{-1}$), the Tucuruí Dam on the Tocantins River in the SE of Pará (flooded area: ~3010 km$^2$; ~0.4 km$^2$ MW$^{-1}$), the Coaracy Nunes Dam on the Araguari River in Apamá (flooded area: ~30 km$^2$; ~0.3 km$^2$ MW$^{-1}$), the Petit-Saut Dam on the Sinnamary River in French Guiana (flooded area:

~350 km$^2$; ~3 km$^2$ MW$^{-1}$), as well as the Boa Esperança Dam on the Parnaíba River in Piauí (flooded area: ~88 km$^2$) (Lehner et al., 2011). The Balbina Dam on the Uatumã River in northeast Amazonas (flooded area: ~4450 km$^2$; ~18 km$^2$ MW$^{-1}$) is located ~60 km northwest of ATTO and, thus represents the closest dam and moreover one of the largest in the basin. It is located downwind according to the main BT directions, how-ever it may be a relevant source during prevailing southwestern air mass advection. Parts of the Balbina

Dam are located within the 25 % contour of highest residence times. Furthermore, localized air flow along the Uatumã River, which receives water from the turbines that are fed from the hypolimnion, might also have the potential to bring methane-rich air to the ATTO site. A detailed analysis of the potential influence of the Balbina Dam on the ATTO observations requires a dedicated future study. Further smaller dams are located in the SE of the ROI$_{foot}$, which are not explicitly mentioned here. A number of additional (major)

dams in the basin are planned or already under construction. The most prominent and controversial example is the Belo Monte Dam on the Xingu River in central Pará (planned capacity: ~11000 MW; estimated lake area: ~670 km$^2$) (also marked in Fig. 17). The construction of the Belo Monte Dam is associated with severe environmental destruction, such as flooding of large areas and destruction of the Xingu Rivers hydrological regimes (Fearnside, 2017). Furthermore, the dam is located directly within the SE segment of the BT foot-

print (i.e., included by the 10 % contour of highest residence times) and, thus, may bring a large-scale an-thropogenic impact for future ATTO observations. Specifically, the flooding of large forested areas upon filling of the dam and the associated anoxic degradation of the biomass will release large amounts of trace gases (i.e., methane, ammonia, etc.), which will likely impact the ATTO measurements.

In the previous sections, we have noted that highways in the Amazon Basin "have a keystone role in de-

forestation" and "stimulate the influx of population and investment" (Fearnside and Graca, 2006). Figure 17 displays the currently existing road network in the ROI$_{foot}$ as well as some major highways that have been proposed. In addition, an unofficial network of smaller roads has been developing in parallel, which is only partly shown in Fig. 17 (Barber et al., 2014). Figure 13 emphasizes how the existing major highways, such



as the BR-230, BR-163, and BR-319, have initiated and fostered extensive deforestation. A particularly severe threat for the Amazon biome is the construction of "all-weather highways into the core of the Basin", which is currently being realized by paving the highways BR-319 (connecting Manaus with the South) and BR-163 (connecting Santarém with the South) (Soares-Filho et al., 2006; Barni et al., 2015). Also within the

yet untouched northeastern segment of the ATTO footprint, major road constructions are being planned. Specifically, plans exist to build the ~1 100 km long BR-210, connecting Boa Vista in Roraima and Macapá in Amapá (de Carvalho and Mustin, 2017). Currently, two initial segments of the BR-210 exist (i.e., ~130 km in Roraima and ~250 km in Amapá). A further project, which is currently being debated, is the construction of the so called Arco Norte (the north arc)[6] – a connection between Boa Vista and Santarém via

an eastern segment of the BR-210 and a southern segment of the BR-163 (see Fig. 16). A potential construction of the BR-210 and BR-163 would cut through the extended and mostly untouched forest areas in Northern Pará and Amapá, which currently constitute the 'pristine' NE and ENE clusters of the ATTO footprint. The shortest distance between ATTO and BR-163 would be about 250 km and, thus, parts of the highway would reach into the 1 % contour of highest residence times (Fig. 7). In the light of the destructive potential

of roads and the proximity to the ATTO site, the outcome of these infrastructure projects is of fundamental importance for future ATTO research. Note that the existing section of the BR-210 in Roraima has initiated the growth of a major deforestation hotspot (see Fig. 13 and 15) and, thus, provides an outlook on the potential destruction associated with the construction plans (Barni et al., 2015).

Figure 18 displays a map, which zooms out from ROI$_{foot}$ and combines the ATTO BT footprint with ship

tracks on the Atlantic Ocean as well as on the South American inland waters. Worldwide, ship traffic is known as a strong source of atmospheric pollution due to the fact that heavy, sulfur-rich oils serve as fuels. Accordingly, ship emissions comprise comparatively high loads of sulfate aerosols in addition to large quantities of further pollutants (e.g., Aulinger et al., 2016). Figure 18 illustrates the major ship routes along the eastern South American coast, which are covered by the footprint. In addition to offshore shipping, a signifi-

cant amount of vessel traffic also occurs on the major rivers and inland waters in the Amazon Basin. In particular, the Amazon River itself represents the predominant route for cargo transportation with major harbors along its shores (e.g., Manaus, Itacoatiara, Santarém, Belém, etc.). Inland water shipping represents a potentially important source as it occurs comparatively close to the ATTO site and uses dirty fuels with little emission controls.

Figure 19 presents a map highlighting the ATTO BT footprint in combination with a GIS layer on mining activities in the basin. Generally, mining – particularly large scale pit mines – has caused strong perturbations of the forest ecosystem in affected areas (e.g., Potapov et al., 2017). Moreover, large mines are potential sources of industrial air pollution as well as soil dust suspension (i.e., coarse mode aerosol particles) (e.g., Huertas et al., 2012). Both of these factors make mining relevant for the ATTO research. Figure 19

emphasizes that comparatively large bauxite mines already exist within the ATTO BT footprint. The largest of those mines, which have been shown in Fig. 13 already, is located within the 1 % contour of highest air

---

[6] See, for example, Ministros assinam contrato que estuda ligação de RR ao Pará pela BR-210. *GLOBO G1* (27 June 2016); http://go.nature.com/2lYa3sa (last access 25 Mar 2018).



mass residence times (Fig. 7) and is, thus, a potentially relevant source for the ATTO observations. It further caused substantial forest loss rates over the last years (~400 ha y$^{-1}$)[7]. Moreover, mining-related exploration and registration activities are being conducted in the entire ROI$_{foot}$, which reflects the strong economic interests in resource extraction in this area. To highlight another relevant example: In August 2017, the Brazilian

federal government has abolished the protection status in major parts of the national reserve of copper and associates (RENCA), located in Northern Pará and Amapá, to open these areas for the extraction of gold, copper and further minerals.[8] The RENCA area is located within the 5 % contour of highest air mass residence times (Fig. 7) and, thus, of major relevance for the ATTO observations (see Fig. 19). The RENCA area is further located in the center of the still untouched forests. In the meantime, this initiative has been

stopped judicially. However, this case emphasizes that political and judicial decisions can change the protection and land use status in the ATTO footprint rapidly and profoundly.

### 3.3.6 Protected areas

The conservation efforts to protect the Amazonian forests are manifold. Often, protection is initiated as re-

sponse to deforestation frontier expansion (Nepstad et al., 2006). Worldwide, a large variety of different types of conserved areas (e.g., in terms of their legal, control, and habitation status) exists. According to the world database on protected areas (WDPA) classification, about 14 types are relevant in the ROI$_{foot}$ (Fig. 20a). About 41 % of the actual footprint region are protected. Basin-wide, about half of the forest area has a protection status, whereas only 10-20 % are considered as strictly protected (Barber et al., 2014; ter Steege

et al., 2015). The establishment of protected and controlled areas, such as parks, reserves, and indigenous lands, can be an effective tool to reduce or even inhibit deforestation and fire occurrence (e.g., Nepstad et al., 2006). However, also contrasting examples have been reported, where protection has been less effective (e.g., Nolte et al., 2013).

The ATTO site itself is located in a sustainable development reserve (i.e., Reserva de Desenvolvimento

Sustentável do Uatumã). Such sustainable development reserves allow a certain level of resource use and extraction, in contrast to strictly protected areas. Figure 20a further shows that the ATTO site footprint is to various degrees covered by protected areas. Specifically, almost the entire northern half of the BT footprint with the NE and ENE clusters (i.e., in Northern Pará and Amapá as well as in French Guiana) consists of protected lands. In contrast, the southern half, which overlaps with the arc of deforestation and the ESE clus-

ters (i.e., ESE2 and ESE3), comprises only few and rather small conservation areas. The part of the Amazon River valley that is covered by the E clusters contains few protected areas, and, therefore, plays an intermediate role in this overall conservation picture. In Fig. 20a, a comparison of the protected areas and the patterns in fire occurrence highlights examples for both, successful and less successful conservation efforts: For example, most of the indigenous areas have been very efficient in preventing deforestation and, thus, do not

overlap with the major fire hotspots (see also Ricketts et al., 2010). Note also the sharp edges between fire

---

[7] Annual forest loss of ~400 ha y$^{-1}$ was obtained from http://www.globalforestwatch.org/ (last access 18 Nov 2017) and has been documented from 2001 to 2016. The total mine-related deforested area corresponds to about 6200 ha.
[8] Some information on the initiative can be found here: https://wwf.panda.org/?309330/WWF-statement-on-Brazil-governments-decision-to-open-up-a-national-reserve-in-the-Amazon-for-mining (last access 14 Nov 2017).



hotspots and indigenous areas in certain cases. In contrast, the formally protected areas along the highway BR-163 have not prevented the continuous forest fragmentation in this area. The future effectiveness of the protected areas in the ROI$_{foot}$ is hard to predict, as conservation efforts depend on (dynamic) contextual factors, such as economic development, accessibility, regional climate, and political willingness/capability to

enforce conservation regulations (e.g., Schwartzman et al., 2000; Soares-Filho et al., 2006; Nolte et al., 2013; Soares-Filho et al., 2013). The RENCA case shows that a conservation status can change rapidly in response to socioeconomic interests.

In addition to the institutionalized network of protected areas, the conservation of the Amazon forest can be discussed as a passive de-facto protection due to the remoteness of yet unperturbed regions (Soares-Filho

et al., 2006). Figure 20b visualizes a 'remoteness map' of the ROI$_{foot}$, based on a global study by Weiss et al. (2018) on the land-based travel times to the nearest densely-populated area. The map illustrates the close relation between accessibility and fire occurrence along major highways and certain rivers. Again, the critical role of highways in forest destruction clearly stands out in this comparison. The still remotest regions in the ROI$_{foot}$ are located in northern Pará, Amapá, French Guiana, and Suriname, being covered by the NE BT

clusters. Note here that the Arco Norte and BR-210 highways have been already been considered in the remoteness map, although they do not exist yet.

### 3.3.7 Deforestation and climate change scenarios

The construction of new roads – in particular paved all-weather highways – has a keystone role in future de-

forestation as it opens large areas for colonization and resource exploitation (e.g., Fearnside and Graca, 2006; de Carvalho and Mustin, 2017). For the ATTO observations, the currently discussed construction of the Arco Norte with the extensions of the BR-210 and BR-163 would be by all means the most severe impact with a profound perturbation of the currently still untouched NE segment of the ATTO footprint. Soares-Filho et al. (2006) conducted a policy-sensitive simulation of future deforestation scenarios. Within the

ROI$_{foot}$, the corresponding results for a "governance" (Fig. 21a) and "business as usual" (Fig. 21b) scenario show large differences and, thus, define the range of possible future deforestation trajectories. Overall, the "business as usual" scenario predicts massive deforestation that would impact the entire ATTO footprint region. However, even the "governance" scenario predicts substantial further deforestation, mostly along the major highways BR-230 and BR-319 southwest of Manaus as well as along the Arco Norte northeast of

ATTO. In both scenarios, the construction of the Arco Norte has been considered and acts as a starting point for massive deforestation in the core of the ATTO BT footprint. The varying degrees of predicted deforestation along the Arco Norte relates to differences in effectiveness of existing protected areas (compare Fig. 20a). It has to be kept in mind that many policy-related input parameters for the modelling approach are subject to high uncertainty. Nevertheless, the modelling results are valuable to assess potential future develop-

ments within the ATTO footprint. The most striking result is the prediction that the construction of the Arco Norte would initiate massive deforestation under both scenarios.



       In addition to the infrastructural perturbations, climate change tends to further increase the pressure on the Amazon ecosystem. In extreme scenarios, a large-scale rain forest die-back – i.e., a climate-driven substitution of moist forests by semi-arid and/or savanna vegetation – due to changing hydrological and seasonal regimes has been predicted (e.g., Cochrane and Laurance, 2008; Nepstad et al., 2008; Cochrane and

Barber, 2009). Furthermore, it has been reported that these effects will likely be most severe in the eastern basin as large parts of its forest are already close to the lower rainfall limit that sustains moist tropical vegetation (Zelazowski et al., 2011). Even minor changes in precipitation patterns could exceed thresholds that irreversibly push the system beyond a tipping point towards seasonal and savanna forests with strong fire feedback cycles (Malhi et al., 2009; Alencar et al., 2015). Accordingly, any changes in dry-season water

supply (i.e., precipitation or stored soil moisture) are of critical importance for the rain forest ecosystem (Boisier et al., 2015). Since climate models are known to differ substantially with respect to regional rainfall patterns in Amazonia, future spatially resolved projections are highly uncertain (Cox et al., 2008; Xie et al., 2015). However, a general drying trend towards more seasonal bioclimatic conditions as well as an increase in frequency and severity of droughts is being observed already and will likely intensify, particularly in the

eastern basin (Good et al., 2008; Lewis et al., 2011; Silva et al., 2013; Hilker et al., 2014). Accordingly, it appears to be a likely scenario that the ongoing ATTO observations will witness substantial, maybe disruptive, further transformations within the eastern basin through the synergistic effects of man-made perturbations and climate change.



## 4 Summary and conclusions

This study presents a backward trajectory (BT) and geographic data analysis as a robust characterization of spatiotemporal patterns in BT advection, climatic conditions as well as current land cover and land use patterns and future trends in the ATTO footprint region. With respect to the BT analysis, we obtained the following main results:

- The multi-year HYSPLIT BT analysis (Jan 2008 - Jun 2016) shows the characteristic central Amazonian air mass advection as the confluence of northeasterly and southeasterly trade winds, feeding boundary layer air into the deep-convective ITCZ. In response to the annual north-south ITCZ migration, the ATTO-relevant BTs follow a seasonal swing between a northeast path during the wet season (Feb-May) and a southeast path during the dry season (Aug-Nov). Furthermore, air mass advection to ATTO (boundary layer) occurs along rather compact BT tracks and at relatively low altitudes.

- The spatiotemporal BT variability includes four main wind directions: northeast (NE, accounting for ~20 % of all BTs), east-northeast (ENE, ~27 %), east (E, ~33 %), and east-southeast (ESE, ~19 %). The BTs within the main wind directions can be further subdivided into faster vs. slower wind speed regimes. Overall, 15 clusters obtained from a $k$-means cluster analysis were found to be appropriate to resolve air mass history in relation to the variability in atmospheric composition at ATTO (including more subtle aspects as a function of wind speed regimes). Southwesterly BTs account for only ~1 % of all cases and thus underline that the influence of the Manaus city area is rare, but not negligible.

- Anomalies in the BT frequency of occurrence have been observed in relation to the meridional gradient in Atlantic sea surface temperature. Specifically, an anomalously warm tropical north Atlantic tends to shift the ITCZ northwards with an increased frequency of southeasterly BT, whereas an anomalously warm tropical south Atlantic tends to shift the ITCZ southwards with an increase in northeasterly BTs.

- The northernmost advection of the BTs occurs around February and the southernmost around July. The north-to-south swing of the BTs takes ~3 months, whereas the south-to-north swing spans ~6 months.

- The BT seasonal cycle is phase-shifted relative to both the rainfall seasonality in the ATTO footprint (rainfall peak in Mar vs. minimum in Sep) as well as the seasonality in pollution abundance (pollution peak ~Sep vs. minimum ~Apr/May).

Based on the multi-year BT data, an ATTO footprint has been defined, which includes a continental area of ~$1.46 \cdot 10^6$ km$^2$. Within the footprint, GIS layers were analyzed to characterize its ATTO-relevant geospatial properties such as climatic conditions, distribution of ecoregions, land cover categories, deforestation dynamics, agricultural expansion, fire regimes, infrastructural development and protected areas. The main findings can be summarized as follows:

- Climatically, the ATTO BT footprint is characterized by tropically warm and moist conditions, however with heterogeneous precipitation patterns, spanning from moist forest to dry savanna regions. Importantly, large parts of the ATTO-relevant forests in the eastern basin are already close to or even below the lower rainfall limit that sustains moist tropical vegetation and, thus, highly vulnerable towards further drying and/or decreasing dry-season water supply.



- Ecologically, the ATTO BT footprint includes four major biome classes: (i) tropical and subtropical moist broadleaf forests, covering ~90 % of its continental area, (ii) tropical and subtropical grasslands, savannas, and shrublands in the Guianan savanna and Cerrado regions (~8 %), (iii) deserts and xeric shrublands in the Caatinga region, (~1 %), and (iv) mangrove forests along the Atlantic coast (~1 %). On a finer ecological scale, the footprint covers about a dozen distinct ecoregions. About 54 % of the continental BT footprint region are considered as being in an untouched state.

- The ATTO-relevant effective land cover mix consists of a rain forest contribution of ~60 %, water bodies (~30 %), wetlands and flooded forests (~5 %), agricultural areas (~4 %), and shrub- and grasslands (~3 %). Strong geospatial gradients were found for some of the categories, resulting in different and characteristic land cover mixes for the BT main wind directions. Classified by land cover categories, two different seasonal patterns in net primary productivity were observed: (i) A seasonal cycle being in-phase with solar radiation (sunlight-enhanced growth) and occurring in moist deep-rooting forests that buffer dry season drought stress. (ii) A seasonal cycle being in-phase with precipitation (rainfall-constrained growth) occurring in low/sparse vegetation categories.

- In terms of deforestation and agricultural expansion, the ATTO site covers the full range of deforestation dynamics spanning from major forest fragmentation and clearing hot spots in the SE (deforestation rates ~1 % a$^{-1}$) to mostly untouched forests in the NE (deforestation rates ~0.01 % a$^{-1}$). Several major highways (i.e., the "soybean corridor" BR-163 and the Trans-Amazon highway BR-230), which have initiated strong deforestation in previously inaccessible regions, are located in the ATTO footprint.

- The fire map shows a heterogeneous distribution, which is closely linked to the agro-industrial expansion. Towards the ESE of the footprint, large-scale fire activities are observed, due to the combination of numerous man-made ignition sources and increased forest flammability as a function of forest fragmentation and degradation. Towards the NE, the forests maintain their fire-immune moist climate and, thus, show no fire activity. A land cover-categorized and BT-weighted fire mix within the ATTO footprint indicates a dominant fraction of fires in rain forest areas (on average 54-63 %), followed by fires in shrub and grassland categories (~10-16 %), and fires in agricultural areas (11 to 28 %).

- Existing or planned infrastructure in the footprint region is of significant relevance for the ATTO observations as it perturbs the natural biosphere-atmosphere exchange and/or represents sources of primary and secondary pollutants. We consider the following categories as relevant for the ATTO observations: extended urban areas and power plants, at least six major dams and reservoirs (including the controversial Belo Monte Dam) with largely flooded areas and decaying biomass, ship traffic in front of the Atlantic coast and particularly on inland waters in the basin, as well as active mining spots as major forest destruction and industrial pollution sources. Particularly, the construction of (all-weather) highways has a crucial role in moving deforestation frontiers northwards. Existing highways in the footprint have acted as origins for significant forest loss. Moreover, a realization of further planned highways (i.e., the Arco Norte by extending the BR-210 and BR-163) in the footprint would be a severe threat for the currently unperturbed footprint regions.



- About 41 % of the ATTO footprint are conserved areas with protection categories of varying effective-ness in deforestation mitigation. Beyond this institutionalized network of protected areas, large parts of the northern footprint region are passively protected by their remoteness.

Based on the BT and GIS results, we draw the following conclusions with respect to the currently ongoing and future observations on biosphere-atmosphere exchange at ATTO:

- The effective continental footprint region includes a wide range of perturbation states. On one hand, it includes the areas of massive deforestation and high fire activity within the arc of deforestation towards the southeast (i.e., southern and eastern Pará as well as Maranhão). On the other hand, remote and
mostly pristine forests that maintain their fire-immunity are covered by the footprint towards the north-east (i.e., northern Pará, Amapá and French Guiana). Accordingly, the frontier of infrastructural devel-opment, agro-industrial expansion, and resource exploitation, which has been moving northwards over decades, is located right in the middle of the footprint region. This emphasizes the unique location of the ATTO site allowing to monitor atmospheric chemistry and physics under the contrasting influence of
strongly perturbed vs. pristine conditions and, thus to explore specific man-made influences on a process level. We suggest that the backward trajectory data presented here may be a helpful tool to filter ATTO data sets in order to derive geospatially dependent signals.

- Measurements and modelling results indicate that the eastern basin is most susceptible to global warm-ing and changing precipitation patterns. In fact, modelling studies have projected a large-scale rain forest
die-back for the 21$^{st}$ century – starting in the eastern basin – due to profound changes in hydrological, seasonal, and fire regimes that may irreversibly push the ecosystem beyond stability thresholds. Since the eastern basin is almost fully covered by its footprint, the ATTO site will provide valuable measure-ments from one of the hot spots of global change and a potential tipping point in the climate system.

- Currently, very clean background conditions as well as air masses that experienced an unperturbed ex-
change with extended rain forest areas can still be observed at ATTO. However, these conditions occur only episodically and only in the wet season. In light of the infrastructural development as well as the increasing climate change-related pressure on the eastern basin, it is likely that the time window for ob-servations of a pristine rain forest atmosphere will irreversibly close in the upcoming years.

- For the near future, the following specific perturbations will presumably have a large influence on the
ATTO measurements: An extension of the BR-210 and BR-163 highways would be the largest threat as this implies major forest destruction in the core of the ATTO footprint. The finalization of the Belo Monte Dam and emissions associated with the flooding of large areas will likely be observed at ATTO. In addition, climate change and strong drought events may irreversibly alter parts of the footprint in the eastern basin with currently unpredictable implications.



*Data availability.* The GIS data used in this study is freely available from sources and websites as specified in the supplement Sect. S1.1. The HYSPLIT backward trajectory data (i.e., center lines of 15 backward trajectory clusters and time-resolved cluster flag for data filtering) has been deposited as a supplement files in NASA Ames format. The contour lines representing areas with characteristic air mass residence times are provides as shapefiles. Furthermore, high resolution versions of all maps in this study are provided as supplementary information. For specific data requests or detailed information on the deposited data, please refer to the corresponding author c.pohlker@mpic.de.

*Special issue statement.* This article is part of the Amazon Tall Tower Observatory (ATTO) special issue. It does not belong to a conference.

*Acknowledgements.* This work has been supported by the Max Planck Society (MPG). For the operation of the ATTO site, we acknowledge the support by the German Federal Ministry of Education and Research (BMBF contract 01LB1001A) and the Brazilian Ministério da Ciência, Tecnologia e Inovação (MCTI/FINEP contract 01.11.01248.00) as well as the Amazon State University (UEA), FAPEAM, LBA/INPA and SDS/CEUC/RDS-Uatumã. This paper contains results of research conducted under the Technical/Scientific Cooperation Agreement between the National Institute for Amazonian Research, the State University of Amazonas, and the Max-Planck-Gesellschaft e.V.; the opinions expressed are the entire responsibility of the authors and not of the participating institutions. Céline Degrendele was funded by the core facilities of the RECETOX Research Infrastructure, project LM2015051, and by ACTRIS-CZ, project LM2015037, funded by the Ministry of Education, Youth and Sports of the Czech Republic under the activity "Projects of major infrastructures for research, development and innovations". We highly acknowledge the support by the Instituto Nacional de Pesquisas da Amazônia (INPA). We would like to thank Reiner Ditz, Jürgen Kesselmeier, Susan Trumbore, Alberto Quesada, Thomas Disper, Thomas Klimach, Andrew Crozier, Björn Nillius, Uwe Schulz, Steffen Schmidt, Niro Higuchi, Antonio Ocimar Manzi, Alcides Camargo Ribeiro, Hermes Braga Xavier, Elton Mendes da Silva, Nagib Alberto de Castro Souza, Adir Vasconcelos Brandão, Amauri Rodriguês Perreira, Antonio Huxley Melo Nascimento, Thiago de Lima Xavier, Josué Ferreira de Souza, Roberta Pereira de Souza, Bruno Takeshi, and Wallace Rabelo Costa for technical, logistical, and scientific support within the ATTO project. Moreover, we thank Britaldo Soares-Filho, Florian Wittmann, Henrique Barbosa, Scot T. Martin, Xuguang Chi, H. Su, Isabella Hrabě de Angelis, and Ovid Krüger for scientific support and stimulating discussions. The authors gratefully acknowledge the NOAA Air Resources Laboratory (ARL) for the provision of the HYSPLIT transport and dispersion model and/or READY website (http://www.ready.noaa.gov) used in this publication. The authors further gratefully acknowledge the European Space Agency (ESA) GlobCover 2009 Project for providing the global land cover maps.



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



**Table A1.** List of acronyms, abbreviations, and symbols.

| Acronym | Description |
| --- | --- |
| AGL | above ground level |
| AMAZE-08 | Amazonian Aerosol Characterization Experiment 2008 |
| AMO | Atlantic multidecadal oscillation |
| ATTO | Amazon Tall Tower Observatory |
| BL | Boundary layer |
| BT | back trajectory |
| CA | cluster analysis |
| CIAT | International center for tropical agriculture |
| E | BT clusters towards east |
| ENE | BT clusters towards east-northeast |
| ENSO | El Niño-Southern Oscillation |
| ESA | European space agency |
| ESE | BT clusters towards east-southeast |
| $f$ | Frequency of occurrence of BTs |
| FAO | Food and Agriculture Organization |
| FFDAS | fossil fuel data assimilation system |
| FLEXPART | FLEXible PARTicle dispersion model |
| FoO | frequency of occurrence |
| GFAS | |
| GHCN | global historical climatology network |
| GIS | geographic information system |
| GUF | French Guiana |
| GUY | Guyana |
| HYSPLIT | Hybrid Single Particle Lagrangian Trajectory Model |
| INPE | Instituto Nacional de Pesquisas Espaciais |
| IPCC | intergovernmental panel on climate change |
| ISLSCP II | international satellite land-surface climatology project, initiative II |
| ITCZ | intertropical convergence zone |
| IUCN | International Union for Conservation of Nature |
| LBA | Large Scale Biosphere-Atmosphere Experiment in Amazonia |
| LRT | long-range transport |
| MODIS | Moderate Resolution Imaging Spectroradiometer |
| NASA | National Aeronautics and Space Administration |
| NCAR | National Center for Atmospheric Research |
| NDVI | normalized difference vegetation index |
| NE | BT clusters towards northeast |
| NOAA | National Oceanic and Atmospheric Administration |
| ONI | oceanic niño index |
| OSM | OpenStreetMap |
| $P$ | precipitation rate |
| $P_{BT}$ | cumulative precipitation along 3-day BTs |
| PCA | principle component analysis |
| RCP | representative concentration pathways |
| PERSIANN-CDR | Precipitation Estimation from Remotely Sensed Information using Artificial Neural Networks for Climate Data Record |
| RENCA | national reserve of copper and associates |


| | |
|---|---|
| ROI | region of interest |
| ROI$_{foot}$ | region of interest covering the continental part of the ATTO BT footprint (62°W, 40° W, 8° S, 6° N) |
| SEDAC | socioeconomic data and applications center |
| SPOT | Satellite Pour l'Observation de la Terre |
| SRTM | shuttle radar topography mission |
| SST | sea surface temperature |
| SUR | Suriname |
| SW | BT cluster towards southwest |
| TNA | tropical northern Atlantic |
| TSA | tropical southern Atlantic |
| UNLC | UN land cover classification |
| UNEP | United Nations Environment Program |
| UTC | coordinated universal time |
| WDPA | world database on protected areas |
| WGS84 | world geodetic system from 1984 |
| WWF | world wildlife fund |
| WMO | world meteorological organization |



**Table 1.** Absolute numbers of individual backward trajectories (BTs) and their frequency of occurrence, resolved by main directions of BT advections (i.e., NE, ENE, E, and ESE) as well as for all 15 BT clusters from $k$-means cluster analysis (see Fig. 4).

| BT clusters | Absolute number of BTs in clusters | BT frequency of occurrence ($f$) [%] |
|---|---|---|
| NE1 | 4261 | 5.7 |
| NE2 | 5836 | 7.8 |
| NE3 | 4679 | 6.3 |
| **Sum of NE clusters** | **14776** | **19.8** |
| ENE1 | 3229 | 4.3 |
| ENE2 | 7054 | 9.5 |
| ENE3 | 6453 | 8.7 |
| ENE4 | 3192 | 4.3 |
| **Sum of ENE cluster** | **19928** | **26.8** |
| E1 | 6942 | 9.3 |
| E2 | 8403 | 11.3 |
| E3 | 6633 | 8.9 |
| E4 | 3058 | 4.1 |
| **Sum of E clusters** | **25036** | **33.4** |
| ESE1 | 2425 | 3.3 |
| ESE2 | 5887 | 7.9 |
| ESE3 | 5419 | 7.3 |
| **Sum of ESE clusters** | **13731** | **18.5** |
| SW1 | 1025 | 1.4 |
| **Total** | **74496** | **100** |



**Table 2.** Summary of GlobCover 2009 categories that account for 99.9 % of land cover variability within the ATTO BT footprint with specification of relative contributions of individual categories. A comprehensive summary with land cover mix for ATTO BT footprint is available in Table S1.

| GlobCover 2009 ID categories | | Contribution to ATTO BT footprint [%] |
| --- | --- | --- |
| **ID** | **Description** | |
| 40 | Closed to open (>15%) broadleaved evergreen or semi-deciduous forest (>5m) | 57.2 |
| 210 | Water bodies | 30.1 |
| 160 | Closed to open (>15%) broadleaved forest regularly flooded (semi-permanently or temporarily) - Fresh or brackish water | 3.0 |
| 130 | Mosaic vegetation (grassland/shrubland/forest) (50-70%) / cropland (20-50%) | 2.8 |
| 20 | Mosaic cropland (50-70%) / vegetation (grassland/shrubland/forest) (20-50%) | 2.1 |
| 180 | Closed to open (>15%) grassland or woody vegetation on regularly flooded or waterlogged soil - Fresh, brackish or saline water | 1.7 |
| 30 | Mosaic vegetation (grassland/shrubland/forest) (50-70%) / cropland (20-50%) | 1.5 |
| 14 | Rainfed croplands | 0.6 |
| 110 | Mosaic forest or shrubland (50-70%) / grassland (20-50%) | 0.3 |
| 140 | Closed to open (>15%) herbaceous vegetation (grassland, savannas or lichens/mosses) | 0.3 |
| 50 | Closed (>40%) broadleaved deciduous forest (>5m) | 0.2 |




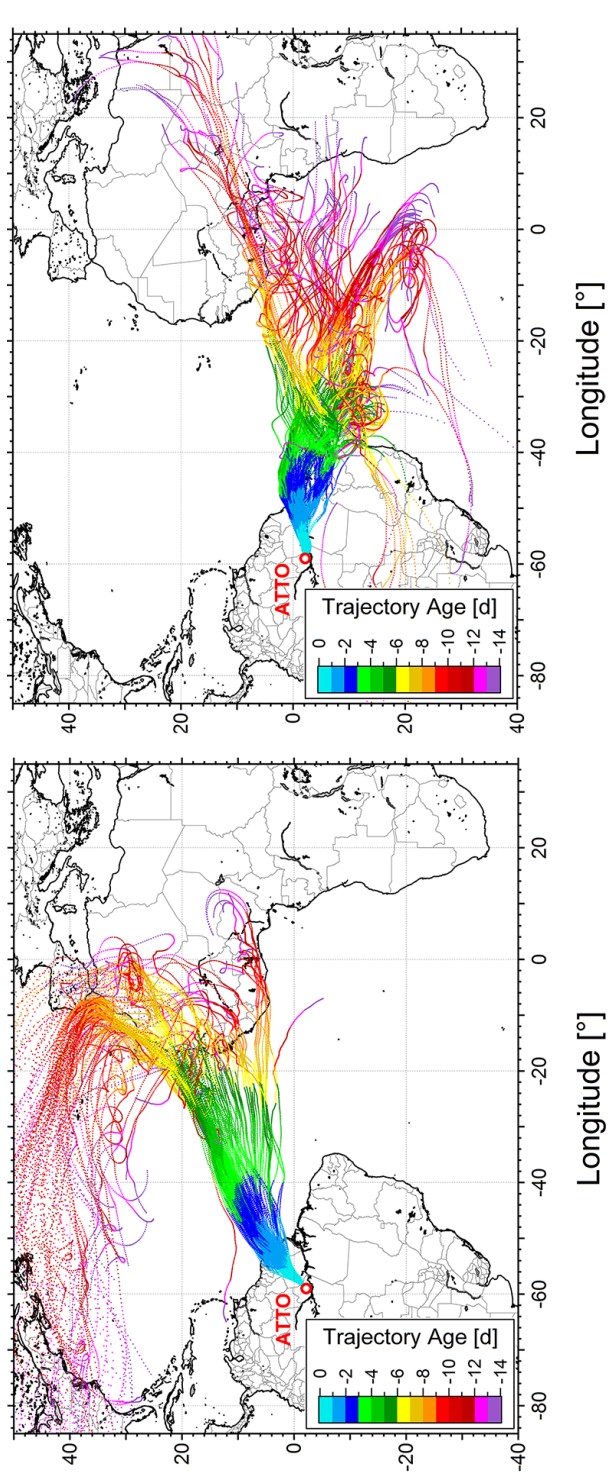

**Figure 1.** Fourteen-day HYSPLIT backward trajectory (BT) ensembles (starting height 1000 m AGL) with color coding representing BT transport times: (left) north-east track of BTs from 1 to 31 March 2014, which is characteristic for the Amazonian wet season conditions, and (right) southeast track of BTs from 1 to 30 Sep 2014, which is characteristic for the Amazonian dry season conditions. The transport times indicate that the BTs spend about 2-3 days over the South American continent. Moreover, transport time from the African coast to ATTO takes about 6-7 days.





**Figure 2.** HYSPLIT back trajectory (BT) ensembles showing the large scale trade wind circulation in the Atlantic region and the pronounced seasonal oscillation between Northern and Southern hemispheric influence at ATTO by mean of air mass residence time maps (**a**, **c** and **e**) and average BT height maps (**b**, **d** and **f**). The BT ensembles comprise all 74 496 individual 9-day BTs, spanning a multi-year time period from 01 Jan 2008 until 30 June 2016. The BT analysis was conducted for the start heights 200 m, 1000 m, and 2000 m AGL. The contour lines in **b**, **d** and **f** were adopted from **a**, **c** and **e** to visualize the patterns in relative BT density on top of the average BT height map. Corresponding data for the BT start heights 80 m and 4000 m can be found in Fig. S6.



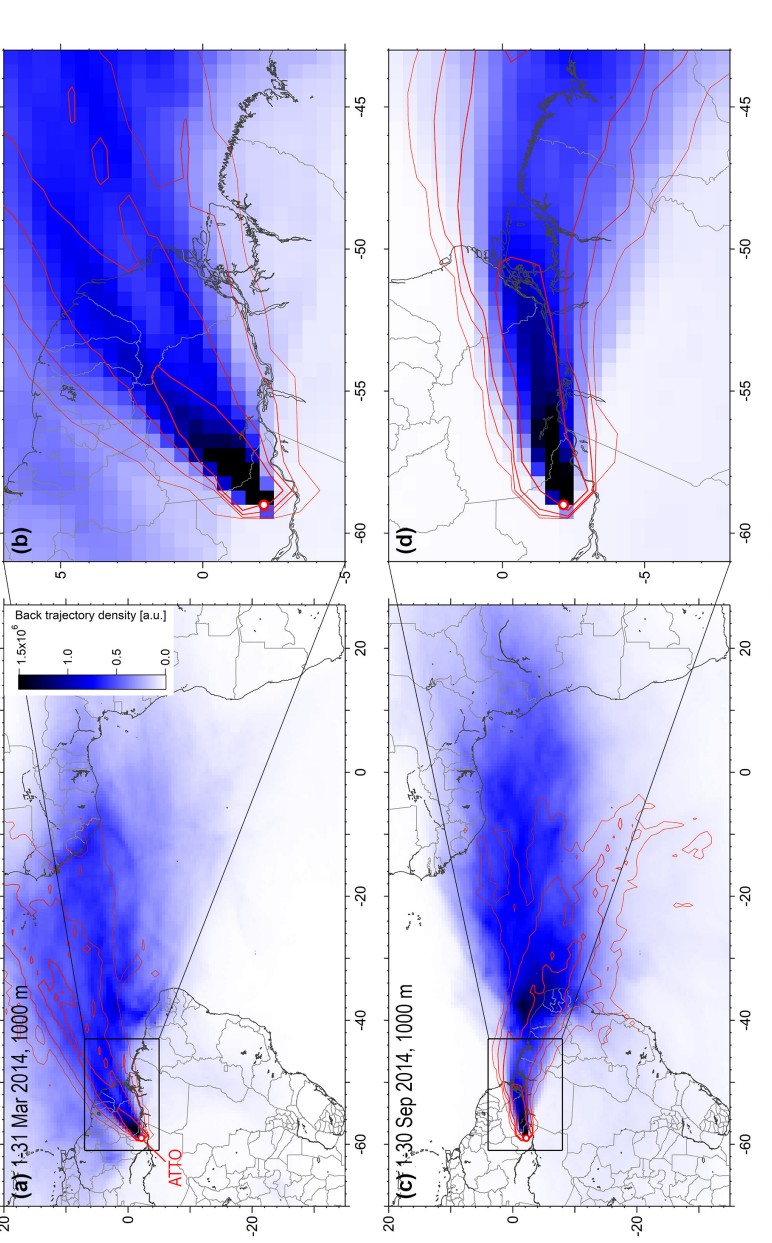

**Figure 3.** Comparison of FLEXPART (blue) and HYSPLIT (red) back trajectory results for wet season (1-31 Mar 2014, BT start height 1000 m AGL, **a** and **b**) and dry season (1-30 Sep 2014, BT start height 1000 m AGL, **c** and **d**) conditions. Both models were run under comparable configuration settings (see Sect. 2.3). FLEXPART results are shown as image plot (color code represents relative BT density). HYSPLIT results are shown as contour lines (similar to Fig. 2), representing BT ensembles. An alternative representation of the results can be found in Fig. S8.



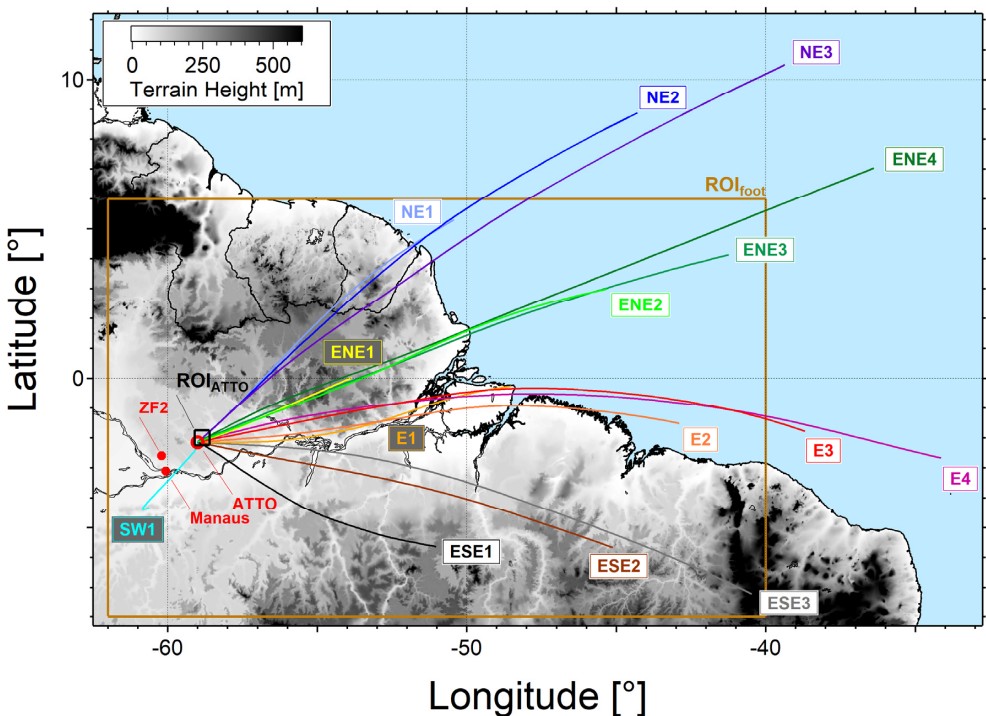

**Figure 4.** Map of northeast Amazon Basin with 15 clusters from systematic *k*-means back trajectory (BT) cluster analysis based on entire HYSPLIT data set (3-day BTs, 1 Jan 2008 - 30 Jun 2016, start height 1000 m). Back trajectory clusters show that air masses arrive at the ATTO site almost exclusively from north-eastern to south-eastern directions. Four major wind directions can be discriminated: (i) Northeastern (NE) clusters NE1, NE2, and NE3; (ii) east-northeastern (ENE) clusters ENE1, ENE2, ENE3, and ENE4; (iii) eastern (E) clusters E1, E2, E3, and E4; (iv) east-southeastern (ESE) clusters ESE1, ESE2, and ESE3. A topographic map is represented by a grey scale, which is capped at 600 m. The regions of interest are shown: a relatively small ROI around ATTO (59.1°W, 58.6° W, 2.25° S, 1.75° S) and ROI$_{foot}$ (62°W, 40° W, 8° S, 6° N) are shown as frames. For comparison, an analogous cluster analysis for BTs at a start height of 200 m can be found in Fig. S9.





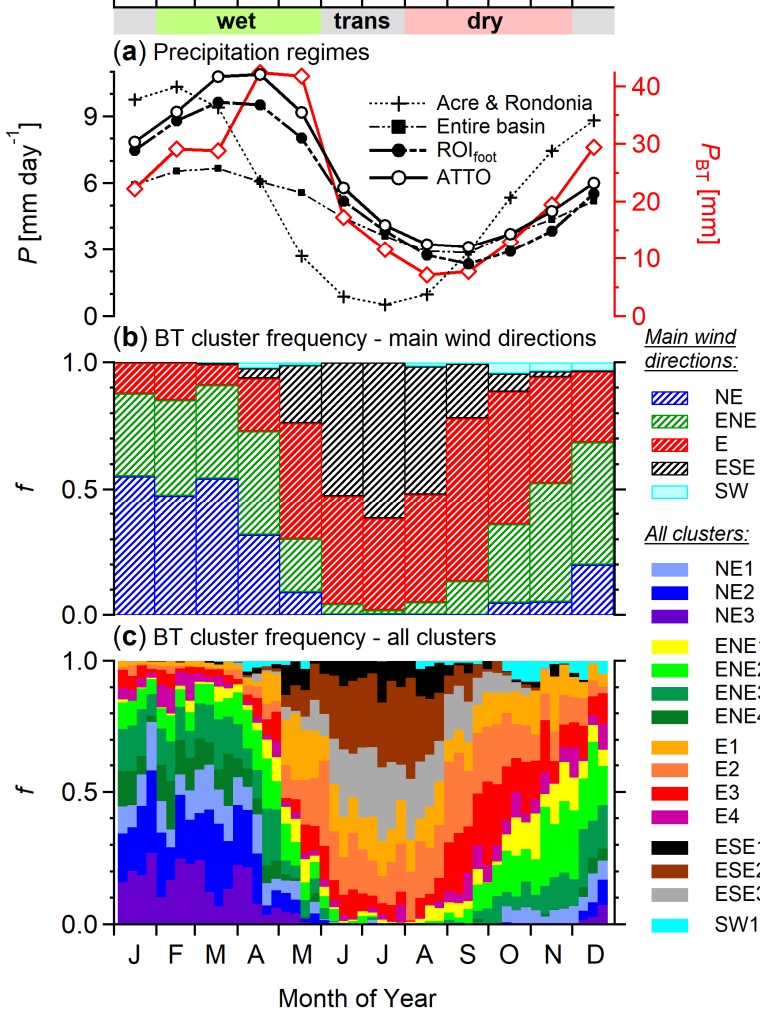

**Figure 5.** Seasonality in precipitation regimes and ATTO-relevant back trajectory advection. **(a)** Area-averaged precipitation rates, $P$, for the ATTO region (see Fig. 4), the $ROI_{foot}$ (continental part only, see Fig. 4), the entire basin (see Amazon watershed region in Fig. 7), and the states Acre and Rondônia (see Fig. 7). Furthermore, cumulative precipitation along 3-days BTs tracks, $P_{BT}$, is shown. All precipitation data is shown as monthly means. **(b)** Frequency of occurrence, $f$, of main directions of BT advection from NE, ENE, E, ESE, and SW. Data is shown as monthly means. **(c)** Frequency of occurrence, $f$, of all 15 BT clusters (see Fig. 4). Data is shown as weekly means. Colored shading in top row represents aerosol-related definition of seasonality in ATTO region according to M. Pöhlker et al. (2016).





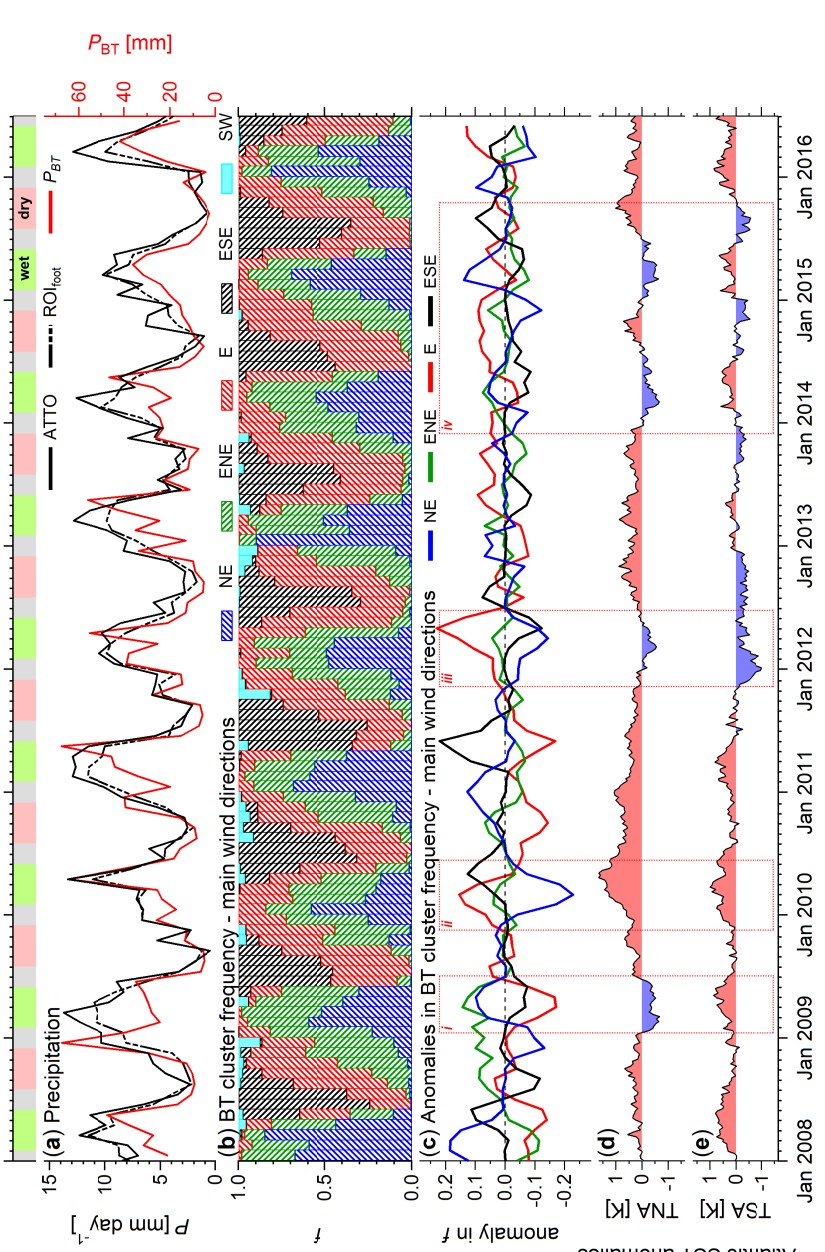

**Figure 6.** Multiyear variability and anomalies in BT advection in relation to Atlantic sea surface temperatures. (**a**) Area-averaged precipitation rates, $P$, for the $\mathrm{ROI_{ATTO}}$ and continental part of the $\mathrm{ROI_{foot}}$ (for both, see Fig. 4). Furthermore, cumulative precipitation along 3-days BTs tracks, $P_{\mathrm{BT}}$, is shown. All precipitation data is shown as monthly means. (**b**) Frequency of occurrence, $f$, of main directions of BT advection from NE, ENE, E, ESE, and SW. Data is shown as monthly means. (**c**) Anomalies in frequency, $f$, of main directions in BT advection. Data is show n as monthly means. Red boxes *i*, *ii*, *iii*, and *iv* highlight examples of pronounced anomalies in $f$. (**d** and **e**) Anomalies in sea surface temperature of tropical north Atlantic (TNA) and tropical south Atlantic (TSA). Colored shading in top row represents aerosol-related definition of seasonality in ATTO region according to M. Pöhlker et al. (2016).





**Figure 7.** Overview map of the Amazon Basin – here represented by the watershed region of the Amazon River and its tributaries – showing the location of the ATTO site and the geographic extent of its footprint. The footprint is represented by (i) an air mass residence time map based on the entire BT ensemble (color code), (ii) contour lines representing the largest 1 %, 5 %, 10 %, 25 %, and 50 % of air mass residence times, and (iii) by 15 cluster center lines from BT cluster analysis (blue dashed lines, see Fig. 4). Green areas represent forest cover map (status 2000) according to Hansen et al. (2013). Red areas represent total forest loss from 2000 to 2014 according to Hansen et al. (2013). The green thick line represents Amazon Basin watershed region. The red rectangular shape highlights the region of interest ROI$_{foot}$ (62° W to 40° W; 8° S to 6° N).





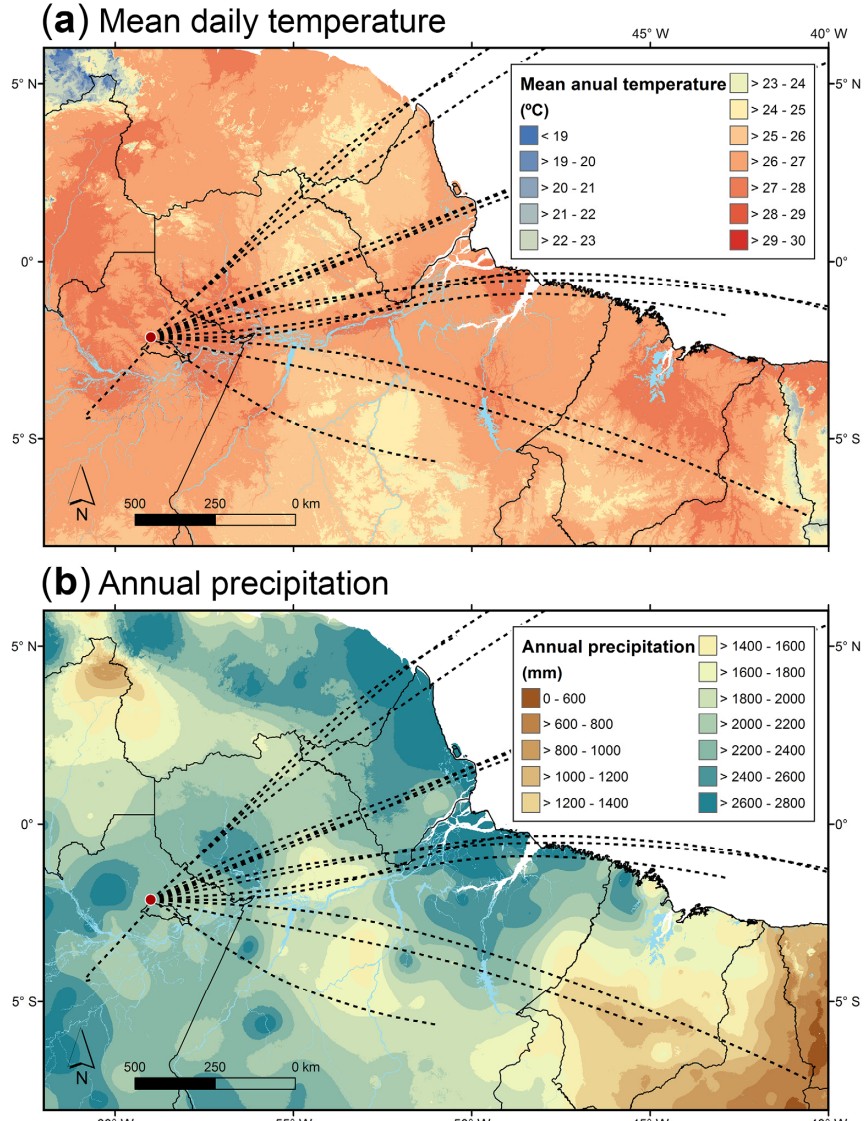

**Figure 8.** Map of the ATTO-relevant eastern Amazon Basin (ROI$_{foot}$) combining the backward trajectory (BT) data with GIS data layers of (**a**) mean daily temperature and (**b**) annual precipitation, both obtained from WorldClim database (Hijmans et al., 2005; worldclim.org). The BT data is represented as center lines of the 15 BT cluster (black dashed lines, see Fig. 4).



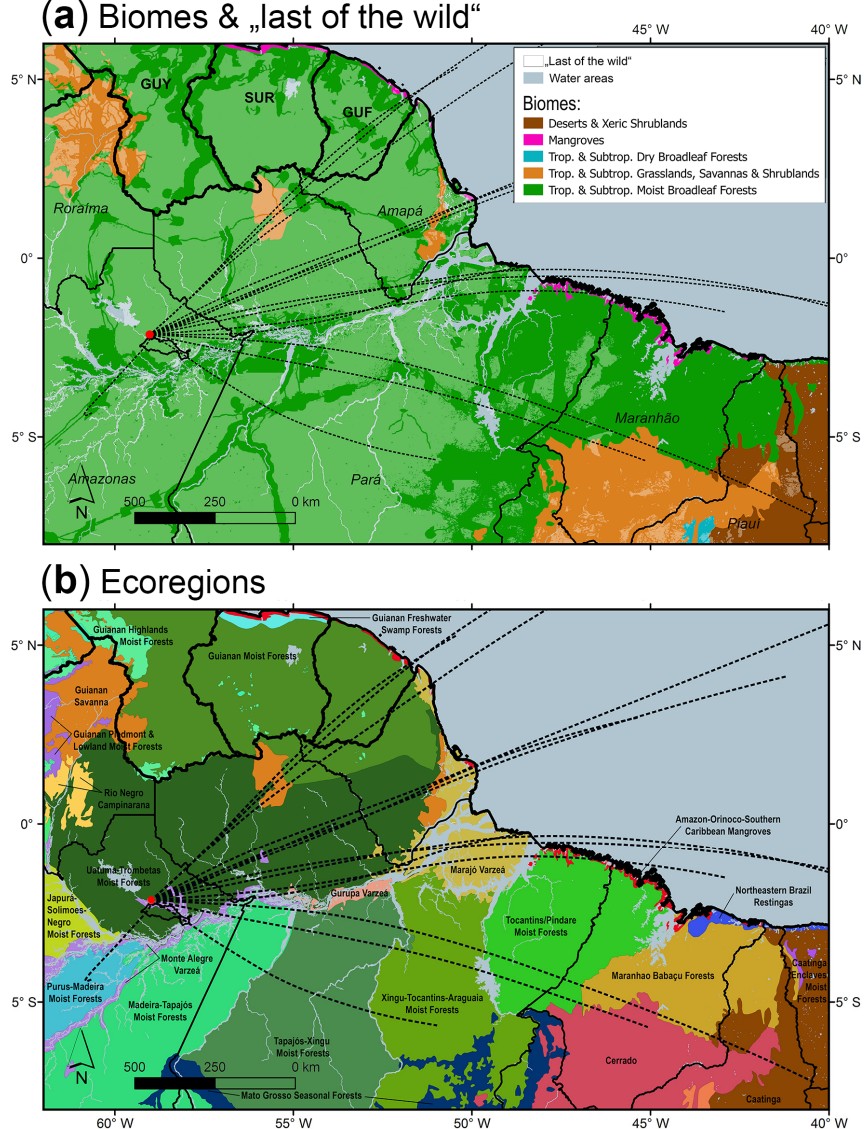

**Figure 9.** Map of the ATTO-relevant eastern Amazon Basin (ROI$_{foot}$) combining the backward trajectory (BT) data with GIS data layers of (**a**) a biome classification according to Olson et al. (2001) and a map of the wildest/most unperturbed areas in the corresponding biomes ("last of the wild", shaded in grey on top of the biome classification map) according to Sanderson et al. (2002) as well as (**b**) ecoregions according to Olson et al. (2001). The BT data is represented as center lines of the 15 BT cluster (black dashed lines, see Fig. 4).





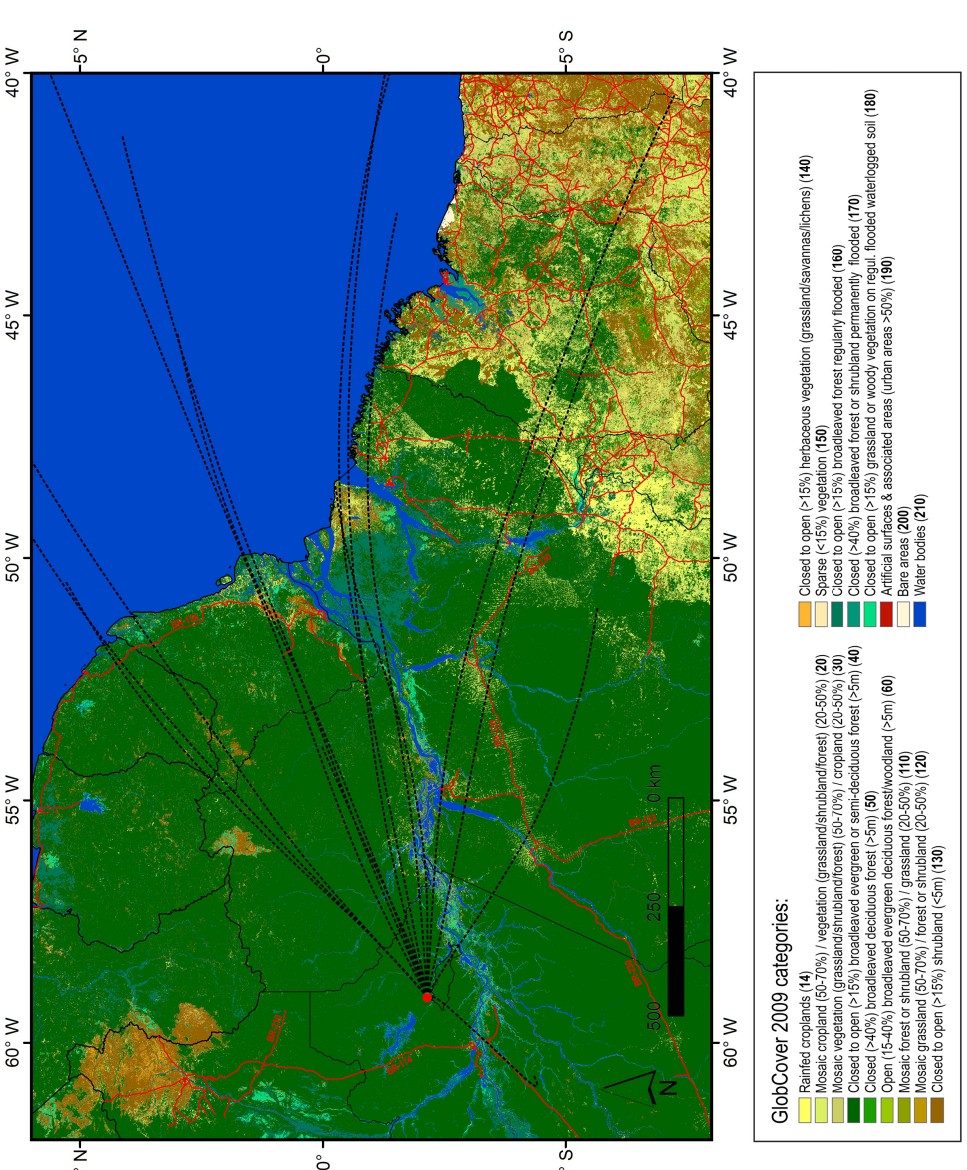

**Figure 10.** Map of the ATTO-relevant eastern Amazon Basin (ROI$_{foot}$) combining the backward trajectory (BT) data with GIS layers of a land cover map based on the GlobCover 2009 data (Arino et al., 2008) and a map of major roads. The BT data is represented as center lines of the 15 BT cluster (black dashed lines, see Fig. 4).



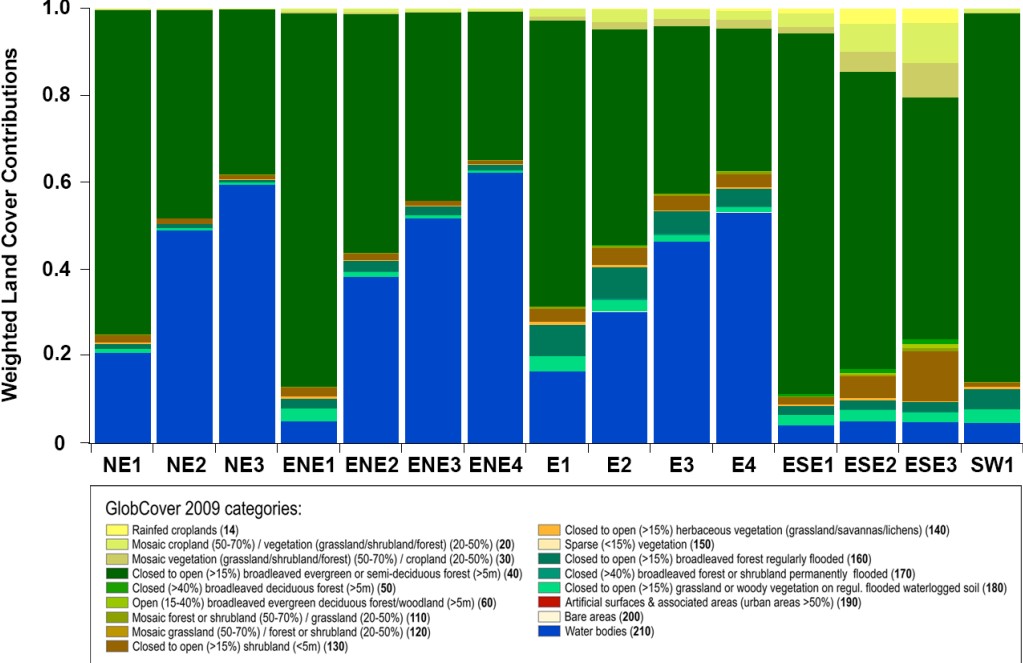

**Figure 11.** Quantitative characterization of land cover types (see Fig. 10) in all 15 backward trajectory (BT) cluster footprints (see Fig. S10). The land cover contributions have been weighted with the relative BT density and, thus, represent an ATTO-relevant 'land cover mix'. A comprehensive summary on the land cover mix within the ATTO BT footprint can be found in Table S1.



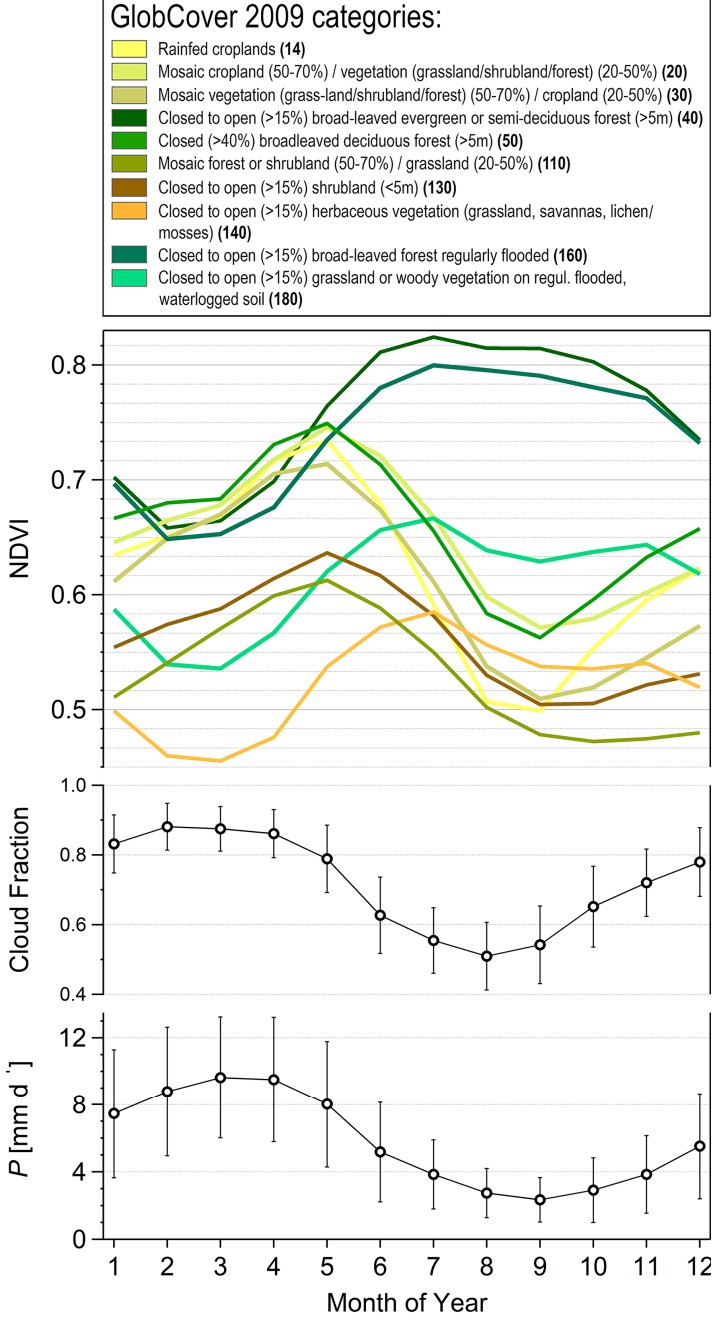

**Figure 12.** Seasonal cycles in normalized difference vegetation index (NDVI) for the ROI$_{foot}$-relevant Glob-Cover 2009 categories as specified in Table 2. The NDVI data is provided as monthly means. Seasonal cycles for precipitation and cloud cover, which indirectly represents solar radiation, have been added and are shown as monthly means with error bars representing one standard deviation.





**Figure 13.** Map of the ATTO-relevant eastern Amazon Basin (ROI$_{foot}$) combining the backward trajectory (BT) data with GIS layers of (i) a forest cover map of the year 2000, according to Hansen et al. (2013), (ii) a map on annual forest loss within the period 2001-2014 (Hansen et al., 2013), and (iii) contour lines representing the dominant biomes (see Fig. 9a). The BT data is represented as center lines of the 15 BT cluster (blue dashed lines, see Fig. 4). A map of the ROI$_{foot}$ with afore mentioned GIS layers is shown in (**a**), whereas (**b**) and (**c**) zoom into two smaller regions of particular interest.



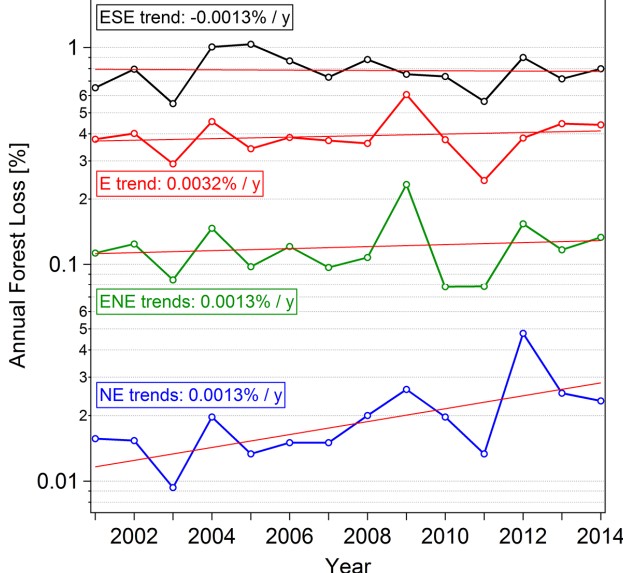

**Figure 14.** Quantitative characterization of forest loss trends in cluster BT footprints for main wind directions NE, ENE, E, and ESE, based on Fig. 13. The annual forest loss has been calculated relative to the forest cover in the year 2000. The forest loss data in the cluster BT footprints has been weighted by the air mass residence time. The corresponding trends for all BT clusters is shown in Fig. S15.





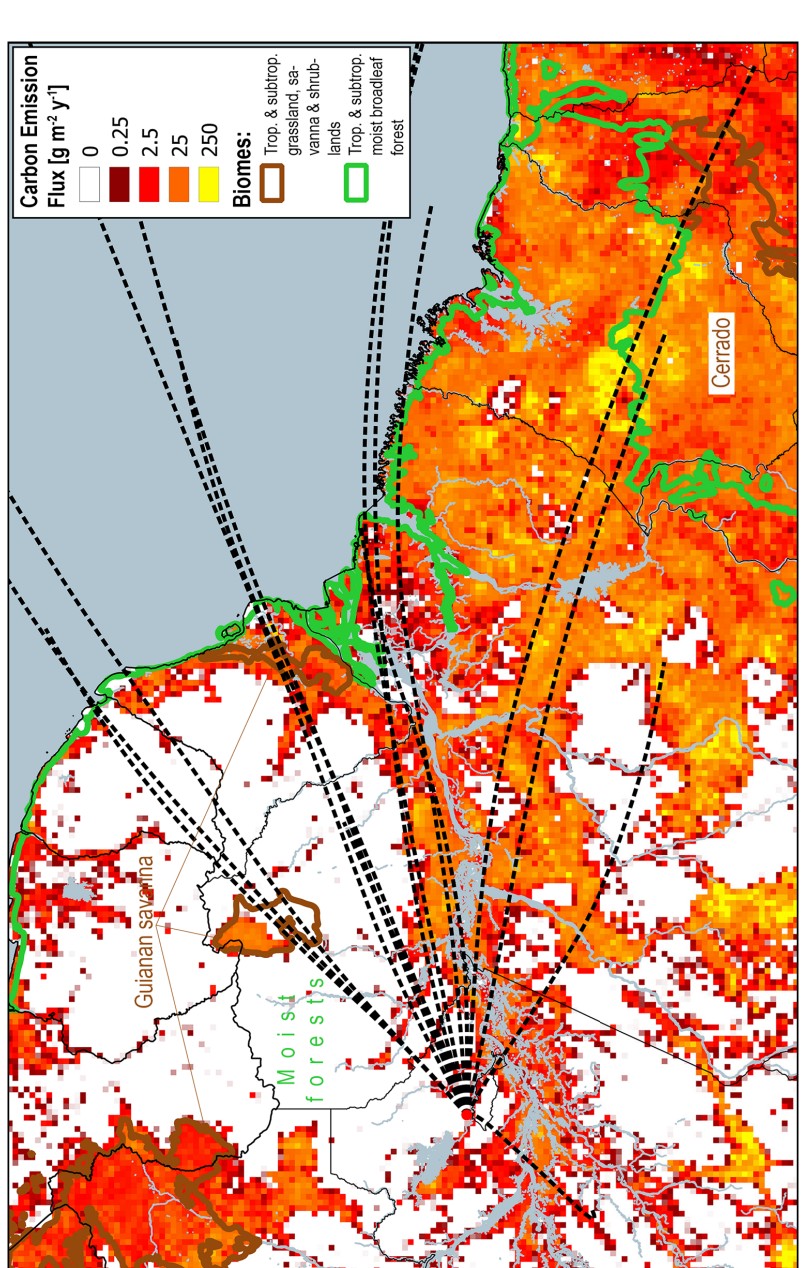

**Figure 15.** Map of the ATTO-relevant eastern Amazon Basin (ROI$_{\text{foot}}$) combining the backward trajectory (BT) data with (i) a GFAS-derived map of the average fire-related carbon emission flux within time period from 2003 to 2017 according to Kaiser et al., (2011) and (ii) a biome classification according to Olson et al., (2001) (see Fig. 9a). The pixel size of the fire map equals ~11 km. The BT data is represented as center lines of the 15 BT cluster (black dashed lines, see Fig. 4). For comparison, Fig. S16 shows a fire map with higher spatial resolution (~1 km resolution), which shows the same geospatial patterns.





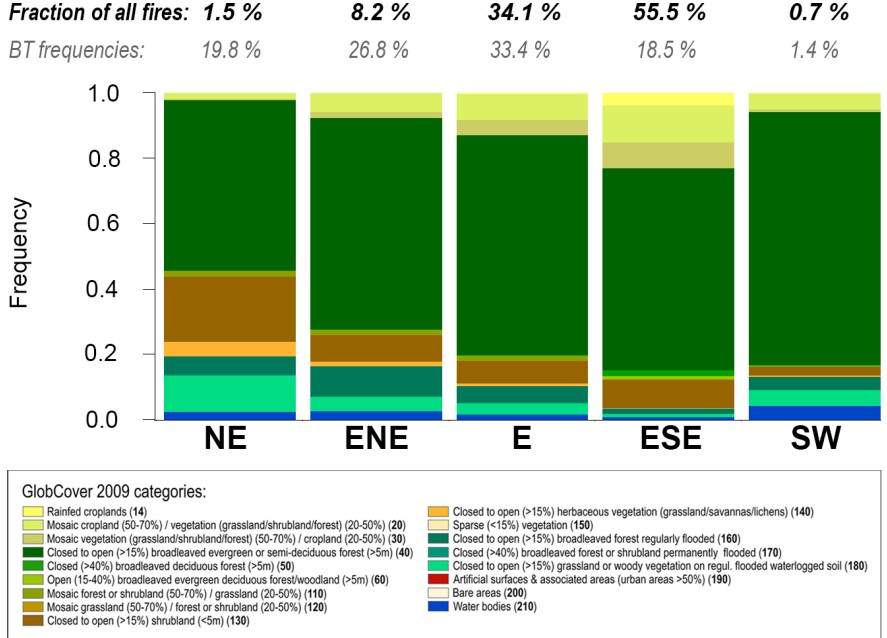

**Figure 16.** Relative fractions of ATTO-relevant fires (weighted with air mass residence times) within the different land cover categories, discriminated by major BT directions: NE, ENE, E, ESE, and SW. Fire analysis is based on the INPE database (see Sect. S1.1). Results shown here are averages of corresponding year-to-year data (2000-2016) as shown in Fig. S17. Fractions of all counted fires in this analysis are provided along with BT frequencies for comparison (see Table 1).




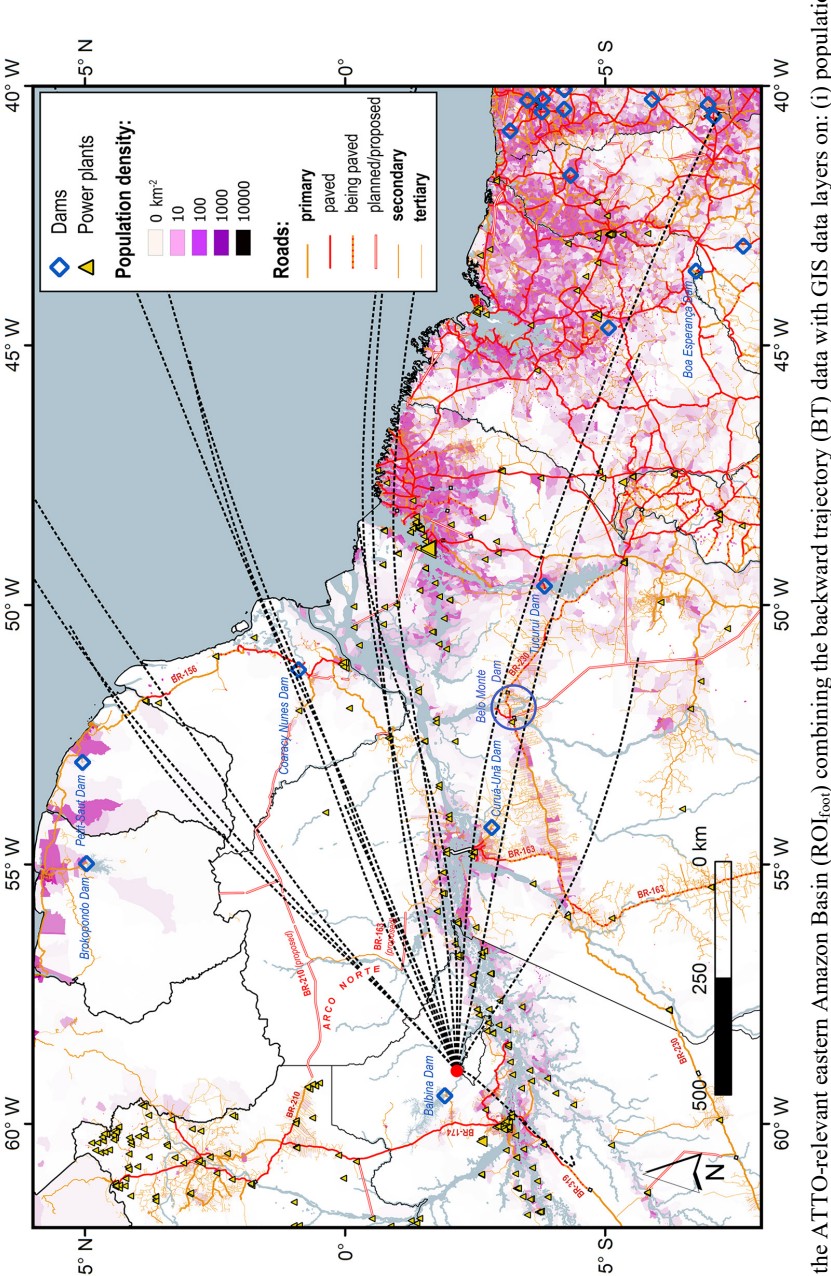

**Figure 17.** Map of the ATTO-relevant eastern Amazon Basin (ROI$_{foot}$) combining the backward trajectory (BT) data with GIS data layers on: (i) population density, (ii) thermoelectric power plants (marker size represents capacity), (iii) major dams and reservoirs, and (iv) the road network. The BT data is represented as center lines of the 15 BT cluster (black dashed lines, see Fig. 4).



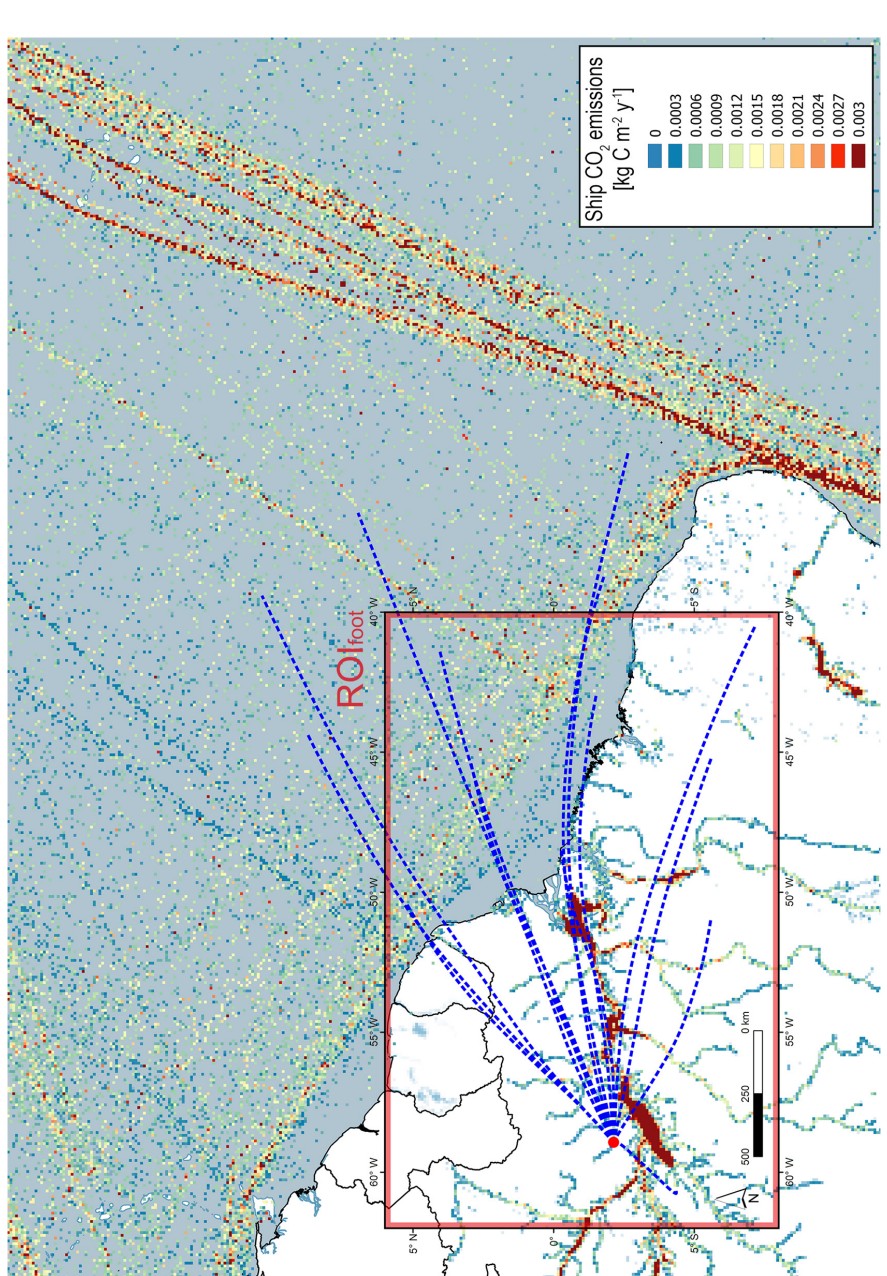

**Figure 18.** Map of eastern Amazon Basin, including $ROI_{foot}$, and western Atlantic Ocean in combination with ship tracks. Ship tracks are represented by modelled shipping-related $CO_2$ emission according to Asefi-Najafabady et al. (2014). The BT data is represented as center lines of the 15 BT cluster (blue dashed lines, see Fig. 4).





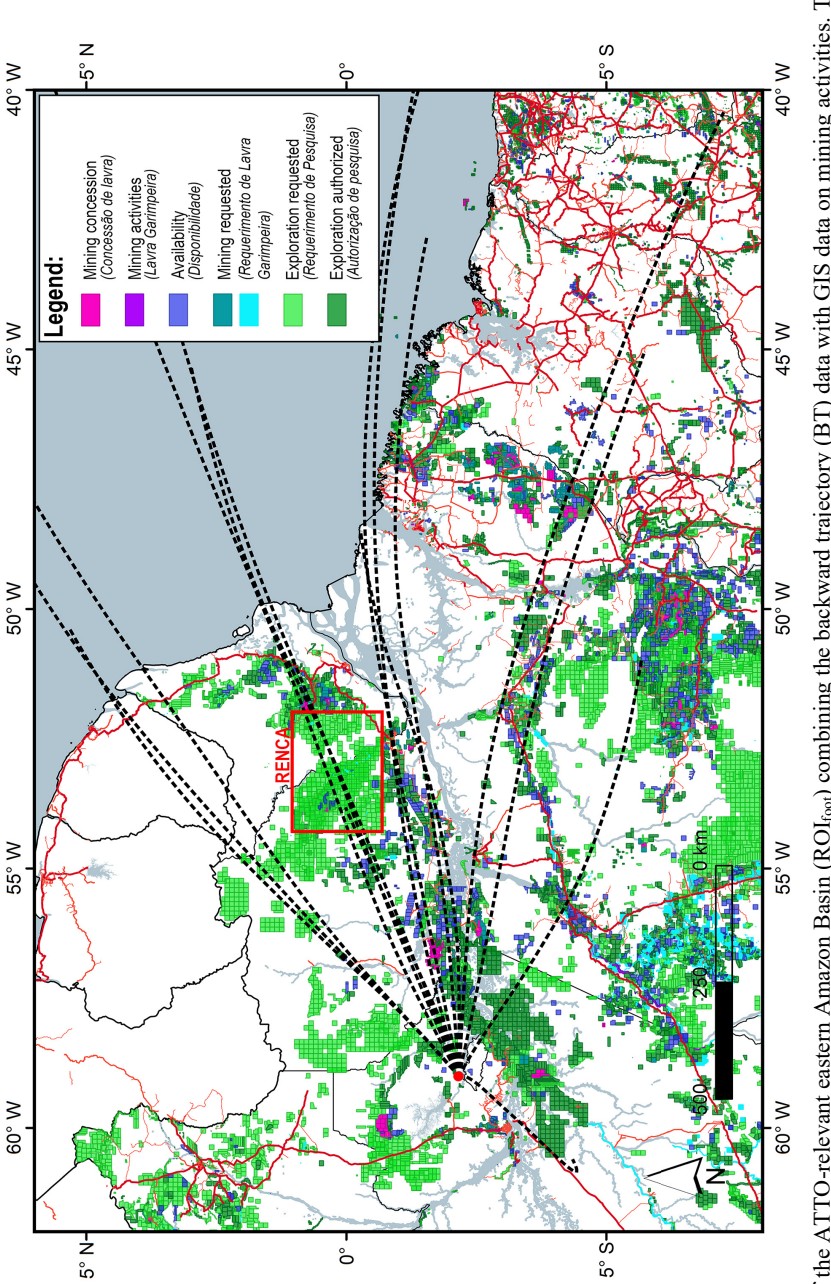

**Figure 19.** Map of the ATTO-relevant eastern Amazon Basin (ROI$_{foot}$) combining the backward trajectory (BT) data with GIS data on mining activities. The legend specifies several categories of mining activities according to SIGMINE (see Sect. S1.1). The BT data is represented as center lines of the 15 BT cluster (black dashed lines, see Fig. 4).



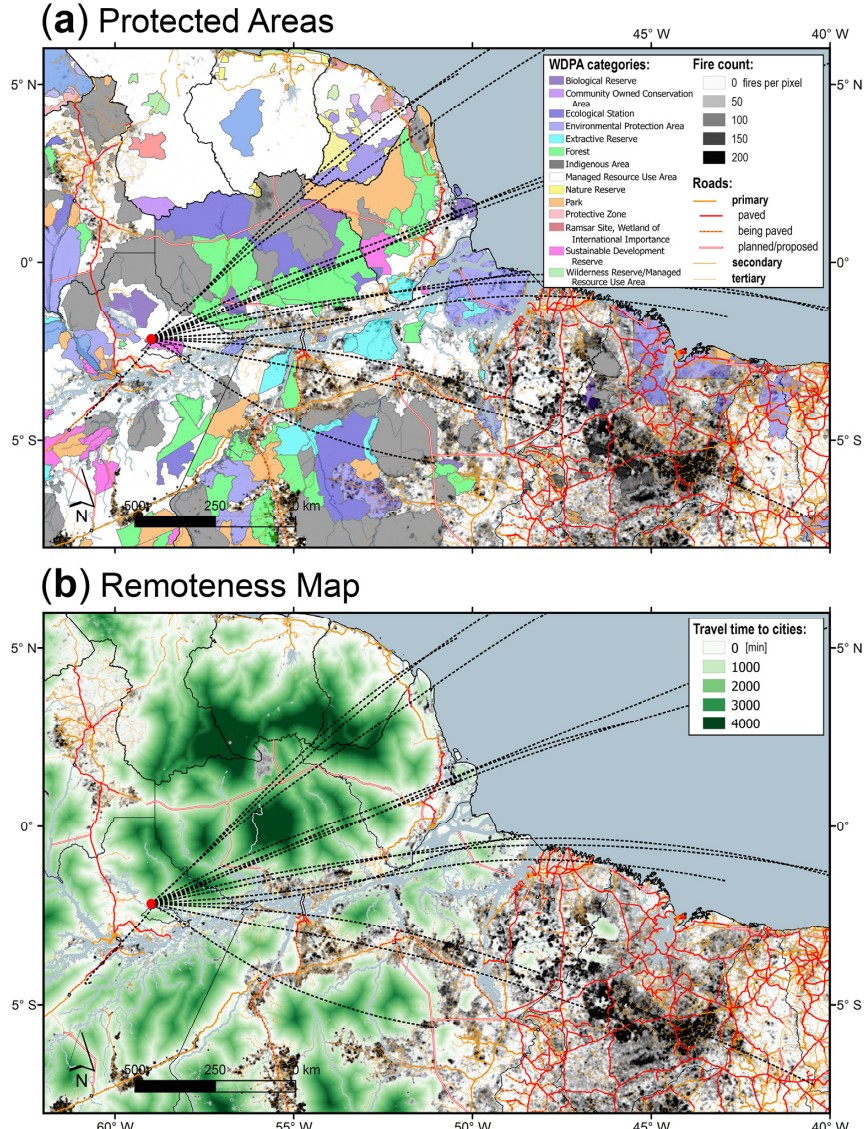

**Figure 20.** Map of the ATTO-relevant eastern Amazon Basin (ROI$_{foot}$) combining the backward trajectory (BT) data with GIS data layers of (**a**) protected areas according to the WDPA categorization as well as (**b**) a global accessibility map with land-based travel times to the nearest densely-populated area according to Weiss et al., (2018) (called 'remoteness map' here). In both maps, the road network from Fig. 17 and the INPE data-derived fire map from Fig. S16 has been adapted. The BT data is represented as center lines of the 15 BT cluster (black dashed lines, see Fig. 4).





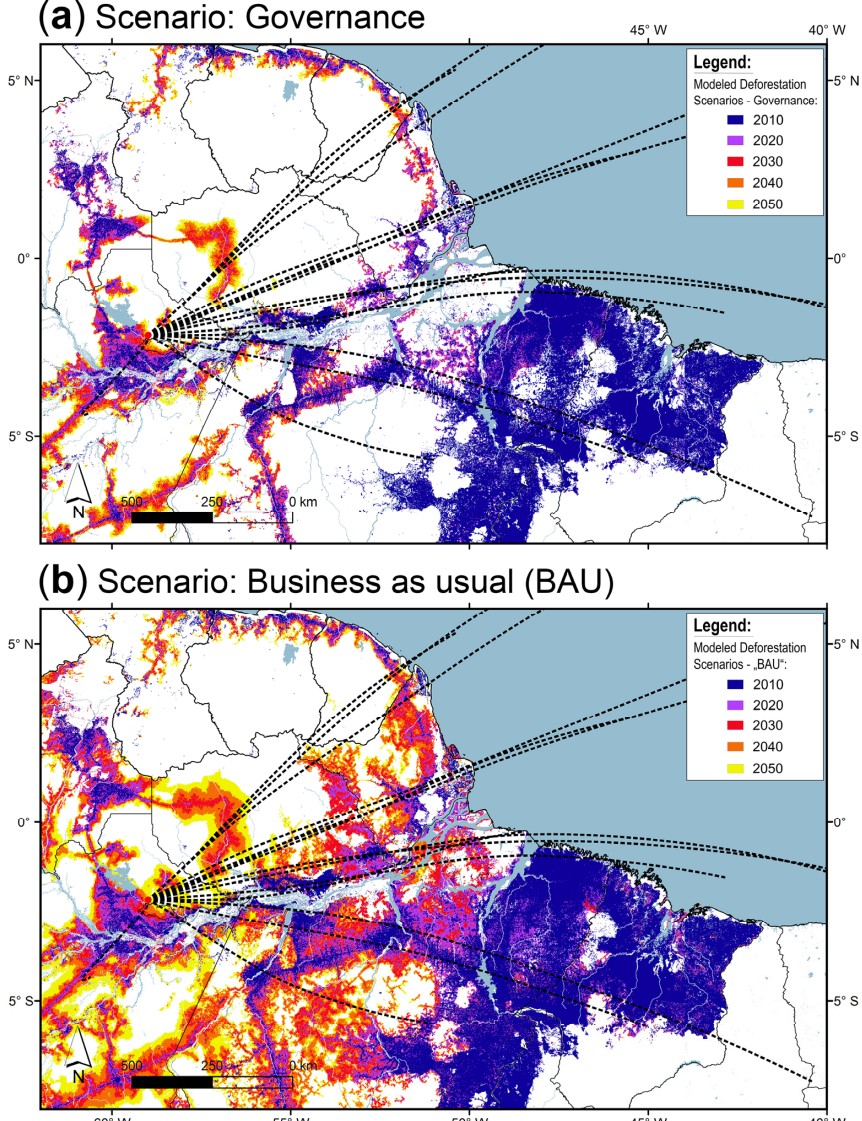

**Figure 21.** Map of the ATTO-relevant eastern Amazon Basin (ROI_foot) combining the backward trajectory (BT) data with GIS data layer on modeled deforestation scenarios, discriminating (**a**) "governance" and (**b**) "business as usual" cases according to Soares-Filho et al. (2006). The BT data is represented as center lines of the 15 BT cluster (black dashed lines, see Fig. 4).