# Peer review of "Land cover and its transformation in the backward trajectory footprint region of the Amazon Tall Tower Observatory"

_Atmospheric Chemistry and Physics, 2018_

## Referee Comment (RC1) · D. R. Fitzjarrald (Referee) · 28 Jun 2018

Review of:
**"Land cover and its transformation in the backward trajectory footprint region of the Amazon Tall Tower Observatory"**,

by Pöhlker and *twenty-seven* additional authors.

Reviewed by:  David Fitzjarrald, ASRC, University at Albany, SUNY

**General comments to the Editor.**

The ATTO team delivers manuscript that reads like notes from a committee white paper.  Clearly a lot of work has been done by some of the authors, but I sure would have been happier had more effort gone in to dealing with the ostensible topic of the paper, the utility of the backward trajectories that lead to the ATTO.

To demonstrate the importance of their effort, the authors offer up a litany of examples of Amazon Basin land cover change and deforestation, but, sadly, too many of these are references to the situation in the SW portion of the Basin, *not* upwind of the ATTO.  (One fine exception is the authors' noting of the location and importance of the Renca Reserve, which is upwind of the ATTO, a site that may or may not be a new locus for mining, depending on politics in Brazil.)

There appears to be a real disconnect between the authors' cavalier attitude that the trajectory results can be accepted intact with little question and the amount of detail presented in identifying the surface land-cover and fire presence categories. The authors assure the reader that: *"Trajectory models have been constantly improved, while gridded meteorological data became more sophisticated…."*  However, please note that there are *still* only a few radiosonde stations to furnish the reanalysis with extremely important input *upwind* of the ATTO (see map below, along with the preferred trajectories identified in this manuscript).  What's more, most of these stations are at the coast, and will not adequately represent the boundary layer inland. Granted, the wind is pretty steadily easterly, switching from NE to SE over the seasons, but the presence of large rivers means that local breeze circulations could significantly alter the trajectories in lower layers, precisely the ones that the authors want to emphasize.

Unless I missed it, there are only two mentions of the 'boundary layer' is in this manuscript (line 30 page 12, and line 38, page 32, where this point is repeated), in the context of discussing the global Hadley Cell and the climatology of the trade winds feeding in moisture.  A few references listed below indicate that identifying the HYSPLIT trajectories in this part of the world is not news anymore.  This is an off-the-shelf effort.  The authors are offering the readers this paper so it can be referenced in later work, but they owe the reader some more depth of understanding of the strengths *and weaknesses* of this approach.  The 'residence time' that matters ought to be the duration *and location* of the presence of the virtual parcel when the airmass is in 'communication'—turbulent connection—with the surface, which I imagine to be during the presence of the midday convective boundary layer.  When the authors mention HaPe Schmid's footprint work as a starting point, they don't carry over the idea that it is the turbulent mixing and subsequent diffusion that defines the 'tower footprint'.

That 1000 m level (close to the ATTO highest level) is likely be in the CBL for some daylight hours during the eastern Amazon Basin dry season, but it will likely be in the cloud layer during the midday wet season CBL (see the illustrative plots below—data from the LBA km67 site near Santarém PA). That will turn out to precess over the course of the day for parcels tracked back from different arrival hours at the ATTO site, and would look like to a 'dashed line' of activity as the trajectory crosses into the continent. It shouldn't really the duration, total time the air is in motion. One should take into account the convective hours of connection upwind, and then the hours of convection at the ATTO to present to the bedazzled readers the 'hot spots' of potential upwind influence, no?

[Figure]

Surface seasonal changes of surface state on a *T* vs *q* phase plane.; LCL, km67. Gray lines show isolines of LCL. Mean conditions for Belterra (near Santarém) and an average of Uruguayan grassland sites

[Figure]

Cloud base and LCL, km67. Black points mark cloud base. Red and magenta lines are LCL at forest km67 and a nearby pasture km77 sites in LBA. Horizontal blue line at 1km altitude.

[Figure]

Radiosonde sites in northern South America.

[Figure]

**Figure 4.** Map of northeast Amazon Basin with 15 clusters from systematic *k*-means back trajectory (BT)

D'Amelio, M. T. S., L. V. Gatti, J. B. Miller, and P. Tans. "Regional N 2 O fluxes in Amazonia derived from aircraft vertical profiles." *Atmospheric Chemistry and Physics* 9, no. 22 (2009): 8785-8797.

Lintner, B. R., & Neelin, J. D. (2010). Tropical South America–Atlantic sector convective margins and their relationship to low-level inflow. *Journal of Climate*, 23(10), 2671-2685.

Nieto, R., Gallego, D., Trigo, R., Ribera, P., & Gimeno, L. (2008). Dynamic identification of moisture sources in the Orinoco basin in equatorial South America. *Hydrological sciences journal*, 53(3), 602-617.

---

## Referee Comment (RC2) · B. Kruijt (Referee) · 16 Jul 2018

Before anything else, I have to declare my lack of detailed knowledge on back-trajectory methods and large-scale meteorology, as well as knowledge on many of the topics discussed in relation to the land surface over which the trajectories pass.

On the whole, this seems an extremely useful contribution to the interpretation of current and future ATTO science. It can potentially serve as a standard reference to most other publications and thus be highly cited.

The methodology to establish the back trajectories seems sound and comprehensive,

but again, I am not an expert on this. The manuscript provides an analysis of almost everything that happens along these trajectory paths, now and in the projected future.

I do have a few reservations though, and I suggest the authors take some action to address these:

1) The paper is very (overly) long. There are several extensive literature reviews embedded in the analysis that to my taste dig too deep into the backgrounds, which often carry whole science debates with them. E.g., where it concerns deforestation, citations refer to the impact of road building (and not everyone is convinced that roads are the main controls of deforestation); (lack of) seasonality is addressed in relation to the sometimes disputable notion that trees have very deep roots; Amazon 'die-back' is addressed as a potential future impact on the footprint properties, while this phenomenon is highly uncertain. I am sure this also holds for the other issues covered where my knowledge of the field is more limited. This carries the risk of being rather uncontrollable, hard to verify for bias. The manuscript does not set out a clear and rigorous strategy for review, so is not completely suited for the status of review paper. Also, there is no need to discuss the underlying science of these impacts in this manuscript, as it distracts from the main purpose: to serve as a reference for future ATTO science.

I did not strictly check, but it seems to me that even in the 'summary and conclusions' new issues are brought in.

I suggest the authors reduce contents here and limit themselves to merely listing potential issues affecting the trajectory, with limited key references.

2) I wonder how directly useful the presented format will be to future ATTO science. Perhaps the authors can synthesise the range of issues affecting the various classes of BT's in a more systematic way: for each class, provide a map, table or matrix quantifying the impact of the (three or five) MAIN impacts (co-ordinates, future year, impacts (1...5). This could be more readily be implemented in future analysis of ATTO results.

For the rest, the MS is well-written and well-documented. Figures are many, and might perhaps also be reduced somewhat, to support a more concise synthesis.

I wish the authors good luck with this extremely useful endeavour.

---

## Author Response (AR1)

**Response to referee comments and suggestions on acp-2018-323 by C. Pöhlker et al.: "Land cover and its transformation in the backward trajectory footprint region of the Amazon Tall Tower Observatory"**

**Manuscript format description:**
Black text shows the original referee comment, red text shows the authors response, and blue text shows quoted manuscript text. We used bracketed comment numbers for referee comments (e.g., [R1.1]) and author's responses (e.g., [A1.1]). Line numbers refer to the discussion/review manuscript.

**David Fitzjarrald as Referee #1**
Received: 28 June 2018

General comments:

[R1.1] The ATTO team delivers manuscript that reads like notes from a committee white paper. Clearly a lot of work has been done by some of the authors, but I sure would have been happier had more effort gone in to dealing with the ostensible topic of the paper, the utility of the backward trajectories that lead to the ATTO.

[A1.1] We appreciate the honest criticism by the referee and can confirm that all text was specifically written for this paper. We are further confident that the text of the manuscript has a higher quality than just a collection of "notes". With respect to the backward trajectories, the referee indeed pointed out some deficits in clarifying crucial aspects of the analysis as specified in detail in our response [A1.3] to [A1.5]. We are grateful for these hints.

[R1.2] To demonstrate the importance of their effort, the authors offer up a litany of examples of Amazon Basin land cover change and deforestation, but, sadly, too many of these are references to the situation in the SW portion of the Basin, *not* upwind of the ATTO. (One fine exception is the authors' noting of the location and importance of the Renca Reserve, which is upwind of the ATTO, a site that may or may not be a new locus for mining, depending on politics in Brazil.)

[A1.2] We are somewhat confused by the referee's comment that "sadly, too many of these are references to the situation in the SW portion of the Basin, *not* upwind of the ATTO". The SW part of the basin would include the states of Acre, Rondônia, and the southwest of Amazonas. However, Fig. 7 clearly shows that the region of interest ROI$_{foot}$, which defines the geographic extent of all subsequent maps (e.g., Fig. 10, 15 and 17), covers the NE and E states Roraima, Amapá, Pará as well as French Guiana and Surinam. The RENCA reserve is located in the center of the ROI$_{foot}$.

[R1.3] There appears to be a real disconnect between the authors' cavalier attitude that the trajectory results can be accepted intact with little question and the amount of detail presented in identifying the surface land-cover and fire presence categories. The authors assure the reader that: "*Trajectory models have been constantly improved, while gridded meteorological data became more sophisticated....*" However, please note that there are *still* only a few radiosonde stations to furnish the reanalysis with extremely important input *upwind* of the ATTO (see map below, along with the preferred trajectories identified in this manuscript). What's more, most of these stations are at the coast, and will not adequately represent the boundary layer inland. Granted, the wind is pretty steadily easterly, switching from NE to SE over the seasons, but the

presence of large rivers means that local breeze circulations could significantly alter the trajectories in lower layers, precisely the ones that the authors want to emphasize.

[A1.3] We agree with the referee that the mentioned aspects require further clarification. Therefore, we added the following text section in p. 5, l. 27 to clarify that the limited number of radiosonde station bears uncertainties to be considered carefully:

> For the HYSPLIT BT analysis presented here it is important to note that the meteorological reanalysis, which serves as input for the trajectory model, relies on comparatively few radiosonde stations in the Amazon region. Moreover, several of the station upwind of the ATTO site are located at the coast and, therefore, likely will not adequately represent the boundary layer height in the basin. The associated uncertainties have to be considered carefully in the interpretation of the BT results.

Moreover, the potential influence of river-related local breeze circulations has been mentioned explicitly by adding the following statement in p. 13, l. 5:

> A potential major uncertainty to keep in mind is the influence of local breeze circulations at the large rivers in the region that could significantly alter BTs in lower atmospheric layers.

Despite the aforementioned uncertainties, however, we are quite convinced that the overall trends in the BT spatiotemporal variability (even with quite some detail) are an appropriate representation of the ATTO-relevant atmospheric circulation. This impression is supported by (partly surprisingly) good agreements between BT patterns and atmospheric measurements at ATTO. Some of these links between BTs and atmospheric composition can be found in several recently published papers (i.e., Moran-Zuloaga et al., 2018; Pöhlker et al., 2018; Saturno et al., 2018).

[R1.4]  Unless I missed it, there are only two mentions of the 'boundary layer' is in this manuscript (line 30 page 12, and line 38, page 32, where this point is repeated), in the context of discussing the global Hadley Cell and the climatology of the trade winds feeding in moisture. A few references listed below indicate that identifying the HYSPLIT trajectories in this part of the world is not news anymore. This is an off-the-shelf effort. The authors are offering the readers this paper so it can be referenced in later work, but they owe the reader some more depth of understanding of the strengths *and weaknesses* of this approach. The 'residence time' that matters ought to be the duration *and location* of the presence of the virtual parcel when the airmass is in 'communication'—turbulent connection—with the surface, which I imagine to be during the presence of the midday convective boundary layer. When the authors mention HaPe Schmid's footprint work as a starting point, they don't carry over the idea that it is the turbulent mixing and subsequent diffusion that defines the 'tower footprint'.

[A1.4] See our response [A1.5].

[R1.5] That 1000 m level (close to the ATTO highest level) is likely be in the CBL for some daylight hours during the eastern Amazon Basin dry season, but it will likely be in the cloud layer during the midday wet season CBL (see the illustrative plots below—data from the LBA km67 site near Santarém PA). That will turn out to precess over the course of the day for parcels tracked back from different arrival hours at the ATTO site, and would look like to a 'dashed line' of activity as the trajectory crosses into the continent. It shouldn't really the duration, total time the air is in motion. One should take into account the convective hours of connection upwind, and then the hours of convection at the ATTO to present to the bedazzled readers the 'hot spots' of potential upwind influence, no?

**(a)**

[Figure]

Surface seasonal changes of surface state on a *T* vs *q* phase plane.; LCL, km67. Gray lines show isolines of LCL. Mean conditions for Belterra (near Santarém) and an average of Uruguayan grassland sites

**(b)**

[Figure]

Cloud base and LCL, km67. Black points mark cloud base. Red and magenta lines are LCL at forest km67 and a nearby pasture km77 sites in LBA. Horizontal blue line at 1km altitude.

**(c)**

[Figure]

[Figure]

Radiosonde sites in northern South America.

**(d)**

[Figure]

Figure 4. Map of northeast Amazon Basin with 15 clusters from systematic *k*-means back trajectory (BT)

**Referee Figure R1.**

[A1.5] We agree with referee that the relevance of the boundary layer for convective contact of the air masses with the ground deserves further and more systematic clarification. Accordingly, we recalculated a sequence of backward trajectory ensembles, taking the suggestion of the referee in his comments [R1.4] and [R1.5] into account. Specifically, we applied a sequence of (increasingly strict) filters to the BT ensembles to filter for (i) trajectory height, (ii) convective periods along the individual trajectories, and (iii) convective hours upon trajectory arrival at ATTO. The comparison of the results from the different filters shows some changes in the periphery of the footprint region, however, no fundamental impact on the shape for footprint's core region. The additional analysis as suggested by the referee has been implemented and clearly strengthens the manuscript. However, no major changes of the main aspects of the study became necessary since all derived footprints regions (filtered vs. unfiltered) have comparable geographic extents.

Specifically, section 2.5 has been edited with experimental details on the aforementioned analysis being implemented:

**2.5 Definition of backward trajectory-based ATTO footprint region**

[revised manuscript text omitted]

Moreover, a new figure has been added as Fig. S5 to the supplement file, which compares the resulting ATTO footprints based on the differently filtered trajectory ensembles:

[Figure]

**Figure S5.** Versions of ATTO BT footprint region based on differently filtered BT ensembles as specified in Table 1 and discussed in Sect. 2.5 and 3.3.

In the main text (p.17, l. 16) the following text block has been added:

The footprint shown in Fig. 7 (the base case) takes the entire BT ensemble into account. In order to assess to what the extent the individual BTs were in convective contact with the ground via the BL, we applied a sequence of filters to the base case BT ensemble (see Sect. 2.5). We found that the filtering does not substantially alter the geographic extent of the footprint's easterly core regions, whereas certain variations in the outer parts of the 25 % contour lines were observed (Fig. S5). With respect to the scope of this study, this indicates that the base case BT footprint is well suited to identify regions and land cover types that are of particular relevance for the ATTO research.

**References:**

Moran-Zuloaga, D., Ditas, F., Walter, D., Saturno, J., Brito, J., Carbone, S., Chi, X., Hrabě de Angelis, I., Baars, H., Godoi, R. H. M., Heese, B., Holanda, B. A., Lavrič, J. V., Martin, S. T., Ming, J., Pöhlker, M. L., Ruckteschler, N., Su, H., Wang, Y., Wang, Q., Wang, Z., Weber, B., Wolff, S., Artaxo, P., Pöschl, U., Andreae, M. O., and Pöhlker, C.: Long-term study on coarse mode aerosols in the Amazon rain forest with the frequent intrusion of Saharan dust plumes, Atmos. Chem. Phys., 18, 10055-10088, 10.5194/acp-18-10055-2018, 2018.

Pöhlker, M. L., Ditas, F., Saturno, J., Klimach, T., Hrabě de Angelis, I., Araùjo, A. C., Brito, J., Carbone, S., Cheng, Y., Chi, X., Ditz, R., Gunthe, S. S., Holanda, B. A., Kandler, K., Kesselmeier, J., Könemann, T., Krüger, O. O., Lavrič, J. V., Martin, S. T., Mikhailov, E., Moran-Zuloaga, D., Rizzo, L. V., Rose, D., Su, H., Thalman, R., Walter, D., Wang, J., Wolff, S., Barbosa, H. M. J., Artaxo, P., Andreae, M. O., Pöschl, U., and Pöhlker, C.: Long-term observations of cloud condensation nuclei over the Amazon rain forest – Part 2: Variability and characteristics of biomass burning, long-range transport, and pristine rain forest aerosols, Atmos. Chem. Phys., 18, 10289-10331, 10.5194/acp-18-10289-2018, 2018.

Saturno, J., Holanda, B. A., Pöhlker, C., Ditas, F., Wang, Q., Moran-Zuloaga, D., Brito, J., Carbone, S., Cheng, Y., Chi, X., Ditas, J., Hoffmann, T., Hrabe de Angelis, I., Könemann, T., Lavrič, J. V., Ma, N., Ming, J., Paulsen, H., Pöhlker, M. L., Rizzo, L. V., Schlag, P., Su, H., Walter, D., Wolff, S., Zhang, Y., Artaxo, P., Pöschl, U., and Andreae, M. O.: Black and brown carbon over central Amazonia: long-term aerosol measurements at the ATTO site, Atmos. Chem. Phys., 18, 12817-12843, 10.5194/acp-18-12817-2018, 2018.

**Response to referee comments and suggestions on acp-2018-323 by C. Pöhlker et al.: "Land cover and its transformation in the backward trajectory footprint region of the Amazon Tall Tower Observatory"**

**Manuscript format description:**
Black text shows the original referee comment, red text shows the authors response, and blue text shows quoted manuscript text. We used bracketed comment numbers for referee comments (e.g., [R1.1]) and author's responses (e.g., [A1.1]). Line numbers refer to the discussion/review manuscript.

**Bart Kruijt as Referee #2**
Received: 16 July 2018

General comment:

Before anything else, I have to declare my lack of detailed knowledge on back-trajectory methods and large-scale meteorology, as well as knowledge on many of the topics discussed in relation to the land surface over which the trajectories pass. On the whole, this seems an extremely useful contribution to the interpretation of current and future ATTO science. It can potentially serve as a standard reference to most other publications and thus be highly cited. The methodology to establish the back trajectories seems sound and comprehensive, but again, I am not an expert on this. The manuscript provides an analysis of almost everything that happens along these trajectory paths, now and in the projected future.

Author response: We appreciate that Referee #2 considers the study "an extremely useful contribution to the interpretation of current and future ATTO science". We further appreciate the constructive criticism that helped to improve the quality of the manuscript.

Specific comments:

[R2.1] The paper is very (overly) long.

[A2.1] We are aware that the length of the study can be a burden for its (linear) reading. Therefore, we paid particular attention during writing to short and concise formulations throughout the entire text as well as a clear overall structure with generic subtitles for all sections. Moreover, the figures have been prepared carefully to make them appealing and informative even while browsing over the study. The figure captions are comprehensive enough to clarify key aspects of the study without reading the entire text. In particular, Sect. 3.3. has been structured in seven subsections (3.3.1 to 3.3.7) that can be read independently from each other and, thus, facilitate nonlinear reading for those readers looking for specific aspects.

In response to the referee's comment, we reworked the opening paragraph of the results and discussion part by adding a dedicated statement of the organization and structure of the text. We anticipate that this paragraph may act as a guideline, helping to make best use of this study as a resource and look-up reference. The following text section has been implemented into p. 17, l. 1:

The results and discussion part of this manuscript consists of two major parts:
- **Sections 3.1** and **3.2** summarize the large-scale geographic patters and seasonal variability of the ATTO BT ensembles as well as their links to precipitation regimes and selected teleconnections.
- **Section 3.3** defines a BT-based footprint region of the ATTO site and relates it to the current state and anticipated future change of the covered land use mosaic.

Particularly, Sect. 3.3 is meant to be a resource and look-up reference summarizing ATTO-relevant land cover information subdivided into the following categories:
- **Sect. 3.3.1**: Climatic conditions, biomes, ecoregions and the "last of the wild"
- **Sect. 3.3.2**: Land cover
- **Sect. 3.3.3**: Deforestation and agro-industrial expansion
- **Sect. 3.3.4**: Fires
- **Sect. 3.3.5**: Infrastructure, cities, traffic and mining
- **Sect. 3.3.6**: Protected areas
- **Sect. 3.3.7**: Deforestation and climate change scenarios

All seven sections 3.3.1 to 3.3.7 begin with a concise literature synthesis section and then relate the discussion to its specific relevance for the ATTO research. Due to its length, the entire Sect. 3.3 has been structured and written in a way that facilitates non-linear reading of specific aspects on interest.

[R2.2] There are several extensive literature reviews embedded in the analysis that to my taste dig too deep into the backgrounds, which often carry whole science debates with them. E.g., where it concerns deforestation, citations refer to the impact of road building (and not everyone is convinced that roads are the main controls of deforestation); (lack of) seasonality is addressed in relation to the sometimes disputable notion that trees have very deep roots; Amazon 'die-back' is addressed as a potential future impact on the footprint properties, while this phenomenon is highly uncertain. I am sure this also holds for the other issues covered where my knowledge of the field is more limited. This carries the risk of being rather uncontrollable, hard to verify for bias. The manuscript does not set out a clear and rigorous strategy for review, so is not completely suited for the status of review paper. Also, there is no need to discuss the underlying science of these impacts in this manuscript, as it distracts from the main purpose: to serve as a reference for future ATTO science.

[A2.2] That is true. Section 3 has several literature synthesis section embedded. The purpose of embedding those sections was not to "dig" particularly deep into ongoing "science debates", but rather to explicitly link the ATTO research to the extended body of literature on land cover observations in Amazonia. As stated in p. 4, l. 12-13: "We envision that this work may serve as a helpful resource and look-up reference for the interpretation of current and future observations in the region." We aimed to make the literature synthesis sections as concise as possible to provide the interested readers a starting point on the issues/debates along with several references for further reading. We understand that this bears a certain risk of being biased. We are convinced, however, that all our statements in the literature synthesis sections are transparently connected to the corresponding references, facilitating further in-depth literature research by the readers.

Relating to the specific aspects criticized by the referee, we modified the corresponding text sections as follows:

In p. 26, l. 28:

(iv) major highways as key drivers for forest fragmentation and degradation

has been replaced by:

(iv) major highways as drivers for forest fragmentation and degradation

Moreover, in p. 27, l. 35, the following statement has been added:

However, it is still being debated to what extent roads have acted as main deforestation controls.

In p. 21, l. 16, the statement:

Apparently, the increasing drought stress in the dry season is buffered by the deep-rooting trees in the moist soils and, therefore, does not (significantly) affect the NDVI (Nepstad et al., 2008).

has been modified to:

Presumably, the increasing drought stress in the dry season is buffered by the comparatively deep-rooting trees in the moist soils and, therefore, does not (significantly) affect the NDVI (Nepstad et al., 2008).

In p. 31, l. 2, the statement:

In extreme scenarios, a large-scale rain forest die-back – i.e., a climate-driven substitution of moist forests by semi-arid and/or savanna vegetation – due to changing hydrological and seasonal regimes has been predicted (e.g., Cochrane and Laurance, 2008; Nepstad et al., 2008; Cochrane and Barber, 2009).

has been modified to:

In extreme scenarios, a large-scale rain forest die-back – i.e., a climate-driven substitution of moist forests by semi-arid and/or savanna vegetation – due to changing hydrological and seasonal regimes has been predicted, although these predictions still comprise large uncertainties (e.g., Cochrane and Laurance, 2008; Nepstad et al., 2008; Cochrane and Barber, 2009).

[R2.3] I did not strictly check, but it seems to me that even in the 'summary and conclusions' new issues are brought in. I suggest the authors reduce contents here and limit themselves to merely listing potential issues affecting the trajectory, with limited key references.

[A2.3] We appreciate this comment, which we took into account to shorten and streamline the summary and conclusions section. Moreover, we counterchecked whether new aspects are brought in here and can confirm that all aspects in the summary and conclusions section have been introduced and discussed in the main text already.

[R2.4] I wonder how directly useful the presented format will be to future ATTO science. Perhaps the authors can synthesise the range of issues affecting the various classes of BT's in a more systematic way: for each class, provide a map, table or matrix quantifying the impact of the (three or five) MAIN impacts (co-ordinates, future year, impacts (1...5). This could be more readily be implemented in future analysis of ATTO results.

[A2.4] Thanks for this suggestion. In a way, we have already tried to realize what the referee seems to suggest by strictly formalizing the scope and layout of the various maps throughout the text. Particularly in Sect. 3.3, the maps summarize the key aspects of the various land cover categories in direct relation to the ATTO-relevant BT information (i.e., BT clusters being plotted in each map for reference). Wherever possible, we further extracted quantitative information, resolved by the main BT directions (see for instance Fig. 11, Fig. 14, Fig. 16, Table S1, Table S2, Fig. S4, Fig. S15, Fig. S17). Beyond that, a quantification of impacts seems difficult for certain land cover classes. Ultimately, we have not found a better strategy how to further synthesize the range of issues presented.

[R2.5] For the rest, the MS is well-written and well-documented. Figures are many, and might perhaps also be reduced somewhat, to support a more concise synthesis.

[A2.5] In the course of writing the manuscript we moved several figures into the supplement already. We feel that the figures currently shown in the main text are required to support the main observations and conclusions. In general, most of the figures (15 out of 21) are maps, which are rather self-explanatory and easy to 'digest' for the readers. Accordingly, we prefer to refrain from further reducing the number of figures in the main text.

[R2.6] I wish the authors good luck with this extremely useful endeavour.

[A2.6] Thanks a lot!

[revised manuscript text omitted]

**List of changes**

All relevant changes have been marked blue in the revised version of the manuscript. The most important changes are listed below. Page and line numbering refers to the ACPD version of the manuscript.

Author List:

The name "Ruckteschler" has been replaced by "Löbs".

Tables:
Table 1 has been added to the manuscript:

**Table 1.** Filters applied to the multi-year BT ensemble. Height filter crops segments along individual BTs if values exceed height threshold (e.g., height of 1000 m; named "H1000"). *En route* convection filter crops segments along individual BTs if sun flux values are below threshold (named "Cer"). Convection filter at ATTO removed entire BTs from analysis if sun flux upon arrival (i.e., in the ATTO pixel) is below threshold (named "Catto").

| filters | height filter [m] | convection filter *en route* [W m$^{-2}$] | convection filter at ATTO [W m$^{-2}$] |
|---|---|---|---|
| No Filter | none | none | none |
| H1500 | <1500 | none | none |
| H1000 | <1000 | none | none |
| H0500 | <500 | none | none |
| H1500_Cer | <1500 | >50 | none |
| H1000_Cer | <1000 | >50 | none |
| H0500_Cer | <500 | >50 | none |
| H1500_Cer_Catto | <1500 | >50 | >50 |
| H1000_Cer_Catto | <1000 | >50 | >50 |
| H0500_Cer_Catto | <500 | >50 | >50 |

General changes:

Page 5, line 20: The following section has been added:
"For the HYSPLIT BT analysis presented here it is important to note that the meteorological analysis in GDAS, which serves as input for the trajectory model, relies on comparatively few radiosonde stations in the Amazon region and that several of the radiosonde stations upwind of ATTO are located at the coast and, therefore, likely will not adequately represent the boundary layer height in the basin. However, GDAS also uses a global data set of surface observations, wind profiler data, aircraft reports, buoy observations, radar observations, and satellite observations, in its 3-D model assimilation, which mitigates the scarcity of available radiosondes. The associated uncertainties have to be considered carefully in the interpretation of the BT results."

Page 8, line 34: The following section has been added:
"The sensitivity of the geographic extent of the BT footprint region towards different BL heights and convective mixing inside the BL has been tested by applying dedicated filters to the BT data. We calculated different versions of the BT footprint region on the basis of multi-year BT ensembles with

[revised manuscript text omitted]

Page 13, line 5: The following section has been added:
"A potential major uncertainty to keep in mind is the influence of local breeze circulations at the large rivers in the region that could significantly alter BTs in lower atmospheric layers. However, in a previous study we found that HYSPLIT performed quite well in reproducing the river breeze circulation over the Amazon River near Manaus (Trebs et al., 2012)."

Page 17, line 16: The following section has been added:
"The footprint shown in Fig. 7 (the base case) takes the entire BT ensemble into account. In order to assess to what the extent the individual BTs were in convective contact with the ground via the BL, we applied a sequence of filters to the base case BT ensemble (see Sect. 2.5). We found that the filtering does not substantially alter the geographic extent of the footprint's easterly core regions, whereas certain variations in the outer parts of the 25 % contour lines were observed (Fig. S5). With respect to the scope of this study, this indicates that the base case BT footprint is well suited to identify regions and land cover types that are of particular relevance for the ATTO research."

Page 32, line 1: The summary and conclusions section has been modified from:

[revised manuscript text omitted]

Supplement:

The supplement file has been updated and a new Fig S5 has been added.

---

## Referee Report (RR1)

Second review of:
**"Land cover and its transformation in the backward trajectory footprint region of the Amazon Tall Tower Observatory"**,
by Pöhlker and *twenty-seven* additional authors.

Reviewed by:  David Fitzjarrald, ASRC, University at Albany, SUNY

**General comments to the Editor and the Authors.**

I do not understand how this journal operates, what with the paper disappearing and reappearing, but so be it! It warms my hard reviewer's heart to see that some of my recommendations were adopted, and, to me, the paper is in much better shape.  Once the Editor is satisfied with the authors' response to the limited issue I raise here, it should be approved.

A key improvement is the good-faith effort to refine the trajectory estimates to allow for the diurnally-intermittent mixing occurring during the convective part of the day.  This will be the period when the air 'parcel' is most influenced by the surface conditions.  In some detail, both in the body of the text and in the Supplementary materials, the authors lay out a their approach, using the information available in the HYSPLIT package, to identify convective conditions.  I think all involved realize that this trajectory approach is likely compromised in the rainy season because of the deep cloud updrafts, but that the situation during the dry season is more amenable to the approach.  I appreciate the new comments that own up to limitations of this type of analysis.

A key question I have is how relevant that Amazon land cover details are.  (I don't see how the identification of Saharan dust in the Amazon validates this part of the study.)

One thing that puzzles me is that their revised 'footprints' have no indication of how the transient areal influences contribute to what the parcel carries when arriving at the ATTO.  All that is demonstrated is that the upwind 'footprint' is not much changed by including this effect. Just how does work?

When I wrote (not too precisely): "That will turn out to precess over the course of the day for parcels tracked back from different arrival hours at the ATTO site, and would look like to a 'dashed line' of activity as the trajectory crosses into the continent.",  I meant for the authors to comment on this intermittent representativeness issue.  Did I miss something? What about this?  Only with this can one visually identify the regions that get "special attention" when measurements at the tower are examined.  I realize that there is likely appreciable horizontal diffusion, but the authors have gone this far with the most elementary way to use HYSPLIT to address this issue, why not a take the small additional step to explain the effort more clearly?

I made a rough cartoon to illustrate what I mean.  The trajectory of the flow inland from the coast can be thought of as a 'characteristic', in the sense one uses to solve certain kinds of differential equations.  Based on analysis of cloud base wind speed from the Santarém soundings that I did for a recent paper (Kivalov and Fitzjarrald, 2018), $\approx 8$ m/s is a fair estimate.  A rough estimate of the distance from the coast to ATTO is about 1200 km, and this gives a travel time of about 40 hours.  I would think that the time of day for the boundary layer to be 'coupled' to cloud base to be roughly a third of the day, 8 LT – 16 LT—look at the cloud reports on the ceilometer graph in the original review.  Anyway, the intersections of the characteristic curve with these time of day bands leads to corresponding bands of longitude, indicating the regions that are more properly linked to the air mass.  The band along the coast is surely compromised by breeze effects, but that is just one price of the simplification in using the HYSPLIT approach. Could you not mention and show what these bands are?  Could you not note to what degree of

specificity one must know the land use categories to comment, as the authors do, on the upwind surface conditions?  In particular, in view of this situation, can you comment on whether or not the presence of mining upwind is likely to be detectec at ATTO?

Kivalov, S.N. and Fitzjarrald, D.R., 2018. Quantifying and modelling the effect of cloud shadows on the surface irradiance at tropical and midlatitude forests. *Boundary-Layer Meteorology, 166*(2), pp.165-198.

---

## Author Response (AR2)

**Response to referee comments and suggestions on acp-2018-323 by C. Pöhlker et al.: "Land cover and its transformation in the backward trajectory footprint region of the Amazon Tall Tower Observatory"**

**Manuscript format description:**
Black text shows the original referee comment, red text shows the authors response, and blue text shows quoted manuscript text. We used bracketed comment numbers for referee comments (e.g., [R1.1]) and author's responses (e.g., [A1.1]). Line numbers refer to the discussion/review manuscript.

**David Fitzjarrald as Referee #1**
Received: 05 April 2019

General comments:

[R1.1] It warms my hard reviewer's heart to see that some of my recommendations were adopted, and, to me, the paper is in much better shape. Once the Editor is satisfied with the authors' response to the limited issue I raise here, it should be approved. A key improvement is the good-faith effort to refine the trajectory estimates to allow for the diurnally-intermittent mixing occurring during the convective part of the day. This will be the period when the air 'parcel' is most influenced by the surface conditions. In some detail, both in the body of the text and in the Supplementary materials, the authors lay out a their approach, using the information available in the HYSPLIT package, to identify convective conditions. I think all involved realize that this trajectory approach is likely compromised in the rainy season because of the deep cloud updrafts, but that the situation during the dry season is more amenable to the approach. I appreciate the new comments that own up to limitations of this type of analysis.

[A1.1] We appreciate this positive feedback by the referee.

[R1.2] A key question I have is how relevant that Amazon land cover details are. (I don't see how the identification of Saharan dust in the Amazon validates this part of the study.) One thing that puzzles me is that their revised 'footprints' have no indication of how the transient areal influences contribute to what the parcel carries when arriving at the ATTO. All that is demonstrated is that the upwind 'footprint' is not much changed by including this effect. Just how does work? When I wrote (not too precisely): "That will turn out to precess over the course of the day for parcels tracked back from different arrival hours at the ATTO site, and would look like to a 'dashed line' of activity as the trajectory crosses into the continent.", I meant for the authors to comment on this intermittent representativeness issue. Did I miss something? What about this? Only with this can one visually identify the regions that get "special attention" when measurements at the tower are examined. I realize that there is likely appreciable horizontal diffusion, but the authors have gone this far with the most elementary way to use HYSPLIT to address this issue, why not a take the small additional step to explain the effort more clearly? I made a rough cartoon to illustrate what I mean. The trajectory of the flow inland from the coast can be thought of as a 'characteristic', in the sense one uses to solve certain kinds of differential equations. Based on analysis of cloud base wind speed from the Santarém soundings that I did for a recent paper (Kivalov and Fitzjarrald, 2018), ≈ 8 m/s is a fair estimate. A rough estimate of the distance from the coast to ATTO is about 1200 km, and this gives a travel time of about 40 hours. I would think that the time of day for the boundary layer to be 'coupled' to cloud base to be roughly a third of the day, 8 LT – 16 LT—look at the cloud reports on the ceilometer graph in the original review. Anyway, the intersections of the characteristic curve with these time of day bands leads to corresponding bands of longitude, indicating the regions that are more properly linked to the air mass. The band along the

coast is surely compromised by breeze effects, but that is just one price of the simplification in using the HYSPLIT approach. Could you not mention and show what these bands are? Could you not note to what degree of specificity one must know the land use categories to comment, as the authors do, on the upwind surface conditions?

[Figure]

**Referee Figure R2.**

[A1.2] We appreciate that the referee emphasizes the "intermittent representativeness issue" and recommends its explicit discussion in the text. In fact, we observed the influence of the "diurnally-intermittent mixing" (dashed-line BTs) in the course of the analysis that we conducted for the first round of revision. It is correct, however, that the additional plots we prepared so far did not visualize this effect. Accordingly, we revisited the analysis, modified the text, and prepared an additional figures to emphasize the "diurnally-intermittent mixing" and its relevance for the discussion of land cover within the BT footprint. The corresponding changes in the manuscript and supplement are outlined below.
In Sect. 3.3, the former paragraphs:

[revised manuscript text omitted]

[R1.3] In particular, in view of this situation, can you comment on whether or not the presence of mining upwind is likely to be detectec at ATTO?

[A1.3] With respect to the newly added Fig. 8 with the refined footprint analysis (i.e., diurnally-intermittent mixing taken into account), we addressed this question by adding the following statement to Sect. 3.3.5:

> According to the refined BT footprint in Fig. 8, which takes diurnally-intermittent mixing along the BTs into account, the RENCA area is located within a region of very effective convective coupling between the ground and overpassing air masses, whereas the aforementioned bauxite mine is located in an area with much less effective coupling. This observation emphasizes that potential future mining in the RENCA area may be of significant relevance to the ATTO observations with potentially even higher impacts than those caused by the much closer bauxite mine.

- Closed to open (>15%) shrubland (<5m) (**130**)
- Closed to open (>15%) herbaceous vegetation (grassland/savannas/lichens) (**140**)
- Sparse (<15%) vegetation (**150**)
- Closed to open (>15%) broadleaved forest regularly flooded (**160**)
- Closed (>40%) broadleaved forest or shrubland permanently flooded (**170**)
- Closed to open (>15%) grassland or woody vegetation on regul. flooded waterlogged soil (**180**)
- Artificial surfaces & associated areas (urban areas >50%) (**190**)
- Bare areas (**200**)
- Water bodies (**210**)

[revised manuscript text omitted]